# Quantifying Uncertainties from Mobile Laboratory Derived Emissions of Well Pads Using Inverse Gaussian Methods

Dana R. Caulton[1], Qi Li[2], Elie Bou-Zeid[1], Jeffrey P. Fitts[1], Levi M. Golston[1], Da Pan[1], Jessica Lu[1], Haley M. Lane[1], Bernhard Buchholz[3], Xuehui Guo[1], James McSpiritt[1], Lars Wendt[4] and Mark A. Zondlo[1]

[1]Department of Civil and Environmental Engineering, Princeton University, 59 Olden St., Princeton, NJ 08540, U.S.A.
[2]Department of Earth and Environmental Engineering, Columbia University, 500 W 120th St., New York, NY 10027, U.S.A.
[3]RMS, Technische Universität Darmstadt, Darmstadt, 64287, Germany
[4]Hunterdon Central Regional High School, Flemington, NJ 08822, U.S.A.

*Correspondence to*: M. A. Zondlo (mzondlo@princeton.edu)

**Abstract.** Mobile laboratory measurements provide information on the distribution of $CH_4$ emissions from point sources such as oil and gas wells, but uncertainties are poorly constrained or justified. Sources of uncertainty and bias in ground-based Gaussian derived emissions estimates from a mobile platform were analyzed in a combined field and modeling study. In a field campaign where 1009 natural gas sites in Pennsylvania were sampled, a hierarchical measurement strategy was implemented with increasing complexity. Of these sites, ~93% were sampled with an average of 2 transects in <5 min (standard sampling), ~5% were sampled with an average of 10 transects in <15 min (replicate sampling) and ~2% were sampled with an average of 20 transects in 15-60 min. For sites sampled with 20 transects, a tower was simultaneously deployed to measure high-frequency meteorological data (intensive sampling). Five of the intensive sampling sites were modeled using large eddy simulation (LES) to reproduce $CH_4$ concentrations in a turbulent environment. The LES output and derived emission estimates were used to compare with the results of a standard Gaussian approach. The LES and Gaussian-derived emission rates agreed within a factor of 2 in all except one case; the average difference was 25%. A controlled release was also used to investigate sources of bias in either technique. The Gaussian agreed with the release rate more closely than the LES underlying the importance of inputs as sources of uncertainty for the LES. The LES was also used as a virtual experiment to determine an optimum number of repeat transects and spacing needed to produce representative statistics. Approximately 10 repeat transects spaced at least 1 min apart are required to produce statistics similar to the observed variability over the entire LES simulation period of 30 min. Sources of uncertainty from source location, wind speed, background concentration and atmospheric stability were also analyzed. The largest contribution to the total uncertainty was from atmospheric variability; this is caused by insufficient averaging of turbulent variables in the atmosphere (also known as random errors). Atmospheric variability was quantified by repeat measurements at individual sites under relatively constant conditions. Accurate quantification of atmospheric variability provides a reasonable estimate of the lower bound for emission uncertainty. The uncertainty bounds calculated for this work for sites with > 50 ppb enhancements were 0.05q-6.5q (where q is the emission rate) for single transect sites and 0.5q-2.7q for sites with 10+ transects. More transects allow a mean emission rate to be calculated with better precision. It is recommended that future mobile monitoring schemes quantify atmospheric variability, and attempt to minimize it, under representative conditions to

accurately estimate emission uncertainty. These recommendations are general to mobile laboratory derived emissions from other sources that can be treated as point sources.

## 1 Introduction

Reducing emissions of short-lived greenhouse gases through regulations has been considered a potentially viable way to mitigate climate change without intensively regulating $CO_2$, which poses economic and political challenges. In particular, reducing $CH_4$, a potent greenhouse gas and the main component of natural gas, may have significant immediate climate change mitigation benefits (Bowerman et al., 2013, Baker et al., 2015, Zickfeld et al., 2017). However, the large numbers and types of components in the natural gas supply chain that may leak require the development of efficient and accurate methods to quantify emissions. Specifically, techniques are needed that are available to researchers at every level (government, industry and academic), accurate enough to locate and quantify specific sources of fugitive emissions and allow for self-monitoring, independent verification and understanding of common leak sources. The range of emissions from these sources can be large as studies have shown that some sources, such as natural gas well pads, have a lognormal distribution where emissions span several orders of magnitude. The most critical target for mitigation efforts are the super-emitters, where the top 5% of samples can contribute ~50% of emissions (Brandt et al., 2016).

To this end, various independent $CH_4$ emission estimation techniques have been implemented. Table 1 shows a brief summary of the methods that have been primarily applied to oil, coal and gas extraction and infrastructure which account for ~30% of the total global anthropogenic $CH_4$ emissions (Kirschke et al., 2013). The myriad of sites and types of emission sources have necessitated the development and application of multiple techniques. Examples include satellites (Kort et al., 2014), remote sensing from aircraft (Kuai et al., 2016, Frankenberg et al., 2016, Thorpe et al., 2016), in-situ aircraft measurements (Karion et al., 2013 and 2014, Peischl et al., 2013 and 2015, Caulton et al., 2014, Pétron et al., 2014, Lavoie et al., 2015), long-term monitoring from short and tall towers (Pétron et al., 2012), unmanned aerial vehicles (Nathan et al., 2015) and various ground-based techniques. Ground-based techniques include flask sampling (Townsend-Small et al., 2015), tracer correlation techniques (Lamb et al., 2015, Subrumanian et al., 2015, Zimmerle et al., 2015, Omara et al., 2016), chamber sampling (Allen et al., 2013 and 2014, Kang et al., 2014), thermal/optical imaging (Gålfalk et al., 2015, Ravikumar et al., 2017) and combined measurement/dispersion modeling techniques from stationary (Brantley et al., 2014, Foster-Witting et al., 2015) and mobile platforms (Lan et al., 2015, Rella et al., 2015, Yacovitch et al., 2015).

Every technique has various advantages and disadvantages related to operational cost, sampling efficiency, processing time and uncertainty. These techniques have been used in different ways, from direct point source emission estimation, to area source emission rate estimation, and they use data that may span a few hours up to years. For example, Kort et al. (2014) used data spanning over 6 years at 0.33° resolution covering the entire U.S., to estimate a large emission rate. Karion et al. (2013) reported emissions from a large natural gas field (~60 km diameter) using aircraft mass-balance with data collected within a few hours while Caulton et al. (2014) reported individual well pad emissions (<1 km diameter)

using the same technique. It is also feasible that as new instruments and data processing techniques become available, any of these techniques may be used at spatial and temporal scales not represented by the works cited here. Table 1 compares the author reported uncertainties for several techniques; this table is intended to illustrate the motivation for this work and not as an exhaustive review of each of these techniques. These uncertainty estimates should be compared with caution: self-reported uncertainties are not computed in identical manners and some may not include the same sources of uncertainty in their considerations or additional systematic biases to the same extent. Most of the author reported uncertainties appear to correspond to 1 standard deviation measurements, though, the ways in which these statistics are derived are not consistent. Notably, Kort et al. (2014) report a 2 standard deviation range and Pétron et al. (2012) reports a minimum and maximum range for emissions. In addition, several studies report 95% confidence intervals, such as all ground-based mobile dispersion estimates, likely because of the asymmetric uncertainty that is more easily reported in this manner. The values reported for *emission* uncertainties, usually significant, are generally not from *measurement* uncertainties. Even instruments that produce high accuracy measurements that must be transformed into an emission rate can be confounded by transformation methods that rely upon limited or unrepresentative meteorological conditions such as small scale turbulence characterization, boundary layer processes or assigning background conditions to a variable atmosphere. Notable, however, is the large range of uncertainties reported by ground-based mobile dispersion techniques. These techniques rely on accurate and precise concentration measurements coupled with dispersion models (Gaussian, AERMOD, WindTrax, etc.) to produce an emission estimate and are subject to various uncertainties in the model, notably atmospheric diffusion coefficients. These techniques are attractive due to their relatively low cost, low computational requirements and high sampling efficiency of individual sources. However, as seen in Table 1, current uncertainty emission estimates using ground based and mobile measurements seem to underperform relative to other measurement techniques in this field. Therefore, there is a critical need for improved sampling methods and/or data processing in this area to improve data quality and better constrain uncertainty.

## 1.1 Theory of the Gaussian Plume Model

Approximations of scalar dispersion were investigated as early as the 1930s and were developed to describe non-reactive pollutant dispersal from elevated stacks (Sutton, 1932; Bosanquet and Pearson, 1936). As models improved, the Gaussian plume model was developed assuming that a scalar concentration has a normal distribution function (Batchelor, 1949, Hilst, 1957). Additional investigation of near surface conditions where particles can either deposit to, or reflect off, the surface led to the current Gaussian plume analytical model (shown in Eq. 1) to predict scalar concentrations (C), which can be directly derived from the advection–diffusion equation under some simplifying assumptions (as explained in Veigele and Head, 1978). Variations of this equation can be found in many papers and textbooks (for example Gifford, 1968, Zannetti, 1990). The Gaussian model with a reflective ground ($CH_4$ is reflected off the surface) is presented here:

$$C(x,y,z) = \frac{Q}{2\pi\sigma_y\sigma_z u} \cdot e^{\frac{-y^2}{2\sigma_y^2}} \cdot \left[ e^{\frac{-(z-h)^2}{2\sigma_z^2}} + e^{\frac{-(z+h)^2}{2\sigma_z^2}} \right] \tag{1}$$

This function relies on the 3-D distances (x,y,z) of a receptor from a source as well as the source height h, mean horizontal wind speed u, and a source strength, Q. Here, the source is defined as the origin with x the downwind distance, y the crosswind distance and z the measurement height above ground. The dispersion coefficients ($\sigma_y$, $\sigma_z$) encode the strength of turbulent mixing or diffusivity, as well as the downwind distance over which the mixing is acted, *x*, which does not appear directly in Eq. 1. They are calculated according to any of several analytical parameterizations based on Pasquill-Gifford's stability class scheme. Atmospheric stability classes range from very unstable (A) to very stable (F). Class D is defined as neutral. The model describes the distribution of a scalar concentration (C) downwind of a source, meaning it describes average plume locations and concentrations. However, the instantaneous observed plume structure deviates greatly from the average behavior, with fluctuating peak concentration magnitude and location. The comparison of instantaneous and modeled concentrations is thus impacted by the averaging timescale associated with the measurements. Studies suggest that appropriate time scales depend on downwind distance, *x*, and stability ranging from 2-60 min (Fritz et al. 2005). For example, in class D at 200 m downwind, 3 min would be sufficient when using the Gifford dispersion coefficients. Insufficient averaging would result in random errors that we refer to in this work as the uncertainty related to 'atmospheric variability.'

The Gaussian plume model is used to calculate emissions by comparing the model output to the observations. This can be done in a variety of ways. The stationary dispersion techniques used by Brantley et al. (2014) and Foster-Wittig et al. (2015) utilize the model at a single point and relate changes in concentration to changes in wind direction and thus speed. These procedures either follow, or are related to, the well-defined U.S. EPA OTM 33a and are not discussed further. The mobile dispersion techniques investigated in this study and others (Lan et al., 2015, Rella et al., 2015, Yacovitch et al., 2015) compare observed concentrations at continuous downwind x and y locations (i.e. along a road) to the modeled output along this road. Various techniques have also incorporated averaging schemes and additional z dimensional data (Lan et al., 2015, Rella et al., 2015). Wind direction and speed are either fixed to the prevailing direction or rotated to match the observations. While data can be collected and processed quickly, the application of a Gaussian model that describes average plume behavior to instantaneous data has apparent shortcomings and no standard uncertainty protocol has been established.

**1.2 Previous Work**

Robust uncertainty analyses of Gaussian emission retrievals are not reported in most studies, which instead focus on the novel application of the methods. Yacovitch et al. (2015) reported an asymmetric 95% confidence interval on their emission rates of 0.334q – 3.34q where q is the reported emission rate by using a controlled release as a proxy. Lan et al. (2015) used a Monte Carlo approach based on assumed uncertainty in source height, wind speed and wind direction for an average 95% confidence interval of roughly 0.5q – 1.5q. Rella et al. (2015) also used a controlled release to calculate the variation in their measurement of a constant emission and reported a 95% confidence interval of 0.28q – 3.6q. These methods report uncertainty analyses that apply to a single 'site' (i.e. 1 well pad regardless of how many times it was actually sampled), though it should be noted that Rella et al. (2015) and Yacovitch et al. (2015) used downwind transects while Lan

used stationary time averaged measurements. Notably, Lan et al.'s (2015) Monte Carlo method produced the smallest confidence interval, but accounted only for assumed uncertainty in three parameters and may neglect other factors (distance, stability, emission variability). The controlled release method employed by Yacovitch et al. (2015) and Rella et al. (2015) is useful in that direct observations of measurement variability can be made and potential bias in the measurements can be

determined. However, these methods produce large uncertainty ranges and it is unclear if they can reliably separate large emissions (e.g. emissions >10x the mean) from mean emissions. Additionally, implementing a controlled release is not trivial due to long set-up times of equipment and restricted access to locations suitable for the release. These conditions may make a controlled release experiment prohibitive for many applications with strict time or budget constraints or for those where site access is limited. As a goal of this work is to identify best practices for quantifying uncertainty, it is important to

understand how feasible a given method is likely to be.

The Gaussian model is attractive as a method for inferring emission rates as it is fast and generalizable with the ability to account for changes in stability, wind speed and source elevation. However, the uncertainties for this method change depending upon how it is implemented and whether it is extended to situations outside the reasonable limits of the generalized form. Such situations would include using the average Gaussian plume model with instantaneous measurements

without uncertainty or sensitivity analysis or applications over complex topography. A method for implementation of this technique is needed that identifies best practices and is supported by observations and modeling.

In our study, we combine traditional Gaussian methods, advanced large eddy simulation modeling and a controlled release to assess in-situ variability of emission retrievals from $CH_4$ plumes downwind of natural gas well pads in the Marcellus Shale in Pennsylvania. We also investigate sources of potential bias in the controlled release and modeling

methods. The basic architecture of our approach to quantify errors associated with mobile Gaussian methods uses (1) advanced modeling of a preselected sample site to enable investigation of optimum sampling strategies, (2) application of strategies to the sample collection process and (3) evaluation of additional sources of uncertainty and bias using advanced modeling and a controlled release.

## 2 Methods

### 2.1 Instrumentation

Field data were collected in Pennsylvania during three campaigns in July 2015, November 2015 and June 2016 using the Princeton Atmospheric Chemistry Experiment (PACE). PACE is a Honda CR-V that has been modified to accommodate a roof rack that holds sensors ~1 m above the car, to limit the possibility of self-sampling. The roof rack is equipped with a LI-COR 7700 to measure $CH_4$ and LI-COR 7500A to measure $CO_2$ and $H_2O$; both sensors record at 10 Hz.

Meteorological data and GPS data were collected at 1 Hz with a Vaisala WXT520 and Garmin unit in July 2015 and June 2016 and with an Airmar WS-200WX in November 2015. More information on the mobile lab design and instrumentation can be found in Tao et al. 2015. The LI-COR sensors were calibrated prior to each campaign using a blank ($N_2$) and a 2.12

ppm $CH_4$ standard in air and were also periodically calibrated with a $1.8724 \pm 0.0030$ ppm $CH_4$ and $394.51 \pm 0.07$ ppm $CO_2$ NOAA standard. The high stated measurement range of the LI-COR 7700 (40 ppm) and the excellent stability of the instrument allow for calibration with a relatively low concentration standard (McDermitt et al. 2010). In addition, enhancements observed were usually less than a few ppm above the ambient concentration. Data were synchronized and

logged using a custom LabVIEW program.

In addition, at select sites, a tower was set up to measure high-frequency meteorological data. This tower included a second pair of LI-COR 7500A and 7700 along with a METEK uSonic-3 Class A sonic anemometer to measure the three-dimensional instantaneous wind vector. The tower was typically set alongside the road at a height between 2 and 3 m. Initially, the tower was constructed using a standard tripod, but was later adapted to the bed of a pick-up truck to allow faster

deployment. The air flow around the pick-up truck was modeled (using Fluent, http://www.ansys.com/Products/Fluids/ANSYS-Fluent) to determine optimum placement of sensors above the vehicle to minimize local flow distortions. Three orientations were tested, with the truck cab facing 0°, 90° and 180° with a 0° mean wind flow. Deflection was observed in all three cases, but the distortion was minimal at ~2 m above the pickup bed. The final design of the mobile lab, the instrumentation rack and the mobile tower are shown in Fig. 1.

**2.2 Hierarchical Sampling**

Unconventional natural gas well pads were selected for sampling using a pseudo-random method to efficiently isolate sites that could be measured from public roads with the prevailing wind direction. All datasets were accessed from the Pennsylvania Spatial Data ACCESS (www.pasda.psu.edu). Sites were screened to remove those that were far (>300 m from public road), had obstructions (buildings and full tree lines) or had large elevation differences between the well and the

roadway (>50 m). These characteristics were determined to be the most crucial for successfully sampling sites in this area as topography and vegetation made detecting plumes farther than 300 m difficult as the source could not be visually verified. The direction from the nearest public road was used to separate sites that could be measured with four prevailing wind conditions (N, E, W, S). This resulted in a database of screened sites for each wind direction that could be used to make measurement routes. Routes were primarily planned for efficiency around the forecast mean wind direction; however, the

distribution of the sample relative to the population of key factors (well age, production, and operator) was routinely examined to identify and correct for over- and under- representation. These data were collected at different hierarchical levels based on the length of time collection took, complexity of data collected and analysis method. The data collected at each level are summarized in Table 2. A discussion of the details and rationale of the sampling strategy can be found in Sect. 3.

As a source of validation, an experimental controlled release of $CH_4$ was also performed. The controlled release allows the retrieved emissions to be compared to known emission rates from a constant source. A pure (99.5%) $CH_4$ cylinder was vented at various controlled flow rates to produce different measurable emission rates. The release site was selected for flat, open topography and isolated from any potential sources. Background transects were collected for approximately 30

min before releasing $CH_4$ to ensure no contaminating signals would be detected. An interfering $CH_4$ signal from a large mulch pit was detected and the release set up was moved away from the source to ensure no signal mixing. The cylinder was set ~100 m from a public road at an altitude of 1 m. The release was performed over several hours, during afternoon and evening to span different stability classes. Figure S1 shows a diagram of the release set-up.

## 2.3 Inverse Gaussian Method (IGM)

The IGM approach has been used extensively as described in Sect. 1.1. Applied here, the method uses the sampled source location as input to first identify downwind transects. The peak $CH_4$ location along with the known source location are used to define the prevailing wind direction and centerline plume (x in Eq. 1). The along-wind and across-wind (y in Eq. 1) distances are then calculated using the synchronous GPS data. Distances are calculated for each measurement point as a transect may not necessarily be perpendicular to the wind. The receptor altitude (z in Eq. 1) is fixed at 2.5m, the height of the instrumentation above the road. Unless measured at intensive sites with a tower, wind speed and stability are taken from NOAA's Ready Archived meteorology (https://www.ready.noaa.gov, Rolph et al., 2017) because mobile wind data showed artefacts after corrections for vehicle heading. These artefacts included unreasonably high wind speed and little correlation to stationary tower measurements. The NOAA Ready archive meteorology dataset is from the National Center for Environmental Prediction's Eta Data Assimilation System model (EDAS, information at https://ready.arl.noaa.gov/edas40.php, Black, 1994). These climate analysis data are available in 3-hour increments at 40 km resolution and constant pressure coordinates. The data were interpolated to 1 hour resolution for use in the model. The hourly data are matched to the closest observation based on time. The stability data are used to identify the proper z and y dispersion parameters based on Briggs (1973) for rural areas in the downwind regime between 100 m and 10,000 m. While there is uncertainty in using interpolated model wind speed and stability, especially as conditions can change in the morning and evening, the sampling period per site lasted on timescales of a few minutes. At this temporal scale, wind speed, stability, and turbulence statistics are assumed to be constant. During rapidly changing conditions, the model interpolated wind speed and stability could indeed be incorrect. The effects of uncertainties in wind speed and stability are discussed in Sect. 5.1.

As discussed in Sect. 1.1, the comparison between the observations and modeled output along a downwind transect is used to calculate emission rate. First, the local background ($C_{Background}$), defined as the $CH_4$ minimum over the transect, is subtracted from observations of a plume ($C_{observation}$) to produce an enhancement value. The uncertainty of the background selection in the specific context of this work is discussed in Section 5.1. Second, Eq. 1 is solved for the x and y measurement points of the measured transect using a reference model emission rate ($Q_{ref}$) taken arbitrarily to be 1 kg s$^{-1}$ to produce $C_{model}$. The ratio between the observations and model is used to infer the observed emission rate. Again, the peak observation value is used to define the plume centerline for the Gaussian model for each transect. A comparison of observations and model output from 21 downwind transects is shown in Fig. 2. Note that the roads were not necessarily perpendicular to the wind, therefore the superposition of the plume on the roadway may not show a full Gaussian profile. Third, the observations and modeled concentrations are both integrated along y (summed since they consist of discrete points) as previously

recommended by Albertson et al. (2016) to minimize the influence of the random variability of the instantaneous plume. Finally, because the concentrations scale linearly with the emission rate according to Eq. 1, the emission rate can be estimated as shown in Eq. 2:

$$Q = \frac{\sum C_{Observation} - C_{Background}}{\sum C_{Model}} \times Q_{ref} \tag{2}$$

where $\Sigma$ implies summation over y. This method of integrated concentrations has the advantage of not relying on regressions between the instantaneous data and model data which may have very low correlation as the instantaneous plume is not expected to adhere to a Gaussian profile on such a short time scale, even when a few transects are averaged. It should also be noted that under ideal conditions (e.g. road perpendicular to prevailing wind direction), integrating the Gaussian equation (Eq. 1) in y creates a Gaussian profile independent of the choice of horizontal diffusion ($\sigma_y$). As the vertical diffusion ($\sigma_z$) is

expected to be independent of averaging time (CCPS, 1996), this also has the advantage of minimizing the effect of different measurement timescales when comparing the observations to the Gaussian model which are described in Fritz et al. (2005).

### 2.4 Large-Eddy Simulation

Large-eddy simulation (LES) is used to simulate the dispersion of $CH_4$ for sites that had been sampled with a tower for approximately 1 hour and sampled with the mobile lab with at least 10 transects at both the beginning and end of the

observation period. The LES turbulent modeling technique is the most suitable for high-Reynolds number flow and dispersion in the atmospheric boundary layer. The LES code used in this study has been widely validated (Bou-Zeid, Meneveau, and Parlange, 2005; Tseng, Meneveau, and Parlange, 2006; Li et al., 2016). Briefly, the LES code solves the resolved continuity, Navier-Stokes, and scalar conservation equations on a Cartesian grid, and models the unresolved motions using the Lagrangian scale-dependent dynamic subgrid-scale model (Bou-Zeid, Meneveau, and Parlange, 2005).

The sharp interface (Mittal and Iaccarino, 2005) immersed boundary method is used to simulate flow with the presence of large solid structures (e.g. tanks in this study) in the field (Chester, Meneveau, and Parlange, 2007; Li, Bou-Zeid, and Anderson, 2016; Tseng, Meneveau, and Parlange, 2006). Scalar sources are located on top of the structures or at other points around the source structure if needed to simulate the gas emissions. A more detailed description of how sources were selected is presented in Sect. 4.1.

A pseudo-spectral method is used for horizontal spatial derivatives and a second-order finite difference method is used for vertical spatial derivative with the needed treatments to overcome the Gibbs phenomenon following Li, Bou-Zeid and Anderson (2016). The second-order Adams-Bashforth method is used for time integration. The inflow velocity is a turbulent logarithmic profile generated from a separate simulation over homogeneous flat terrain mimicking upwind conditions. The inflow scalar is kept at a constant background concentration.

In total, 5 sites were simulated in neutral conditions. Most sites were set up with 1 or 2 m horizontal and vertical grid resolution with total simulation domain size of 256 m in x (along-wind) and y (cross-wind) directions and 100 m in z (vertical) direction. Site 5, the controlled release, was set up with 1 m horizontal resolution and 0.2222 m vertical resolution

with a full z dimension of 33.33 m. This was done to because the release source is at a low elevation and such high resolutions are needed by LES to resolve a sufficient fraction of the turbulent scales near the surface. It is important to note that all sites were set up to ensure that the vertical dimension was at least 10× the tallest simulated obstruction, which is equal to the height of the emission sources. This is significant as according to Townsend's theory of attached eddies,

turbulent scales that contribute significantly to vertical diffusion of a plume are proportional to the distance of this plume from the surface (thus 'attached' to the surface, Townsend 1961). Townsend's theory has been confirmed experimentally by many studies (Perry and Li 1990, Nickels et al. 2007, Woodcock and Marusic 2015) for near-neutral conditions. Since the domain size of the LES will limit itself the largest scale of eddies that can be resolved, the large z dimension used for these sites will allow full contributions from eddies of size up to one order large than the emission point elevation, which will

diffuse and spread the plume. The real atmospheric boundary layer might contain even larger eddies (up to ~1 km) that are not captured in the LES; since these eddies are much larger than the source height and thus the plume cross sectional scale, they will cause plume meandering, but will not diffuse the plume in the y and z directions. As a result, LES might underestimate plume meandering, a point that will be revisited in Sect. 4.2.

        Site layouts are shown in Fig. 3. Sites were simulated for at least 30 min to allow the simulated turbulence to reach

a statistically stationary state, where average and standard deviations of wind and scalars approach a constant value as shown in Fig. S2. The equations solved in LES are non-dimensionalized using the friction velocity ($u_*$). The advantage is that results from LES apply more broadly to any problem when the non-dimensional quantities, e.g. as $u_{nondimensional}=u/u_*$, are considered. LES outputs can then be scaled (dimensionalized) with the measured field friction velocity and the scalar flux rate imposed in the simulation to match the reference emission of $Q_{ref} = 1$ kg s$^{-1}$ that allows direct comparison to the

Gaussian model estimates. Table 3 summarizes conditions and domain parameters for all 5 sites. Sites were primarily selected for simple geometry with flat terrain and homogeneous upwind conditions. Generally, elevation differences across the domains were less than 4 m and structures could be easily seen and photographed from the road to aid in site set-up.

## 3 Sampling Strategy

### 3.1 Model-Based Design of Sampling Strategy

25        LES has been previously used to investigate plume dispersion and is used here as a reference that represents the best estimate for the 'truth' of how a plume evolves in a turbulent near-neutral environment (Nieuwstadt and de Valk, 1987, Weil 1990. Wyngaard and Weil, 1991, Mason 1992, Weil et al. 2004). A useful extension of the LES analysis would be to examine the output as a reference case to understand how 'sampling' the model environment by taking instantaneous 'measurements' of the concentration fields affects emission retrievals. The turbulent structures that LES can resolve are

illustrated in Fig. 4, which contrasts instantaneous plumes and averaged plumes in both the horizontal stream-wise (x-y) and vertical cross-wind (y-z) perspective. To optimize sampling there are two important variables (1) the number of

measurements and (2) the time interval between measurements. Increasing the number of measurements is expected to increase the accuracy of the retrieval; however, the time interval may also affect results as measurements with short spacing may resample the same coherent plume and thus the same plume realization (Metzger at al., 2007, Shah and Bou-Zeid, 2014).

Using the LES output, which is saved with 1 Hz resolution to match our instrument sampling frequency, sample transects were picked from the full time series, varying the number of repeat transects and their time intervals. Time intervals of 30s, 1 min, 2 min and random (meaning the time interval was not consistent or constrained) were imposed upon the sample picks and the number of repeat transects ranged from 1 to 70. These 'transects' were then integrated and used as $C_{observation}$ in Eq. 2 (LES is the experiment here) and the average LES profile was used as $C_{Model}$ to produce an emission rate.

For each combination of transect number and time spacing, 100 random samples were picked and the mean and standard deviation of the emission rate were calculated and compared to the known LES emission rate. An ideal scenario would result in a low percent difference and a standard deviation of the sample that is roughly equal to the standard deviation of a fully random sample (random time spacing). The random time spacing should be representative of a fully random sample as points are drawn from the full 30 min time simulation and are less likely to resample similar plume structures. Standard

deviations are being compared instead of standard error as each sample strategy is being treated as a population so that the resulting standard deviation may be used as an approximation of the population standard deviation. Box plot of the results for the 100 random samples are shown in Fig. 5. The effect of increasing the number of measurements clearly reduces the range of retrievals, but the benefits of adding transects becomes incremental around 10 samples beyond which increasing the number of transects reduces the retrieval scatter very slowly (individual box plots are available in Fig. S4). The 5-95% range

of observations for absolute percent difference (pd, always relative to the mean of the compared observations) decreases by 60% at 10 transects, but only decreases by an additional 10% by extending up to 70 transects. The 5-95% range of observation for relative standard deviation (rsd) follows a similar but less extreme pattern with the range of observations decreasing by 20% up to 10 transects and decreasing by an additional 15% by extending up to 70 transects. Additionally, retrievals with 30s spacing show increased bias (seen by higher absolute pd) as even high numbers of samples may measure

plume structures that are similar as indicated by the low scatter, but are not very representative of the whole simulation. However, the 1 min, 2 min and random intervals look very similar indicating a 1 min interval can be used as a practical lower limit (this might somewhat depend on turbulence intensity and stability in the atmosphere however). Notably, the random time spacing sample shows an rsd of ~25% even at the maximum number of repeat measurements (70). This confirms that there is variation in the quantified emission rate expected simply due to atmospheric variability. Atmospheric

variability may have many sources; in these simulations, variability is only attributable to turbulence. However, in a real dynamic environment atmospheric variability could also include effects of mean wind flow change and plume meandering, especially in low wind speed conditions (Vickers et al., 2008, Mortarini et al., 2016). Another way of describing the atmospheric variability as used in this work would be transect to transect variability, which encompasses all random errors that lead to differences between one transect through the plume and the next. These are hence the errors associated with

insufficient averaging of the turbulent field and would be reduced as the number of averaged transect increases (Salesky and Chamecki, 2012). These results indicate that in order to sample such that the measurements minimize the effect of these random error, sites must be sampled with at least 10 transects with >1 min spacing.

### 3.2 Field Implementation

5       Field measurements were designed to target neutral stability found in the morning and evening with each sampling outing typically lasting four hours; this minimizes the errors related to assigning stability and coincides with LES conditions. Most sites were sampled 1-3 times (denoted as standard sampling), occasional sites were sampled with ~10 transects (replicate sampling) and a few sites were sampled with >10 transects as well as a tower (intensive sampling). Typically two replicate sampling sites were picked per outing to capture atmospheric variability for a given condition. As depicted in

Figure 6, the goal of the sampling strategy was to produce more standard sampling sites, with fewer replicate sampling sites and even fewer intensive sampling sites. This was based upon the approximate amount of time to acquire each sample and the limited amount of time to collect samples overall.

      Field campaigns were deployed in the Marcellus shale spanning northeast and southwest Pennsylvania. In total, 940 well pads were sampled with standard sampling, 53 with replicate sampling and 16 sites with intensive sampling. These

replicate sampling sites were generally chosen at the beginning and end of each four hour sampling period to observe changes in variability over the course of the sampling period that may be due to changes in atmospheric conditions. For the population of standard sampling sites with multiple passes, the average rsd of emissions from repeat passes was 67% and the average maximum percent difference (highest observational deviation from the mean of repeat measurements) between emission estimates at a single site was 58%. The average rsd of the population of 53 emissions estimates for the replicate

sampling sites was 77% and the average maximum percent difference was 150%. The rsd of emissions from repeat passes ranged from 12% to 260%. These populations offer insight into how sampling strategy may change estimates of these statistics and offer the chance to compare real results to the LES results shown in Sect. 3.1. These results are consistent with the LES results shown in Sect. 3.1 predicting small numbers of transects will yield an artificially low rsd and more transects are needed to produce an accurate measure of variability. Additionally, the lower maximum percent difference for standard

sampling is consistent with the Sect. 3.1 LES results showing few transects will sample more similar plume structures. While there is a large range in the rsd observed, ~75% had rsd values less than 100%.

### 4 Results of Source Strength Determination

### 4.1 Strategy for Comparing LES and IGM Results

      Comparisons between the IGM calculated emission rates and LES output should be done with care because the LES

cannot be scaled to different distances and wind angles easily. The base scenario for the Gaussian approach at all sites (described in Sect. 2.3) assumes there is only one source at the 1 m elevation well-head location because the well-head is the

only geolocated structure in a public database and is the only structure common at every site. Sites may have varying numbers of well-heads, but they are generally very close together (<10 m) so a centralized point is used for sites with multiple well-heads. However, the Gaussian can in fact be adjusted to include multiple sources and source heights. In order to ensure that the differences between outputs are due to the calculated model diffusion and not differences in model set-up,

we compare three scenarios: (i) the base scenario is the IGM approach used for all sites that assumes there is a single-source Gaussian at the well-head at 1 m (SS Gaussian), (ii) the second scenario assumes the sources are other structures on the domain (i.e. storage tanks and processing equipment) that are taller and uses a multi-source Gaussian (MS Gaussian) model where all sources have the same strength and (iii) the LES that simulates sources at the same locations and heights as the MS Gaussian. A schematic of the emission rate calculation strategy is shown in Fig. 6. Generally, the more information available

(e.g. source location), the less uncertain the results are likely to be. Results will be compared from different scenarios to address to what extent uncertainty can actually be reduced. As on-site access was not available and well pads may contain multiple sources, all large structures were treated as separate point sources. These were visually identified during measurements and exact coordinates were confirmed in Google Earth. The center of all sources was used as a point source. The identified sources were always gas processing units or storage tanks.

To compare to the LES results, the observations were indexed to coordinates on the 256 by 256 m horizontal LES grid. The resulting transects were interpolated within the range of observations to account for grid cells with multiple data points or missing data points. The LES time series spanning ~30 minutes were averaged to produce a pseudo Gaussian distribution excluding a ~5 min initialization period (time until stationary state is achieved). LES statistics (mean and std. dev. of scalar and wind components) were plotted as a function of time to determine the onset of a steady state (Fig. S2).

Because all LES runs are non-dimensionalized, the LES output needs to be scaled to represent the actual field conditions for a given observation. The LES non-dimensional and dimensionalized (denoted with a superscript D) outputs can be related by Eq. 3, which can be inferred from the stationary advection diffusion equation or by analogy to the scaling of the Gaussian model in Eq. 1:

$$\frac{C_{LES}}{\dot{M}_{LES}} \times u_{LES} = \frac{C_{LES}^D}{\dot{M}_{LES}^D} \times u_{LES}^D \tag{3}$$

Rearranging Eq.3, the LES non-dimensional scalar output ($C_{LES}$) was dimensionalized ($C_{LES}^D$) to units of kg m$^{-3}$ according to Eq. 4:

$$C_{LES}^D = \frac{\dot{M}_{LES}^D}{\dot{M}_{LES}} \times \frac{u_{LES}}{u_{LES}^D} \times C_{LES} \tag{4}$$

Here, $\dot{M}_{LES}^D$ is the normalized mass (1 kg s$^{-1}$) and $\dot{M}_{LES}$ is the actual emission rate introduced into the LES system. Normalizing by $\dot{M}_{LES}$ is necessary to match the reference emission rate ($Q_{ref}$) also used in the Gaussian retrievals. The non-

dimensional wind speed ($u_{LES}$) is divided by the dimensionalized ($u_{LES}^D$) wind speed, which is set equal to the on-site tower measurements at the measurement altitude. The LES generated winds thus retain a vertical wind profile. The peaks of the interpolated filed observations were centered to align with the peak location of the LES average plume as shown in Fig. 7; as

previously discussed LES does not replicate the small changes in wind direction that can occur in the real-world that cause meandering (unless they are known and imposed). The LES scaled concentration at 3 m (the mobile lab measurement height) was treated as $C_{Model}$ in Eq. 2 to produce LES derived emission rates. This methodology is used to calculate the LES emissions shown in Sects. 4.3 and 4.4.

## 4.2 Clarification of 'Meandering' and a Conceptual Proof of Plume Centering

As terminology in boundary layer meteorology can be ambiguous and is not always used consistently, the specific meaning of 'meandering' is discussed here in the context of the nearfield regime applicable to this work. Many previous boundary layer works have used 'meandering' to describe the general movement of a plume that does not correspond to the time averaged profile, meaning it includes all scales of motion (see Venkatram and Wyngaard 1988). However, Gaussian models do not simulate large scale meandering or shifting wind directions unless these are included in the diffusion coefficients. Thus we clarify that large scale 'meandering' as used in this work corresponds to the effect of larger (>> plume diameter) scale motions (>~100 m). For instance, Seinfeld and Pandis (1998) acknowledge that plumes diffuse more with increased averaging time, which is in line with the previously discussed Fritz et al. (2005) work that observed there are optimum time scales to match a plume horizontal dispersion profile to the modeled output and observations can be quite different if their timescales are longer or shorter. The larger scale motions would be responsible for creating a broader mean plume, but do not diffuse the instantaneous plume cross-section (since they are larger than the plume's diameter). As discussed earlier, the size of eddies important for vertical diffusion are on the order of the source height; consequently, we also assume that larger scale motions do not affect the vertical profile. Deardorff and Willis (1988) suggested that the downwind plane could be redefined every 20 min to remove larger mesoscale and synoptic scale motions.

As a conceptual exploration of the effects of the centering of the Gaussian on individual plumes (as described in Sect. 2.3), a simulated meandering plume with a 100 m period and 10° wind shift was investigated. The simulated plume and a comparison of a single Gaussian plume are shown in Fig. 8 (panels a and b). Using a 100 m downwind transect, the reconstructed plume, created by averaging the re-centered individual components, is compared to the Gaussian with an average wind direction and the average of all the meandering profiles. As shown in Fig. 8 (panels c and d), the exact horizontal plume structure is duplicated simply by re-centering and averaging the individual plume realizations and the vertical plume structure is always preserved whether additional meandering is included or not. When calculating emission rates using the IGM approach in this work, the instantaneous Gaussian plumes centered to the observations are used, instead of using a single average Gaussian profile. However, while the individually calculated emission rates can vary when using the aligned or average of the aligned Gaussians, the differences are expected to be small.

To investigate whether the difference between the approaches would lead to a significant bias, the results for the controlled release experiments were compared for three approaches, the Gaussian centered on each observation that we have employed, the average of the individually simulated Gaussian profiles for multiple observations, and a single Gaussian that matches the apparent average wind direction (Fig. S3). The results show that all three scenarios are virtually identical.

Additionally, the centering of the Gaussian on the plume is expected to be more suitable as suggested by the standard deviations that are 0-20% lower than results that rely on a single average Gaussian. This is likely due to the occurrence of wind conditions that are not symmetrical about the centerline and road geometries that allow for plumes of closer and farther distances. This 'centering' can be thought of as a correction for the apparent conditions not matching ideal conditions in which a single Gaussian in the mean wind direction is theoretically appropriate. Yacovitch et al. (2015) also employed Gaussian centering to observations on the assumption that the source location was better constrained than the wind direction. Foster-Wittig et al. (2015) explored a similar concept of applying corrections to their methodology, which is based on the EPA OTM33A protocol, to account for conditions that did not match the assumptions of the EPA protocol. They also observed that the differences between their methods were small. Overall, we expect uncertainty from the methodology we have employed to be constrained by the analysis present here, including using Gaussian and LES outputs that are not varied based on the observed peak location. It should be noted that this centering may not always be necessary or optimal based on the specific goal of a campaign that might include source localization or atmospheric dispersion quantification.

When centering the observations to the LES (which cannot be modulated to match apparent changing wind direction in the way the Gaussian can) the sum of the horizontal (y) distribution is conserved whether the observations are centered or not. This means that the emission rate does not change whether centering or not centering the observations. The only potential for deviation is caused by moving part of the observations outside the imposed 256 m window during centering. The plume centering that we have chosen to apply to the observations is, in this case, expected to correct for the large scale plume meandering not simulated in the LES and will not affect the calculated results; it is a way to filter out the meandering impact of the large scales in the field observations. Filtering observations for the purpose of matching the domain on the LES is also discussed by Agee and Gluhovsky (1999ab) and Horst et al. (2004). Agee and Gluhovsky (1999ab) specifically address the need to remove the influence of larger scales from observations to appropriately compare observations to the LES results. Finally, it is important to highlight that the integrated plume is used to calculate emissions, which mitigates the potential for concentration mismatch when comparing instantaneous plumes with structure to averages LES or Gaussian output due to averaging timescale differences. As noted in Albertson et al. (2016), the use of an integrated plume removes the influence of the random nature of the instantaneous plume, making this a beneficial procedure for these kinds of measurements.

### 4.3 Controlled Release

The controlled release experiment (Site 5) utilized 3 leak rates: $0.97 \pm 0.01$ kg hr$^{-1}$, $0.216 \pm 0.002$ kg hr$^{-1}$ and $0.090 \pm 0.002$ kg hr$^{-1}$. Release rates were controlled by two MKS mass flow controllers with stated flow accuracy of 1% of the set point (Model GE250A) or 0.05 SLM (Model 1179C). Site set-up was discussed in Sect. 2.2 and specific site details are available in Table 3. Due to the low height of the release, the LES domain was modified to have vertical resolution of 0.2222 m with a total domain height of 33.33 m. Boxplots of the populations of emission retrievals from three scenarios (SS Gaussian with NOAA winds, with measured winds and LES) are shown in Fig. 9 and statistics are summarized in Table 4.

Figure 9a shows the emission rates using NOAA winds while Fig. 9b shows the emission rates calculated with measured wind. The two Gaussian retrievals are shown to compare using NOAA winds, which is the base scenario for all sites, and using the in-situ measured wind. Since the controlled release used one release point, there is only one source at a known height and the SS Gaussian approach is used. The agreement increases greatly when using the in-situ wind data; NOAA overestimated the winds during the latter two release rates.

The Gaussian approach with in-situ measured wind agrees quite well with the release, surprisingly better than the LES. This may be due to effects of stability the Gaussian can account for, but were not simulated in the LES where neutral conditions were assumed; this will be further discussed in Sect. 5.1. In this case, the conditions shifted from slightly unstable to neutral during the second release, with the friction velocity (an important scaling parameter for LES) decreasing from 0.21 m s$^{-1}$ to 0.10 m s$^{-1}$. The slightly better performance of the Gaussian method in the experiment suggests that the correction factors for stability are at least as important an input as the direct calculation of diffusion in LES across a dynamic environment. While the LES investigation is useful to assess sources of error induced by sampling strategy, a controlled release is the most direct way to detect sources of bias, which otherwise would not be apparent. In general, the close agreement across a range of release rates shows no apparent bias in the Gaussian with measured winds results, with the results scattered low and high relative to the release rate and no results significantly different (at the 95% CI) from the release rate. The Gaussian with NOAA winds showed no bias when the winds matched with the observations as in the first sample, but may have biases if the winds aren't well represented. The wind speeds measured (and NOAA winds) for releases 1-3 were 3.0 m/s (4.1 m/s), 1.91 m/s (4.5 m/s), and 1.5 m/s (5 m/s), respectively. While LES can readily account for unstable or stable conditions, this would come at an increased computational cost as multiple simulations would be needed. The results here show that this in fact may not be necessary, at least over flat homogeneous terrain, as the Gaussian model provides a comparable performance at a very small fraction of the modeling effort.

The controlled release was also used as an observational constraint to investigate the sampling strategy identified by the LES. This was done by randomly selecting an increasing number of transects from each release and comparing the inferred mean release rate using the IGM. The results in Fig. 10 are in excellent agreement with the LES results pattern seen in Fig. 5 where the average converges beyond 10 transects. Note that the boxplots correspond to the distribution of means obtained from randomly selecting the indicated number of transects, not total uncertainties on the means for each number of transects. This reiterates the importance of the sampling protocol and also shows the range of results possible if only a limited amount of transects are used. Additionally, Fig. 10 shows that the IGM rarely overestimates (>2x average release rate), but more often underestimates (<0.5x average release rate). By this definition the IGM results underestimated the release rate by up to 30% of the time during the release rate using only 1-5 transects. During release 1 and 3, even though a constant source is being emitted, a few (1-3) transects showed no observed plume whatsoever.

## 4.4 Intensive Field Sites

Comparison between mean emission rates calculated from all available transects for sites 1-4 are shown in Table 5 and site specific details are shown in Table 3. Relatively good agreement is found between the LES emission retrieval and the SS and MS Gaussian approach for Sites 1-4. This indicated there is consistency in retrieved emissions from a range of scenarios despite using different amounts of information to set up the sources. The range of emission retrievals vary within a factor of two in most cases. For sites 1-4, the average percent difference (difference from average value) between the LES and IGM emissions (SS or MS) was ~25%. Due to the effort to standardize the comparison between all approaches by centering and correcting the observations, the difference in emission is entirely due to the difference in the dispersion each model produces. The Gaussian models assume stability corrected diffusion coefficients from Briggs (1973) while the LES makes no assumptions and allows turbulence to be numerically solved. The LES, however, was only run under neutral stability in this study. As shown in Fig. 11, the horizontal dispersion generally matches well between the LES and the MS Gaussian, while the vertical dispersion exhibits slightly different behavior. In the sites studied the LES predicts peak vertical flux (obtained by multiplying mean concentration × mean wind speed) at lower altitudes closer to the source and at higher altitudes farther from the sources with the equivalence point around 100 m downwind (additional figures of Gaussian and LES dispersion are shown is Figures S5-S7). The differences are due to the distance scaling in the Briggs (1973) model being different from the LES. While there are other analytical models for distance specific dispersion coefficients, the Briggs 1973 actually matches the LES profiles better across the range of sites explored here than several other common models (Gifford, 1976, Smith, 1968; see Figures S8-S11 and Appendix A). Vertical dispersion is generally more important than the horizontal dispersion as integrating across a transect will effectively nullify any differences resulting from horizontal diffusion. However, if a difference in vertical dispersion exists, this can significantly change the retrieved emission rate. There is considerable difference between the SS and MS results, for instance. Without observations at multiple heights, it is impossible to verify which assumption is correct. However, the SS results are typically closer to the LES results in these simulations, suggesting that this is the better approach compared to assuming source height and locations. In the range of distances investigated in this study (<200 m) the overall discrepancy between the different model outputs is, however, small.

## 5 Uncertainty Analysis and Discussion

### 5.1 Other Uncertainty Sources

For this analysis, we have assumed the source to be constant during the time span of the measurements (typically less than 1 hour). This may not be true for all sites and may be a driver of variability, for instance tanks are known to emit sporadically and Goetz et al. (2015) have shown emissions varying over the course of a few hours. However, it is not clear that there is a need to quantify emission variability at scales less than 1 hour for most sources as there is a practical limit to

the time resolution that can be included in inventory estimates, for instance. We thus expect any changes in emission rate at <1 hour to be reflected in what we have termed atmospheric variability, or transect to transect variability.

Other sources of uncertainty considered are source location and source height. While well pads can be a few thousand m$^2$ in area, infrastructure that could generate leaks is usually clustered such that observed potential sources span a range of 50 m. This can be investigated theoretically by comparing the expected model sum as a function of distance assuming a 50 m shift in source location as shown in Fig. S12. This scenario assumes typical conditions observed in this dataset (3 m receptor height, 1.5 m s$^{-1}$ wind and neutral stability). Generally, changing the along-wind location of the sources changes the emission retrieval by less than 35% when measuring at >100 m downwind. However, at closer distances where the uncertainty in source location is on the order of the downwind distance this could be a major source of error. For reference, the median distance between observation and sources in this dataset is about 200 m with no sites closer than 30 m and only 5 sites less than 50 m. At 200 m uncertainty is expected to be ~20%. We also investigated the sensitivity of source height, which we estimate ranges from 1 m for wellheads to 8 m for some large storage tanks, as shown in Fig. S13. The results indicate that source height variation changes the emission retrieval by less than 15% at downwind distances ~200 m, again due to the large distance from the source. Uncertainty of the source location in the cross-wind direction is not expected to contribute to significant change in emissions.

Additionally, as shown in Sect. 4.1 (and Fig. 9/Table 4), inaccurate wind data can be a potential source of error. Because the modeled CH$_4$ concentration scales with wind in both the Gaussian and LES models, uncertainty in this parameter is necessary to constrain. In the context of this analysis, we compared the NOAA wind to the mean on-site tower measurements of winds at 16 tower sites. NOAA wind speeds reported higher and lower values than the mean measured winds; the absolute difference was 50% on average. Given the linear relationship between the inverse of wind speed and Q in the Gaussian equation (Eq. 1), relative uncertainty in the wind speed should produce the same magnitude relative uncertainty in the emission rate.

Another important consideration is the assignment of the background value to calculate plume enhancement. For this work, the background was calculated as the minimum value from the plume transect because the averaged 1 Hz data generally showed a uniform background near the plumes. This criteria was also compared to a scenario where the background was calculated from the average of the lowest 2% of observations in a transect with very similar results again indicating that the background value is very stable. When comparing samples with repeat transects, the backgrounds identified for each transect had a median standard error of 5 ppb. When the 5 ppb tolerance is applied to the same data set, the median change in calculated flux was 4.4%. This means that the background is expected to contribute an additional ~5% uncertainty. It should be noted, however, that for very low signals the background can become a major source of uncertainty. The average peak enhancement was ~1250 ppb with a median value of ~260 ppb. Sites with enhancements less than 50 ppb were identified as having emissions that were indistinguishable from zero, when considering all other sources of error. These data were designated as non-emitting sites in the emission database.

The final additional source of uncertainty investigated pertains to stability. The stability class determines the analytical equation used to derive the diffusion coefficients, thus affecting the emission rate. By again comparing a theoretical case, the effect of changing the stability class by 1 can be seen in Fig. S14. The tolerance of 1 stability class reflects the fact that the atmosphere does not usually change multiple stability classes rapidly at a scale of 1-3 hours (i.e. a change from class A to class E would not be feasible). While there are certainly cases where the atmosphere can change rapidly in this time period (i.e. from class B to D), generally a miscategorization class difference of at most 1 is expected. For instance, neutral stability was targeted (Class D) for the ~1000 sites measured as part of this work. Class D was the most frequent stability class observed, with 90% of the data occurring between +/- 1 stability class. Making the stability class less stable will decrease the modeled concentration and consequently increase the emission retrieval while making the stability class more stable will have the opposite effect. The magnitude of the difference between +/-1 stability class is relatively consistent at farther downwind distance, averaging a change of 40% at 200 m downwind.

Not investigated here, but potentially very important to uncertainty, is the effect of terrain including both non-uniform slopes and structures such as trees. In this analysis, we have intentionally sampled sites that were determined to be relatively flat and open. All of the sites modelled in this study follow this criteria, even though not every site in our sample is as simple. The geometric mean of the absolute terrain slope, defined as the absolute value of terrain rise over the distance between sources and observation, for all of our ~1000 sampled sites was 3% and ~60% of the sites sampled had an absolute terrain slope of less than 5%. Nevertheless, some sites did contain more complex topography that could cause drastically different dispersion parameters. Such sites would need to be analyzed on a case by case basis as dispersion over complex topography is usually not generalizable because every site is unique (e.g. the inverse Gaussian modeling approach might work very well at one site but poorly at another). This analysis would be non-trivial and requires high resolution topography data, surface heat flux fields and many other inputs for accurate modelling. Another possible pathway to fully investigate the effects of terrain is to investigate correlation between site emissions determined using Gaussian models and terrain slope. From these analyses one can determine screening criteria to preserve data quality and examine the skill of Gaussian models over complex terrain in general. This is the subject of forthcoming work using the larger dataset and not discussed here. A final note on this topic is that the present LES represented the tanks and structures of the sites as bluff bodies that blocked the flow and created wakes, while the Gaussian models did not. This did not result in drastic difference between the Gaussian and LES results indicating that perhaps small obstructions do not have a disproportionate impact on the retrieved emission rates.

## 5.2 Total Uncertainty Estimate

The sources for uncertainty and bias in the Gaussian measurements discussed in this analysis are summarized in Table 6. These include the uncertainty in the Gaussian diffusion constant by comparing to LES calculated diffusion (Sect. 4), uncertainty due to source location and height and uncertainty due to wind speed and stability class (Sect. 5.1). In addition, the LES was used to observe bias in the Gaussian derived concentration distributions (Sect. 4.4) and the controlled release

was used to evaluate bias in both the Gaussian and LES results (Sect. 4.3). Finally, the LES was used to determine the optimum sampling pattern to constrain actual atmospheric variability (Sects. 3.1 and 3.2). The largest contributor to total uncertainty is atmospheric variability, the random error induced by insufficient averaging of the turbulent instantaneous plume. As atmospheric variability is impossible to separate from other sources of uncertainty, such as wind speed, it is not

surprising that it is the largest source of uncertainty. As described in Sect. 3.1, the LES derived atmospheric variability (defined as the standard deviation of emissions retrieved) is expected to be ~25%, considerably less than the standard deviation observed directly since the LES can capture effect of turbulence, but not the effect of changes in the mean flow and meandering plumes that can contribute significantly to overall atmospheric variability (Vickers, Mahrt and Belusic, 2008, Mortarini et al., 2016). LES has a limited ability to represent the very-large scale motions (Kunkel and Marusic, 2006) or

some eddy features (Glendending, 1996) due to its limited horizontal domain size and idealized forcing (e.g. de Roode et al, 2004, Agee and Gluhovsky, 1999ab). In addition, the limitation of LES in the surface layer due to applications of the Monin-Obukhov similarity theory is also one of the concerns (e.g. Khanna and Brasseur, 1997). We do not intend to represent all sources of uncertainties pertaining to the atmospheric conditions. Instead, the focus of the LES is to resolve the range of scales that are critical for the turbulent diffusion of the plume, which is often represented as a single eddy diffusivity

coefficient in the Gaussian plume models. Thus, the LES predicted variability is the practical lower limit of uncertainty for this method since other sources of uncertainty could be mitigated by better on-site measurements and source location detection, but some random atmospheric variability is more difficult to constrain. While the higher observed atmospheric variability may be partially explained by far more complex real world conditions leading to higher standard deviations, this also emphasizes the possibility that other sources of uncertainty are contained in this realization of atmospheric variability.

For instance, most sites, while relatively flat, still have some inhomogeneous terrain that can influence and deflect wind or lead to increased turbulence. To combine the remaining sources of error, a Monte Carlo simulation of errors through the Gaussian equation was performed. Inputs are available in Table 7 and an example output if Fig. S15. This approach was determined to be the most appropriate as uncertainties may not be normally distributed and emissions are constrained to be above zero in this scenario, causing a skew to the emission retrievals. A generic scenario matching average conditions

experienced during the measurements was devised using a downwind distance of 200m, 1.5 m s$^{-1}$ wind speed, neutral stability and 260 ppb observation enhancement. The specifics with regards to distributions assumed for each uncertainty parameter are shown in Table 7 for the SS Gaussian and MS Gaussian approaches with 1 and 10 transects, as well as a theoretical lower limit scenario. For each scenario, 1000 randomly generated samples of $C_{Observation}$, $C_{Background}$, and $C_{Model}$ were obtained and used according to Eq. 2 to obtain a distribution of Q samples. The Q samples were then used to estimate

the 95% confidence interval. The combined effects produce a skewed distribution of emission rates as shown in Fig. S15. Using this methodology we obtain an uncertainty range for single transect emissions of 0.05q–6.5q where q is the emission rate; for sites that had multiple passes and wind measurements (replicate/intensive) the range is 0.10q –3.0q. As discussed in Sect. 5.1, this analysis focuses on relatively flat, simple sites and is not intended to be generalized to extremely complex

sites. We caution that the plume diffusion uncertainty is therefore a minimum expected value and could be a major source of uncertainty for very complex sites not investigated here.

## 5.3 Advantages and Disadvantages

Of the approaches compared in this analysis, the LES results require far more computational and processing time. Though inputs can be estimated from other sources (i.e. NOAA), we chose to measure meteorological variables directly, which contributed to significantly longer measurement time. While this should be considered best practice when producing computationally expensive LES outputs, there are no inherent differences between inputs needed for LES and Gaussian approaches and the Gaussian approach also benefits from on-site measurements making in-field measurement time theoretically similar. In practice, the additional set-up time for meteorological instrumentation can increase total measurement time for sites intended for LES to >1 hour. Table 8 summarizes the main advantages and disadvantages for each technique. The main advantage of the LES is the ability to directly calculate the plume diffusion rather than rely on simplified models. However, the LES simulations shown here have all been initialized for neutral conditions, which is the stability class we targeted during sampling. This is a relatively transient atmospheric phase typically only occurring in the morning and evening around sunrise and sunset and generally other stability classes are encountered. The Gaussian method enables easy corrections for different stabilities allowing quick processing of data collected in these regimes. While LES can be programmed with different stabilities, the additional computational cost is great and the surface energy budget must be known, which introduces another source of uncertainty as actual heat fluxes may vary over the domain of interest and single point measurements may not be accurate. For these reasons, LES alone would not be the recommended method of calculating emission rates based on our study. Likewise, the single transect method, though fast, has many sources of uncertainty. Hence, the strategic combination of all of these approaches described in Sect. 3 is expected to maximize sampling efficiency while minimizing uncertainty. Less frequent higher intensity measurements (from on-site meteorological data and multiple transects) can be used to provide a better estimate of uncertainty for single transect approaches.

## 6 Comparison to Previous Uncertainty Estimates

Overall, we find LES to be a useful tool to examine Gaussian sampling strategy and sources of uncertainty for mobile laboratory measurements. Subsampling the LES output generates an optimum sampling pattern of at least 10 transects per site to obtain reliable statistics of measurement uncertainty due to atmospheric variability, which is the largest source of uncertainty. When sampling at distances greater than 150 m downwind of sites, the uncertainty due to source location and height are generally less than 20% (for cases where source location is known within 50 m and source height is known within 10 m). Using the LES and a controlled release, we confirm that the Gaussian model performs well when in-situ winds are available. The NOAA estimated winds can be a source of error, but we did not observe a systematic difference between the NOAA and in-situ winds, thus no sources of bias using our approach are expected on average. We note that this

result is valid for this general area, and other locations, which have different challenges and data density may differ. Area specific analysis, or on-site winds, should always be used to reduce bias. LES is therefore not required for studies where source strength calculation is the main goal and other complicating factors such as complex topography are not present. The emission retrievals generally fall within a range of two. From this we use Monte Carlo analysis to extrapolate that the 95%

confidence interval for sites with standard sampling (n=2) ranges from 0.05q–6.5q where q is the emission rate. Using the same approach, sites that had multiple passes and wind measurements (replicate/intensive) can be further constrained to 0.10q –3.0q. This uncertainty estimate is higher than Lan et al. (2015) who reported 0.5q–1.5q at the 95% confidence interval. Their study is identical to the theoretical lower limit of uncertainty we calculate by assuming only the LES predicted atmospheric variability of 25%. It should be noted that Lan et al. (2015) did incorporate some averaging over a

time frame of >10 min to their measurements, which, as discussed in Section 5.3, can also decrease uncertainty. However, this would not mitigate all other sources of error previously discussed.

       In addition, our standard sampling uncertainty range is greater than other Gaussian approaches with controlled releases which reported 0.28q – 3.6q and 0.334q – 3.34q (Rella et al., 2015, Yacovitch et al., 2015); the uncertainty range of the multi-transect sites studies here was similar in magnitude (0.05q–6.5q). However, their analysis did not account for

additional sources of uncertainty (source location, stability, wind speed), which can result in uncertainties larger than the reported values. In addition, Rella et al. (2015) incorporated vertical information to inform their Gaussian plume model creating a mass balance Gaussian hybrid, though only a small vertical profile was available (4 measurement points up to 4 m). As the vertical flux was seen to be a potential source of error in this work, measurements of this metric may reduce uncertainty. As described in Sect. 3.2, the observed atmospheric variability (observed from transect to transect emission rate

variability) can range from 10-200%. Atmospheric variability (random error) was the single largest driver of uncertainty in the Monte Carlo simulations (see for example Fig. S15) because the model can be improved with better information and wind measurement. To reduce the uncertainty from atmospheric variability more observations are needed; however, sometimes this is impractical. In-situ observations of variability from repeat measurements at a single site may potentially be used as a post screening method to exclude conditions that will lead to extremely high uncertainty in single transect sites.

Other factors such as wind speed and stability can have a strong effect and can be quantified to reduce uncertainty.

**7 Recommendations**

       While the uncertainty derived for mobile Gaussian techniques is large compared to many other techniques discussed in Sect. 1, it is low enough to reliably separate 'normal emissions' from 'extreme emissions' that are orders of magnitude larger. Many emission sources exhibit lognormal distributions where this condition is met, making mobile sampling a

reliable way to identify extreme emission sites. However, longer sampling time, reliable mobile wind sampling, and visualization of plume distributions are needed to feasibly constrain this method to under 50% for routine measurements. In

summary, to facilitate more constrained uncertainty from other mobile platform based Gaussian emission estimates, we recommend the following:

1. Sites should be isolated to reduce contamination from other sources and be accessible from thoroughfares at least 100 m away.
2. On-site wind measurement should be collected whenever possible.
3. Additional data should be collected such as photographs (used here) or IR imagery to precisely locate the sources whenever possible.
4. Ideally, all sites should use $\geq$ 10 sampling transects to reliably constrain atmospheric variability.
5. For experiments where sampling frequency is at a premium, at least one site per sampling outing should be repeated with $\geq$ 10 sampling transects to reliably constrain atmospheric variability which is expected to be the largest source of the uncertainty estimate.
6. Uncertainty analysis should be a systematic part of Gaussian sampling design.
7. In the absence of other experiments to study measurement uncertainty (controlled releases), the repeat measurements may be a suitable approximation for the *minimum* expected uncertainty.

While the strategies described in the study (see Fig. 6) were developed for well pads, the recommendation may be applied to other 'point-like' sources with simple terrain, appropriate scaling, and no means of site access for other, potentially more accurate, methods. Examples may include (if far enough downwind to consider the source a point) compressor stations, feed lots, waste water treatment plants, landfills, natural gas pipeline leaks, industrial facilities, and geologic seeps/vents. While the scales involved may differ significantly from a well pad, the necessity for constraining atmospheric variability is critical toward understanding the uncertainties of inverse Gaussian derived emissions.

## 8 Data Availability

A data file containing all the emissions, LOD, uncertainty estimate, meteorology, site locations and traits including spud date, operator, production and status will be submitted to DataSpace at Princeton University (https://dataspace.princeton.edu/jspui/). This archive is free and open to the public.

 ## 9 Appendix A

Diffusion Equations (m) as presented in Zannetti (1990).

### Briggs 1973 Rural

| Stability | Horizontal ($\sigma_y$) | Vertical ($\sigma_z$) |
| --- | --- | --- |
| A | $0.22x(1 + 0.0001x)^{-0.5}$ | $0.20x$ |
| B | $0.16x(1 + 0.0001x)^{-0.5}$ | $0.12x$ |
| C | $0.11x(1 + 0.0001x)^{-0.5}$ | $0.08x(1 + 0.0002x)^{-0.5}$ |
| D | $0.08x(1 + 0.0001x)^{-0.5}$ | $0.06x(1 + 0.0015x)^{-0.5}$ |
| E | $0.06x(1 + 0.0001x)^{-0.5}$ | $0.03x(1 + 0.0003x)^{-0.5}$ |
| F | $0.04x(1 + 0.0001x)^{-0.5}$ | $0.016x(1 + 0.0003x)^{-0.5}$ |

### Briggs 1973 Urban

| Stability | Horizontal ($\sigma_y$) | Vertical ($\sigma_z$) |
| --- | --- | --- |
| A | $0.32x(1 + 0.0004x)^{-0.5}$ | $0.24x(1 + 0.001x)^{0.5}$ |
| B | $0.32x(1 + 0.0004x)^{-0.5}$ | $0.24x(1 + 0.001x)^{0.5}$ |
| C | $0.22x(1 + 0.0004x)^{-0.5}$ | $0.20x$ |
| D | $0.16x(1 + 0.0004x)^{-0.5}$ | $0.14x(1 + 0.0003x)^{-0.5}$ |
| E | $0.11x(1 + 0.0004x)^{-0.5}$ | $0.08x(1 + 0.00015x)^{-0.5}$ |
| F | $0.11x(1 + 0.0004x)^{-0.5}$ | $0.08x(1 + 0.00015x)^{-0.5}$ |

**Gifford 1961**

$$\frac{k_1 x}{[1 + \left(\frac{x}{k_2}\right)]^{k_3}} \qquad \frac{k_4 x}{[1 + \left(\frac{x}{k_2}\right)]^{k_5}}$$

| Stability | Horizontal ($\sigma_y$) | Vertical ($\sigma_z$) |
| --- | --- | --- |

|     | $k_1$ | $k_3$ | $k_2$ | $k_4$ | $k_5$ |
| --- | --- | --- | --- | --- | --- |
| A | 0.25 | 0.189 | 927 | 0.102 | -1.918 |
| B | 0.202 | 0.162 | 370 | 0.0962 | -0.101 |
| C | 0.134 | 0.134 | 283 | 0.0722 | 0.102 |
| D | 0.0787 | 0.135 | 707 | 0.0475 | 0.465 |
| E | 0.0566 | 0.137 | 1070 | 0.0335 | 0.624 |
| F | 0.0370 | 0.134 | 1170 | 0.022 | 0.7 |

$$\sigma = ax^b$$

**Smith 1968**

| Stability | Horizontal ($\sigma_y$) | | Vertical ($\sigma_z$) | |
| --- | --- | --- | --- | --- |
|  | $a$ | $b$ | $a$ | $b$ |
| A | 0.4 | 0.91 | 0.41 | 0.91 |
| B | 0.36 | 0.86 | 0.33 | 0.86 |
| C | 0.36 | 0.86 | 0.33 | 0.86 |
| D | 0.32 | 0.78 | 0.22 | 0.78 |
| E | 0.32 | 0.78 | 0.22 | 0.78 |
| F | 0.31 | 0.71 | 0.06 | 0.71 |

## 10 Author Contributions

D.R.C, Q.L. and E.B.-Z. implemented LES modeling for sites. D.R.C, J.M.L and H.M.L. analyzed raw data and provided Gaussian fluxes. D.R.C., J.M.L, H.M.L, J.P.F., B.B., L.M.G., X.G., J.M., D.P., L.W., and M.A.Z. collected in-situ data. D.R.C, L.M.G., J.F.P., E.B.-Z. and M.A.Z. assisted in sampling design. L.M.G, D.P., B.B. and X.G. provided technical support for data collection and instrument integration. J.M. provided mechanical assistance in mobile lab design and deployment. M.A.Z., E. B.-Z., and J.P.F. came up with the overall project design. D.R.C prepared the manuscript with feedback from all authors.

## 11 Competing Interests

The authors declare that they have no conflict of interest.

## 12 Acknowledgements

We would like to thank all members of the fieldwork team including Stephany Paredes-Mesa, Tanvir Mangat and Kira Olander. We would also like to thank Maider Llaguno Munitxa for her help with the mobile tower modeling. We thank LI-COR Biosciences for lending instrumentation used in this campaign. This work was funded by NOAA CPO/AC4, #NA14OAR4310134.

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

**Tables**

Table 1. Comparison of $CH_4$ emission measurement techniques and reported uncertainties.

| Technique | Referenced Study | Reference Uncertainty Range* |
|---|---|---|
| Ground-Based Thermal Imaging | Gålfalk et al., 2016 | 3-15% |
| Aircraft Remote Sensing | Kuai et al., 2016, Frankenberg et al., 2016, Thorpe et al., 2016 | 5-20% |
| Satellite Remote Sensing | Kort et al., 2014 | 15% |
| Chamber Sampling | Allen et al., 2013 and 2014, Kang et al., 2014 | 20-30% |
| Ground-Based Tracer Correlation | Lamb et al., 2015, Roscioli et al., 2015, Subramanian et al., 2015, Zimmerle et al., 2015, Omara et al., 2016 | 20-50% |
| Aircraft/UAV Mass Balance | Karion et al., 2013 and 2015, Peischl et al., 2013, 2015 and 2016, Caulton et al., 2014, Pétron et al., 2014, Lavoie et al., 2015, Nathan et al., 2015 | 20-75% |
| Ground-Based Stationary Dispersion | Brantley et al., 2014, Foster-Wittig et al., 2015 | 25-60% |
| Tall Tower Monitoring | Pétron et al., 2012 | 50-100% |
| Ground-Based Mobile Dispersion | Lan et al., 2015, Rella et al., 2015, Yacovitch et al., 2015 | 50-350% |

* Uncertainty range reflects author reported uncertainty on *emission* numbers, not necessarily measurement uncertainty. Some authors specify a 95% confidence interval, others use 1 or 2 standard deviations and others compute upper and lower bounds.

Table 2. Summary of acquired data and data level

| Sample Type | No. of Samples | Average No. of Transects | Measurement Time (min) | Wind Source | Emission Calculation Technique | Level |
|---|---|---|---|---|---|---|
| Standard Sample | 940 | 2 | 1-5 | NOAA | IGM | 1 |
| Replicate Sample | 53 | 10 | 5-15 | NOAA, On-site | IGM | 2 |
| Intensive Sample | 17 | 20 | 15-60+ | On-site | IGM, LES | 3 |

Table 3. Summary of LES site conditions and parameters.

| Characteristic | Site 1 | Site 2 | Site 3 | Site 4 | Site 5* |
|---|---|---|---|---|---|
| Sample Date | 7/15/2015 | 7/21/2015 | 6/21/2016 | 6/23/2016 | 8/5/2016 |
| Time (EDT) | 18:15-20:15 | 7:00-9:15 | 9:45-10:30 | 20:00-20:30 | 16:15-20:00 |
| No. of Transects | 26 | 28 | 21 | 22 | 81 |
| Wind Speed (m s$^{-1}$) ($\pm 1\sigma$) | $1.30 \pm 0.64$ | $1.30 \pm 0.70$ | $1.9 \pm 2.2$ | $0.62 \pm 0.27$ | $1.70 \pm 0.89$ |
| Wind Direction ($\pm 1\sigma$) | $325 \pm 101$ | $252 \pm 40$ | $276 \pm 50$ | $40 \pm 47$ | $218 \pm 28$ |
| Friction Velocity (m s$^{-1}$) | 0.26 | 0.29 | 0.48 | 0.18 | 0.10-0.21 |
| Distance from Source (m) | 176 | 154 | 29 | 49 | 110 |
| **Number of Sources** | 2 | 2 | 2 | 2 | 1 |
| **Height of Sources** (m) | 5, 8 | 5, 8 | 5, 8 | 5, 8 | 1 |
| Elevation gain (m) | 3.5 | 1.5 | 2 | 2 | 2 |
| Simulation Stability | D | D | D | D | D |
| Simulation Time (min) | 59.2 | 33.33 | 33.33 | 33.33 | 30 |
| Simulation Warm-up Time (min) | 10 | 5 | 5 | 5 | 5 |
| Simulation x-dimension (m) (x resolution) | 288 (2) | 256 (1) | 256 (1) | 256 (1) | 256 (1) |
| Simulation y-dimension (m) (y resolution) | 256 (2) | 256 (1) | 256 (1) | 256 (1) | 256 (1) |
| Simulation z-dimension (m) (z resolution) | 100 (1) | 100 (1) | 100 (1) | 100 (1) | 33.33 (0.2222) |

* Controlled release site.

Table 4. Summary of mean emissions from three scenarios for the controlled release (± 1 std. dev. of mean).

| Scenario | Release 1 (kg hr$^{-1}$) | Release 2 (kg hr$^{-1}$) | Release 3 (kg hr$^{-1}$) |
|---|---|---|---|
| Release Rate | 0.97 ± 0.01 | 0.216 ± 0.002 | 0.090 ± 0.002 |
| No. of Transects | 19 | 10 | 13 |
| Gaussian w/ NOAA winds | 0.97 ± 0.17 | 0.79 ± 0.13 | 0.35 ± 0.04 |
| Gaussian w/ measured winds | 0.72 ± 0.12 | 0.23 ± 0.04 | 0.10 ± 0.01 |
| Large Eddy Simulation | 0.94 ± 0.17 | 0.44 ± 0.06 | 0.22 ± 0.03 |

Table 5. Comparison of mean emission rates using three scenarios from four sites (± 1 std. dev. of mean).

| Scenario | Site 1 (kg hr$^{-1}$) | Site 2 (kg hr$^{-1}$) | Site 3 (kg hr$^{-1}$) | Site 4 (kg hr$^{-1}$) |
|---|---|---|---|---|
| No. of Transects | 26 | 28 | 21 | 22 |
| Single Source Gaussian | $1.0 \pm 0.3$ | $0.19 \pm 0.03$ | $1.09 \pm 0.16$ | $0.19 \pm 0.06$ |
| Multi-Source Gaussian | $2.2 \pm 0.9$ | $0.24 \pm 0.06$ | $1.8 \pm 0.3$ | $0.29 \pm 0.06$ |
| Large Eddy Simulation | $1.5 \pm 0.4$ | $0.18 \pm 0.03$ | $0.76 \pm 0.11$ | $0.18 \pm 0.03$ |

Table 6. Sources and magnitude of uncertainty in IGM emission estimates.

| Uncertainty Source | Notes | Expected Uncertainty |
|---|---|---|
| Atmospheric Variability[1] | Requires ≥10 transect to quantify, not independent from other uncertainty sources | 77% |
| Instrumental Uncertainty | LI 7700 expected 1Hz precision of 1.6 ppb | <1% |
| Plume Turbulent Diffusion[2] | Potentially source of uncertainty and bias | 25% |
| Source Height[3] | Z uncertainty low at distances >150 m | 15% |
| Source Location[3] | Combined x and y uncertainty low at distance >150m | 20% |
| Stability[3] | Potential 1 stability class discrepancy | 40% |
| Wind Speed[3] | Uncertainty scales linearly | 50% |
| Background[3] | Important for small concentration enhancements | 5% |

[1]Derived in Sect. 3.2

[2]Derived from the comparison of LES calculated dispersion and IGM modeled diffusion in Sect 4.

[3]Derived in Sect. 5.1

Table 7. Monte Carlo inputs for each uncertainty estimate.

| Uncertainty (Assumed Distribution) | SS Gaussian | MS Gaussian | SS Gaussian | MS Gaussian | Lower Limit |
|---|---|---|---|---|---|
| No. of Transects | 1 | 1 | 10 | 10 | 1 |
| Source location in x-direction (Gaussian) | 1 sigma = 16.7 m | 1 sigma = 0 | 1 sigma = 16.7 m | 1 sigma = 0 | 1 sigma = 0 |
| Source height (uniform) | 1-8 m | 1 m | 1-8 m | 1 m | 1 m |
| Atmospheric variability in observation (Gaussian) | 1 sigma = 100% (260 ppb) | 1 sigma = 100% (260 ppb) | 1 sigma = 100% (260 ppb) | 1 sigma = 100% (260 ppb) | 1 sigma = 25% (65 ppb) |
| Wind speed (Gaussian) | 1 sigma = 50% (0.75 m s$^{-1}$) | 1 sigma = 0 | 1 sigma = 50% (0.75 m s$^{-1}$) | 1 sigma = 0 | 1 sigma = 0 |
| Stability (uniform) | C-E | D | C-E | D | D |
| Background (Gaussian) | 1 sigma = 5 ppb | 1 sigma = 5 ppb | 1 sigma = 5 ppb | 1 sigma = 5 ppb | 1 sigma = 0 ppb |
| **Uncertainty Range (95% CI)** | **0.05q-6.5q** | **0.08q-3.2q** | **0.5q-2.7q** | **0.6q-1.6q** | **0.5q-1.5q** |

Table 8. Summary of advantages and disadvantages of each technique.

| | Measurement Time | Processing Time | Sources of Uncertainty |
|---|---|---|---|
| Single Transect Gaussian | Short (<10 min) | Short (~few minutes) | Many sources, atmospheric variability unconstrained |
| Multi-Transect Gaussian | Moderate (15-30 min) | Short (~few minutes) | Atmospheric variability constrained, diffusion unconstrained |
| Multi-Transect LES | Moderate to Long (15 min-1 hr) | Very Long (several days) | Atmospheric variability constrained, diffusion constrained |

**Figures**

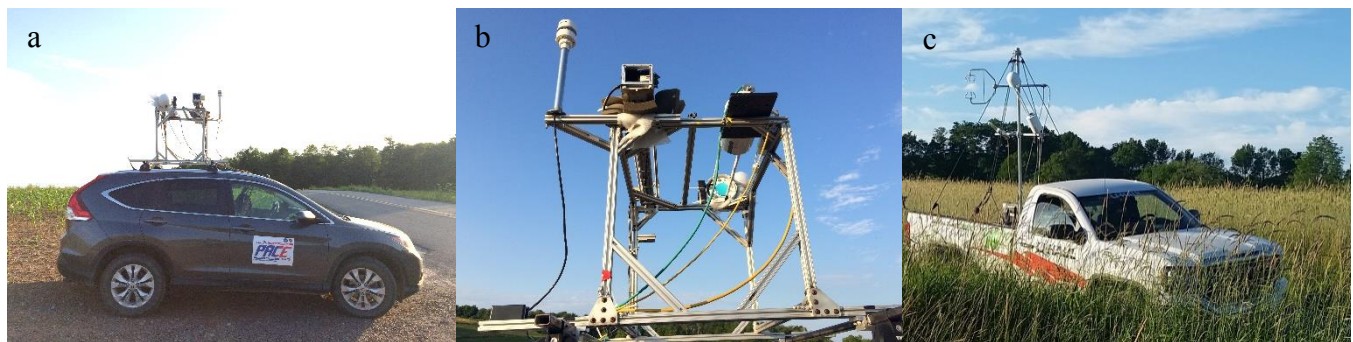

**Figure 1. (a) The Princeton atmospheric chemistry experiment (PACE), (b) close-up of PACE roof rack with instrumentation and (c) mobile tower platform.**

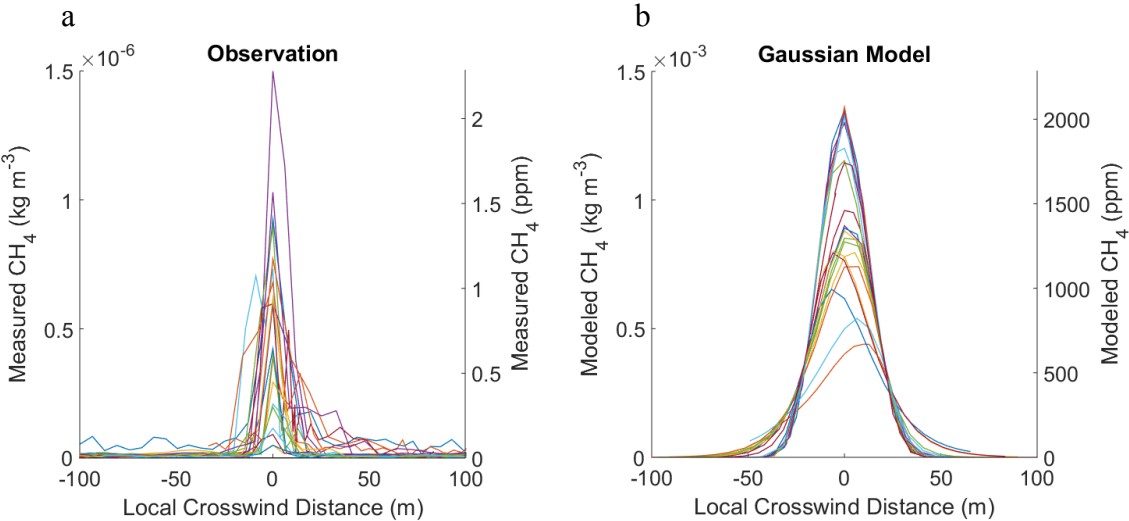

**Figure 2. (a) Observations of CH₄ at a site showing multiple downwind transects versus the local crosswind (y) distance. (b) Gaussian outputs along the same downwind transects. Note that the scale difference between the panels is due to the inverse methodology used where the model is set to a reference emission rate (of 1 kg s⁻¹).**

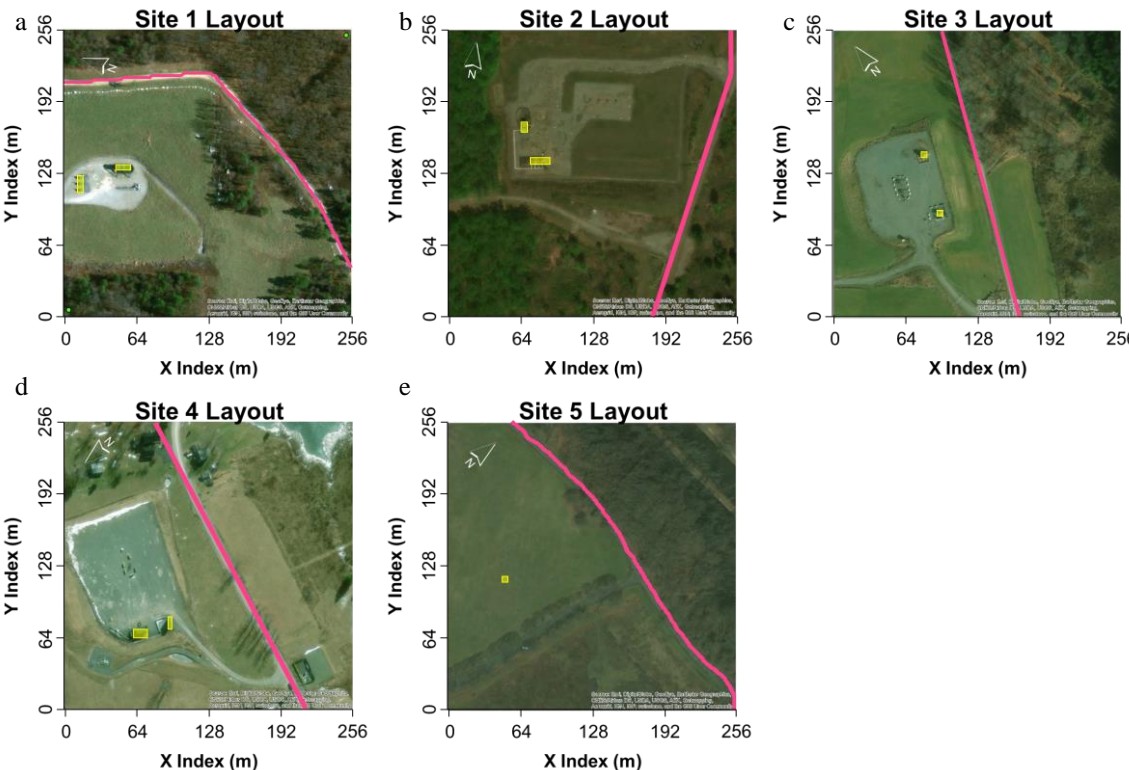

**Figure 3. Site layouts showing emitting structures in yellow and the road in magenta for (a) Site 1, (b) Site 2, (c) Site 3, (d) Site 4 and (e) Site 5. Wind is always entering the domain from the left.**

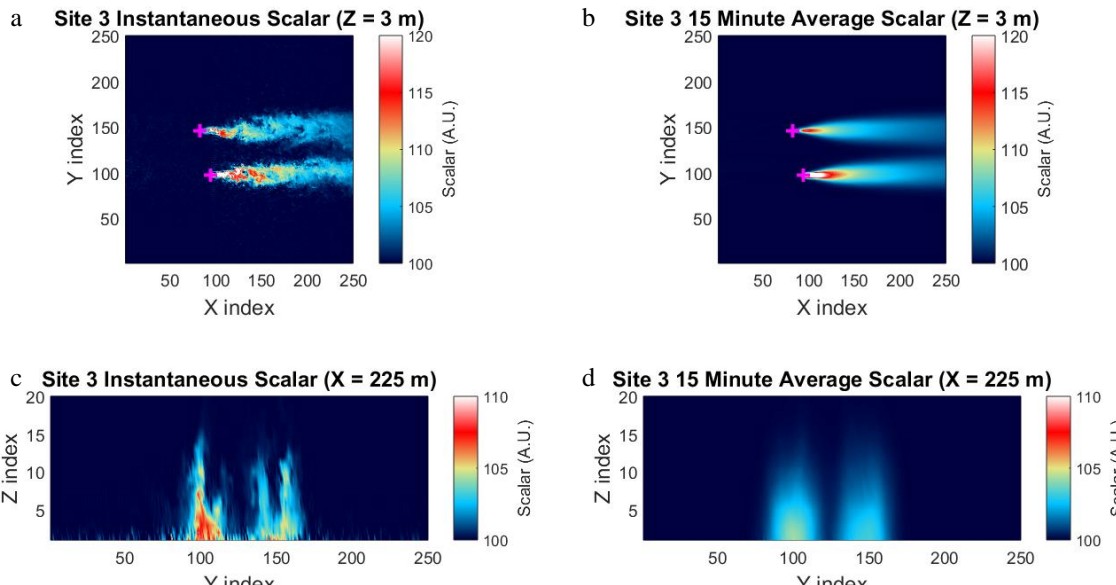

**Figure 4. Comparison of instantaneous (a/c) and 15 min averaged (b/d) plume for Site 3 from the LES. Panels a and b show scalar with arbitrary units (A.U.) in a x-y cross-section at an altitude of 3 m and panels c and d show a x-z cross-section at a 225 m downwind distance. The release locations are shown with a magenta marker (+).**

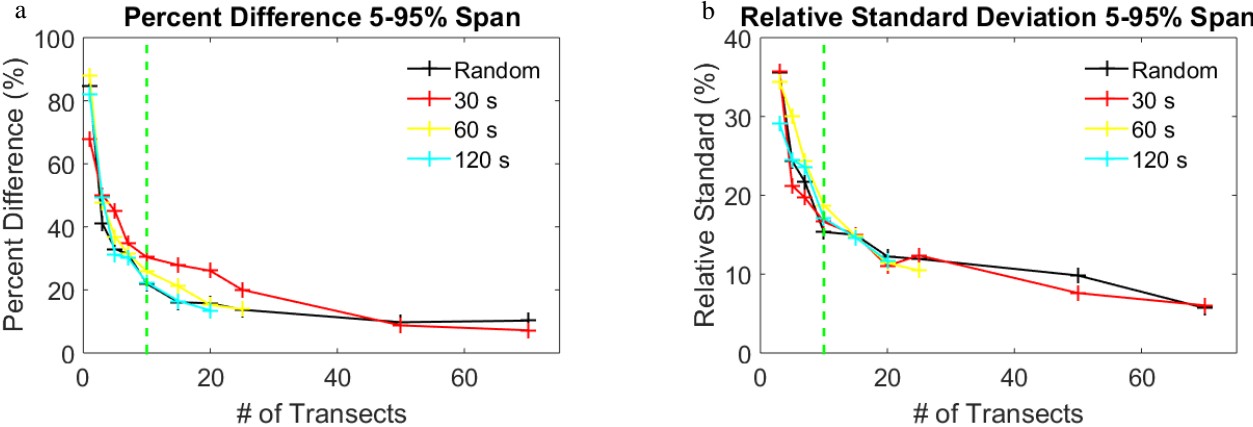

**Figure 5. The 5-95% percent difference (a) between emission retrieval and known simulation emission rate and (b) rsd of the emission retrieval using various amounts of transects and random, 30 second, 1 minute or 2 minute transect spacing. The recommended 10 transect criteria is shown in green.**

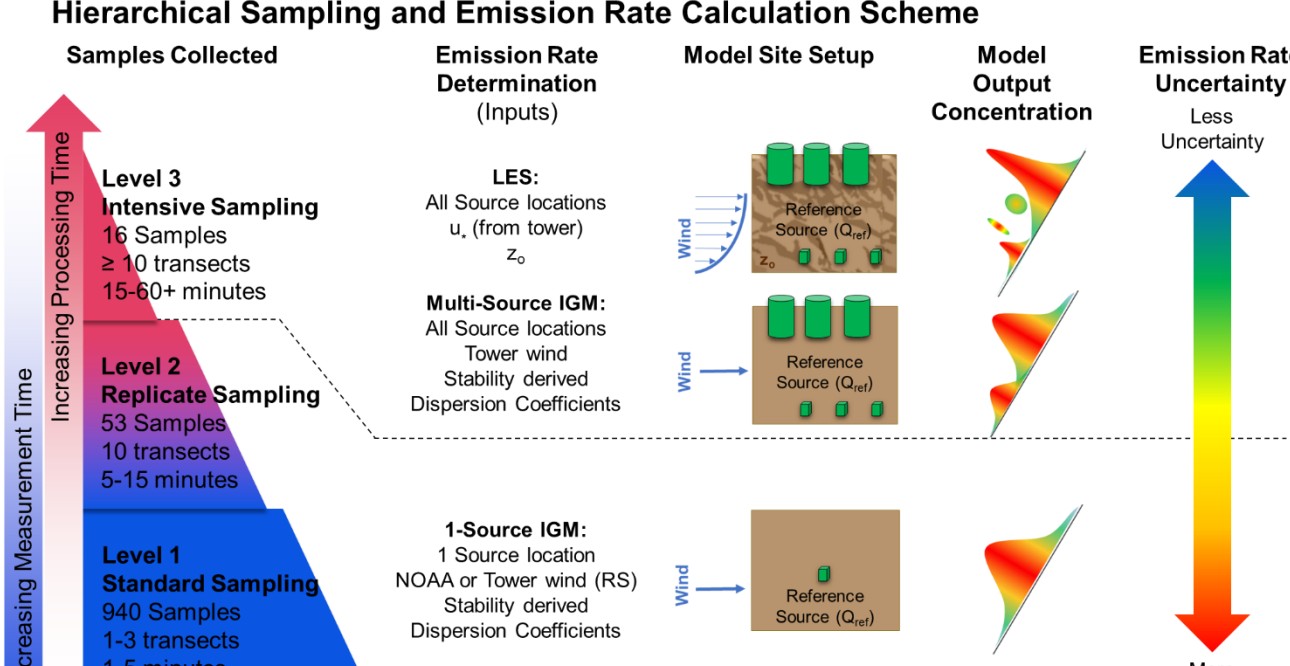

**Figure 6. Finalized sampling and emission rate calculation strategy employed in this study showing actual measurements with increasing complexity and decreasing sample size as well as models with increasing complexity and decreasing uncertainty. In this schematic $u_*$ is friction velocity and $z_o$ is terrain roughness length. The dashed lines denote the pathways data typically followed. Standard and Replicate sampling always went through 1-Source IGM pathway. Note that 1-source IGM can also be calculated for Intensive Samples as is shown for comparison purposes in this work. Abbreviations: Large Eddy Simulation (LES), IGM (Inverse Gaussian Model).**

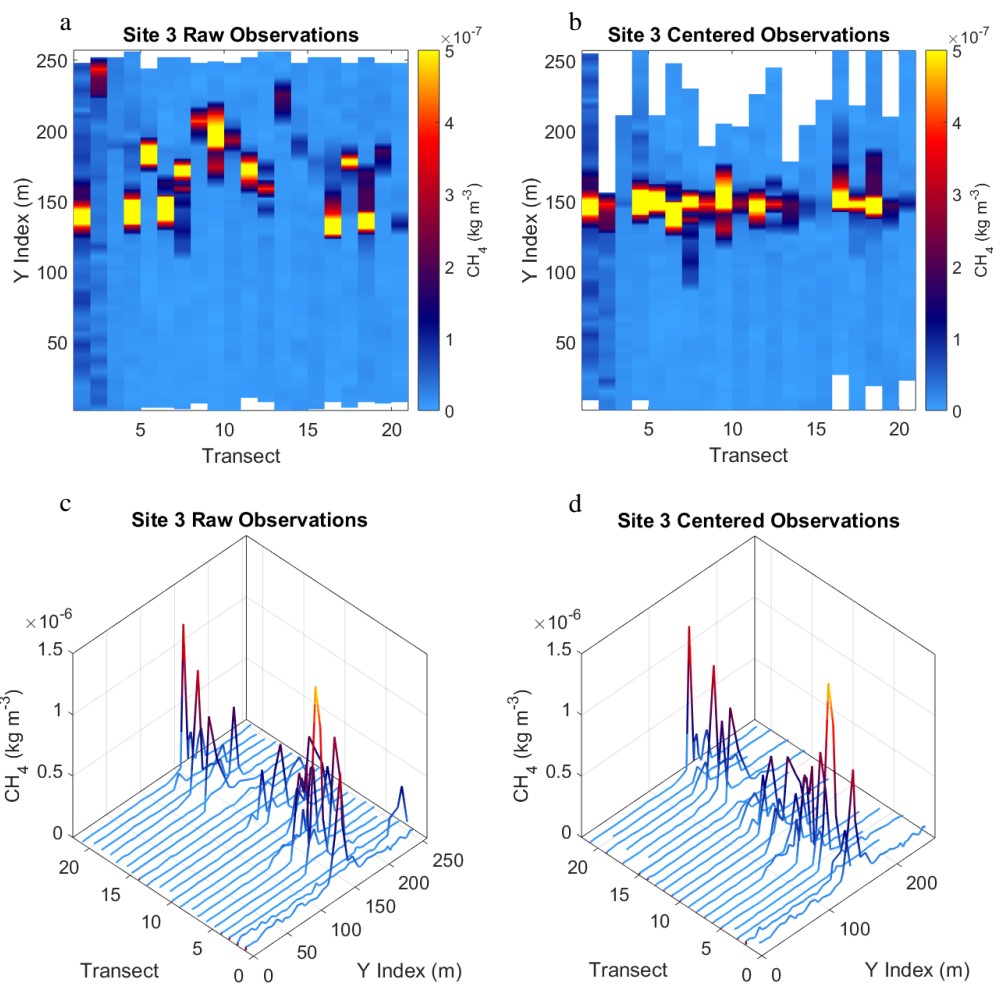

**Figure 7. (a) Site 3 raw observations as indexed to the LES domain and (b) after they have been aligned with the peak of the LES plume. Waterfall plots of the same data for (c) raw observations and (d) centered observations.**

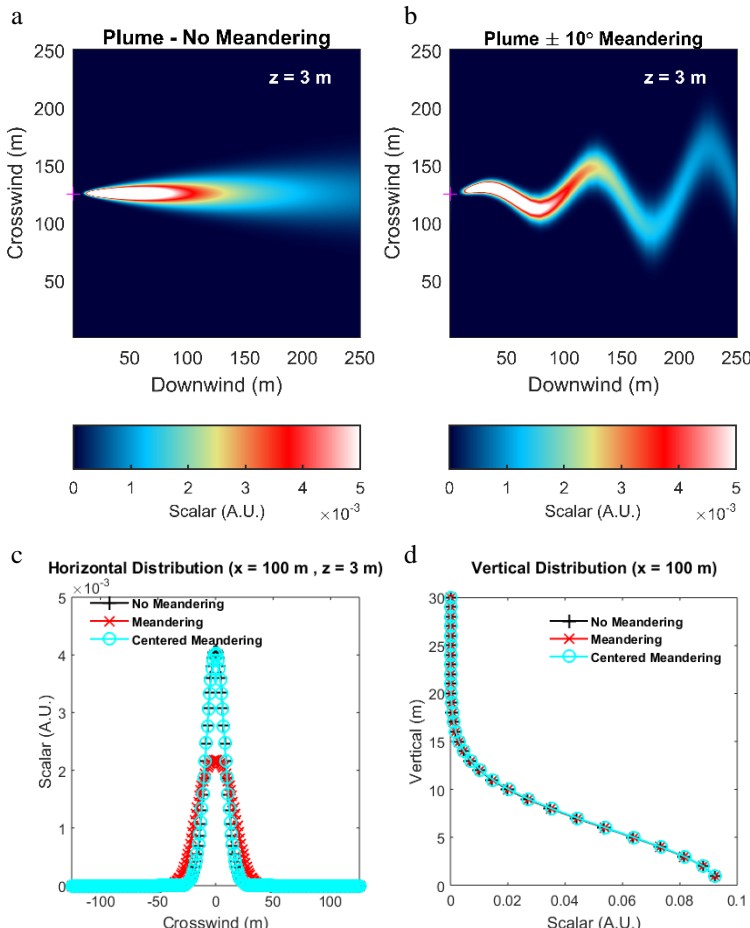

**Figure 8. A theoretical site with an additional plume meandering scale added shown in a top-down view at 3 m of (a) Gaussian with no meandering, (b) instantaneous Gaussian with 10° meandering. The comparison of (c) horizontal distributions of concentration and (d) vertical distributions of concentrations for three scenarios (no meandering, averaged meandering and centered meandering) are also shown. All units are arbitrary.**

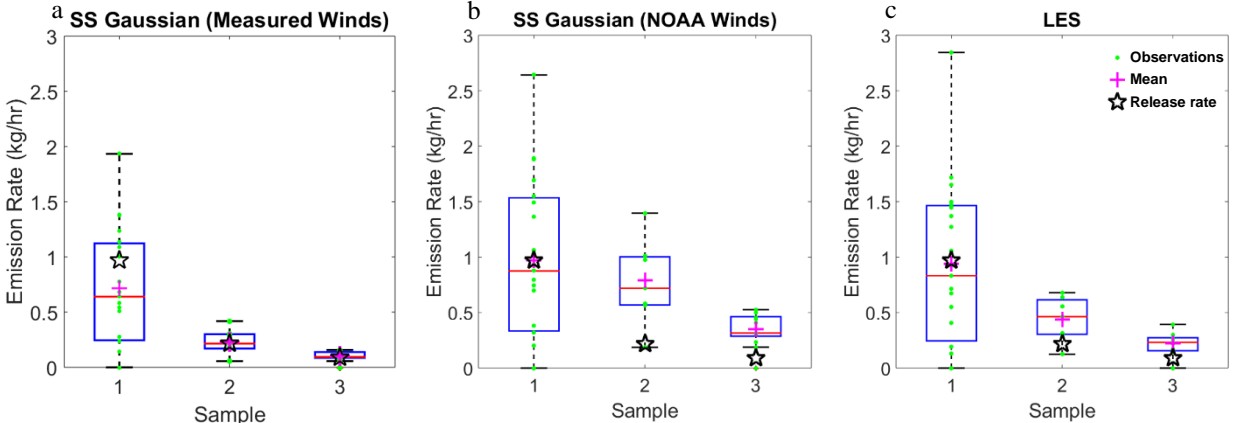

**Figure 9. Results from three controlled release experiments for (a) the Gaussian approach using tower measured winds, (b) the Gaussian approach using NOAA winds and (c) from the LES. Box and whiskers plots show the 50% percentile (red), 25 and 75% percentile (blue) and minimum and maximum values (black). The mean is shown in magenta, green dots represent individual measurements and the black star is the actual release rate. NOAA winds diverged from the measured winds significantly in releases 2 and 3.**

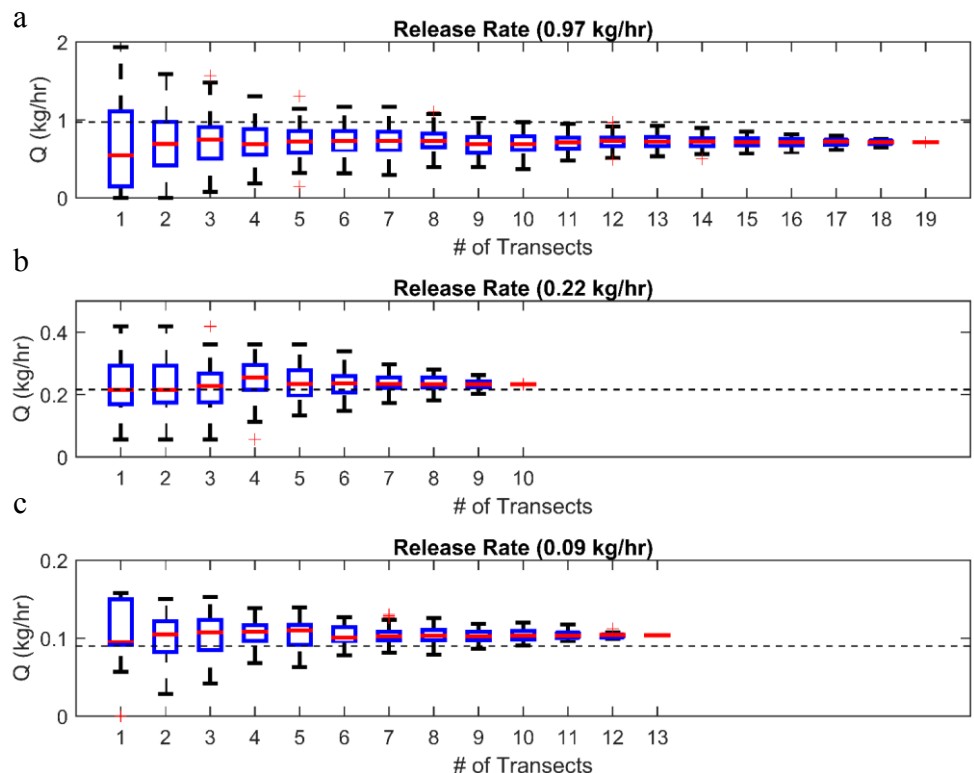

**Figure 10. A comparison of the convergence of the mean plume rate inferred from the IGM by averaging randomly selected transects for (a) the 0.97 kg hr$^{-1}$ release rate, (b) the 0.22 kg hr$^{-1}$ release rate and (c) the 0.09 kg hr$^{-1}$ release rate. The actual release rate is shown as a dashed black line. Box and whiskers plots show the 50% percentile (red), 25 and 75% percentile (blue) and minimum and maximum values (black) of the means obtained for each number of transects. Outliers are shown in red.**

# Site 3

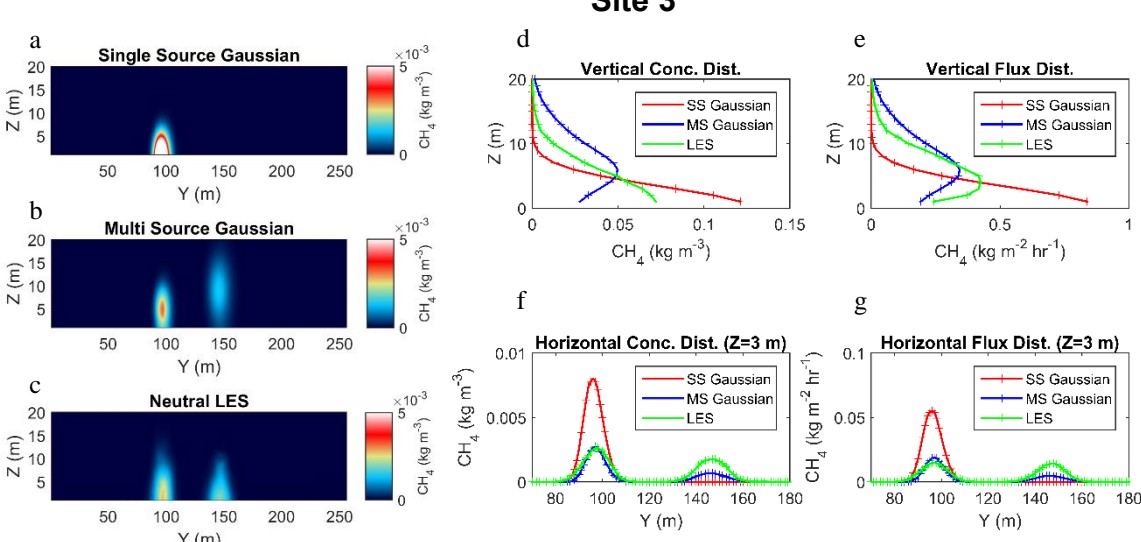

**Figure 11.** Comparison of three scenarios for Site 3 showing images through the downwind road plane (~30m downwind as shown in Fig. 3) of (a) single source Gaussian, (b) multi-source Gaussian and (c) averaged LES. The comparison of vertical distributions of (d) concentrations and (e) fluxes and of the horizontal distributions of (f) concentrations and (g) fluxes are also shown. The vertical LES flux corresponds to resolved fluxes only.