# Peer review of "Quantifying Uncertainties from Mobile Laboratory Derived Emissions of Well Pads Using Inverse Gaussian Methods"

_Atmospheric Chemistry and Physics, 2017_

## Referee Comment (RC1) · Anonymous Referee #1 · 26 Dec 2017

This paper presents an ambitious and valuable attempt to quantify uncertainty in estimate of methane emissions from natural gas wells using downwind, automobile-based measurements of methane concentrations. I believe that the strengths of the manuscript include the assessment of sampling error derived from idealized LES of plumes, and the attempt to provide an overall uncertainty assessment for this methodology. The methodology is not especially unique, but the breadth of the effort is important and worthy of publication.

The manuscript at present, however, contains shortcomings that make it unsuitable for publication in its current form. Major revisions are necessary. I first present my

overarching concerns, followed by a line-by-line assessment following the order of the manuscript.

Major issues.

1. Basic issues concerning the methods are not explained, thus many of the most important results cannot be clearly interpreted. I have included specific comments for many figures, tables, and text in my detailed comments. I mention what I consider to be the most important overall points here.

1.a) Table 5 is arguably the most important product of the manuscript, but the basic elements that are used to build this table are never clearly articulated. The manuscript should be reorganized to clearly explain the basis for the uncertainties being imposed in the inputs to the Gaussian plume emissions estimates, and the uncertainties in these inputs should be clearly stated. This is true with a limited number of inputs in this table (instrument precision, stability class), though even for one of these inputs, the source of the uncertainty in the inputs to the model is not clear (e.g. why is one class the assumed bound of uncertainty?, what time resolution is associated with 5 ppb precision in the LiCor 7700?). Atmospheric variability, the most important issue according to the table, is never defined. The main finding of the manuscript must be clearly defined to be useful and interpretable.

1.b) The method for determining single vs. multiple sources, or the method for evaluating the number and spatial distribution of multiple sources, is never explained.

1.c) The method used to remove background concentrations from the field measurements is critical, and is never discussed. This is a major source of uncertainty in plume dispersion estimates, and is not included in the results. This must be addressed.

2. The averaging applied to the downwind transects is arguably inappropriate and suggests a fundamental misuse of the Gaussian plume model. The Gaussian plume model describes the ensemble average of atmospheric concentrations downwind of a

point source. This includes the fact that the instantaneous plume will meander over the course of time due to atmospheric turbulence. The model is based on the assumption of a separation of scales between atmospheric turbulence, and mesoscale or synoptic-scale atmospheric flow variations. The authors have chosen, however, to align each measurement downwind of their point sources by the peak concentration measured on each transect. The wind direction, therefore, changes with every transect. The mean wind in the Gaussian plume model should be the average wind across the time span of all of the downwind transects (which are conducted within an hour, over which the atmospheric dispersion conditions are assumed to be steady-state), and should not vary every few minutes with each transect. This approach to averaging would create a significantly broader observed plume. It is not clear to me how this might alter the estimated emissions, but this is a misapplication of the plume dispersion model that should be corrected. If the dispersion conditions within the time span of the downwind transects are not steady, this must be explicitly identified and treated in the analyses.

3. Given the limited spatial domain, the LES used in this study has no large-scale turbulence, thus its value for comparison to plume observations is uncertain. Its use for an observational system simulation experiment is acceptable, given the caveats that it only simulated neutral conditions and has no larger-scale turbulence. The authors do explain the limitations of neutral stability, but do not acknowledge that the simulation will by necessity truncate the spectrum of atmospheric turbulence that will contribute to issues like meandering of the downwind plume.

4. A major element of the uncertainty assessment relies on the controlled release experiments. This requires characterization of the uncertainty in the rate of release. No assessment of the uncertainty in the release rates are given. This must be addressed.

5. The measurement protocol chosen focused on morning and evening conditions to search for neutral atmospheric stability. Sounding on calm mornings, however, are far neutral, and morning and evening conditions are characterized in general by rapidly changing atmospheric stability. Plume dispersion modeling assumes steady atmospheric stability and mixing conditions. What is done to screen the observations to avoid rapid transitions in stability that clearly violate the conditions for the Gaussian plume model?

Detailed comments.

1. Page 1, Line 14-15. "...a hierarchical sampling with increasing complexity was implemented" is, I believe, what you mean to say. Though I'm not sure what "with increasing complexity" means. It is hierarchical, thus complex.

2. Page 1, transects, lines 15-20. How are multiple transects separated in time? Across multiple days? Or all on the same day / within the same hour? This is an important methodological distinction.

3. Page 1, Line 20. "in most cases with average differences" Be more precise. Is 25% the average difference or not?

4. Page 1, Line 24-25. "Approximately 10 repeat transects spaced at least 1 min apart are required to produce statistics similar to the observed variability over the entire LES simulation period of 30 min." This is not very informative. What is relationship between number of transects and precision of determination of the emission rate?

5. Page 1, Line 25-26. "In addition, other sources of uncertainty including source location, wind speed and stability were analyzed." This is too vague to be informative.

6. Page 1, Line 26. "atmospheric variability" is too vague. Do you mean to say sampling error caused by atmospheric turbulence? If so, isn't that a function of the sampling strategy? Is atmospheric sampling still the dominant source of uncertainty with 20 downwind transects?

7. Page 1, Lines 28-30. What is "this condition?" What is "this metric?"

8. Page 3, line 1. "lag behind the standard." In what respect?

9. Page 3, equation 1. Please incorporate the equation into the English, and please

define C and the origin.

10. Page 3, line 15. "Gifford's"

11. Page 3, line 16-17. It is not necessary to describe the stability classes. The model does not predict a PDF of the concentration. It predicts the average concentration. These are very different.

12. The uncertainties in Table 1 are not helpful as currently presented. This Table must be revised prior to publication. I understand that they are all reported by the individual authors, but this manuscript does present them for the purpose of comparison. Many of the techniques and papers have radically different spatial and temporal domains and sampling density, and the uncertainties quoted are thus not at all comparable. How, for example, can aircraft estimates of the mass balance for an entire basin be compared directly to the uncertainty of a single chamber sample, or a single Gaussian dispersion estimate, without an understanding of what this uncertainty bound is trying to represent? This table could be enhanced to include some information about the domain and sampling density associated with these studies. This would be challenging, but much more interpretable.

13. Page 4, lines 3-12. Are these uncertainties for single emissions estimates from a site using an individual "transect" of data? Please clarify. See my concerns above. Without specifying more about the temporal and spatial domains, these uncertainties are difficult to compare or interpret.

14. Page 4, line 19. What is "an averaging plume for unconstrained instantaneous measurements"? Please clarify. I study boundary layer meteorology but do not understand what you are trying to say.

15. Page 4, line 25. I don't understand, "this method." Are the authors proposed a new standardized methodology for quantifying emissions? Or is this a study of the uncertainties in plume measurement / source quantification methods?

[Figure]

16. Page 5, line 5. The methane span gas concentration is quite low for the measurement of plumes at close range. Are there any tests of the linearity of the instrument at higher concentrations that I assume are encountered driving downwind of strong methane sources?

17. Page 5, lines 19-20. "site were screened": Does this mean that sites with trees were eliminated from the sample? The text later states that it is hard to measure a plume at a distance greater than 300m if trees are present. Can you please explain the site selection algorithm more carefully?

18. Page 6, line 7-9. Does the NOAA web site provide observations, or numerical model reanalyses? Please clarify. How does this NOAA web site account for the impact of local roughness on atmospheric stability? The hilly, forested, heterogeneous landscape of Pennsylvania will have a significant impact on local dispersion.

19, Page 6, equation 2. As with equation 1, please make equation 2 part of the sentence.

20. Page 6, line 20. This description is insufficient. The Gaussian plume model describes only the concentrations caused by the local source. The measurements do not. A concentration background must be defined and subtracted from the observed time series. This is a critical and non-trivial step. How was this done? I am also concerned that this method does not take into account the correlation between the modeled and observed concentration enhancements. Systematic errors in the dispersion would not be identified by simply matching the integrated downwind enhancement.

21. Figure 2. This figure is concerning. First, the figure shows concentrations that differ by a factor of 1000, and no explanation is offered. Second, the background that must have been removed from the observations is not discussed. Third, there appears to be a systematic difference in the width of the observed vs. modeled plume. This implies a systematic error in the dispersion coefficient, which will lead to a systematic error in the emissions estimate. How is this addressed? Finally, given that instrument

characteristics are expressed in units of ppb, it would be useful to also present down-wind concentrations in these units. It would be easy to include both mass density and mole fraction as two different y axis labels.

22. Page 7, line 11. How is steady-state evaluated?

23. Page 7, line 11. What does it mean to "rescale" the LES output to the measured momentum flux? Isn't the LES set up to simulate these sites? Why is rescaling necessary?

24. Section 3.1 The LESs are truncated in the vertical and thus cannot include large-scale turbulence that will influence the sampling statistics. This will be particularly important for unstable conditions. How is this dealt with in designing the sampling strategy?

25. Section 3.1. The LESs say nothing about sensible heat fluxes. Turbulence statistics change dramatically as a function of stability conditions. What stability conditions are simulated?

26. Page 7, line 25. Short intervals will sample the same structures in the turbulence. This is well known. Some of the existing literature on sampling statistics in atmospheric turbulence should be cited here.

27. Page 7, line 26. What does "1-N" mean? What is N? I read the description, but I do not understand what this means. What defines the number of samples available in a numerical simulation?

28. Page 8, lines 8-9. What is the difference between turbulence and plume meandering? Isn't plume meandering caused by atmospheric turbulence?

29. Page 8, lines 10-12. What does it mean to "reflect the actual variability in the atmosphere"? This is not meaningful. Please define a quantitative tradeoff between transects and flux retrieval.

30. Figure 3 is not very helpful. Please include vegetation and topography.

31. Figure 4. Images of instantaneous vs. averaged plume structures do not need to be published. This is well known. The manuscript has no references to several decades of simulations of plume dispersion. This is a serious hole in the scholarship of this manuscript. Look up, for example, the publications of Jeffrey Weil, circa 1990.

32. Figure 5. Issues. 1. There is a lot of blank space. 2. The percent difference figures show results from a distribution. What are the elements of the distribution? Is each "pseudo-transect" an element of the distribution? Please define this clearly in the caption. 3. A standard deviation is a description of a population. Are there multiple populations whose statistics are being compared? Please clarify. 4. What does "random seconds" mean? Why is this useful? 5. There is little additional information given in the 30/60/120 second sample repeat figures, and the differences are difficult to see. It would be more effective to show one of these, then have one additional figure showing the differences caused by changing the sampling interval.

33. Page 8, lines 18-19. The original plan is not relevant at this point. Present the number of samples obtained.

34. Page 8, line 24. What is the "average maximum percent difference"? Maximum among what? Average across what? And why is this relevant? Please also explain the populations used to define the standard deviations.

35. Page 8, last sentence. What is the population being discussed here? Why are these numbers important?

36. Figure 7 does not explain how the number of sources will be determined. I also disagree with the far right-hand bar. A more complex simulation does not ensure less uncertainty, and "model uncertainty" is not defined. If there is a precise definition of uncertainty that can be shown to be reduced with the LES, please explain and define this.

[Figure]

37. Page 9, line 17. What is a "warm up time"? (I see here that the determination of steady state is explained.)

38. Page 9, line 21-24, and figure 8. This discussion and figure displays a fundamental problem with the LES setup, and the authors' interpretation of these data. The "real world" meandering of the wind that is being described is turbulence in the atmospheric boundary layer. A Gaussian plume model describes the average state downwind of a point source, which would be properly represented by the average of all of the transects, without "aligning" them to match the peak concentration on each transect. Aligning these is the equivalent of performing some sort of high-pass filter.

39. Similarly, I am now concerned about the processing of all of the transect data. In Figure 2a, for example, have all of the transects been arranged so that the x direction changes from transect to transect? If so, the same odd filtering of turbulence has been applied. This is not appropriate for comparison to a Gaussian plume model. I expect this could explain the difference in plume widths that is evident in Figure 2 (note the modeled plume is wider that the observed plumes that may have been filtered to remove large-scale turbulence)/.

40. Please explain the theoretical basis for equation (3)?

41. Section 4.1. How do you distinguish among single vs. multiple sources? This is not explained.

42. Page 10, line 2. The two Gaussian retrievals "compare"? What does this mean?

43. Figure 9. Once again, please explain the populations of data that go into these box and whisker plots. Is each point a transect? If so, how many transects make up each population?

44. Page 10, line 13-14. I do not agree that the results show no apparent bias. Figure 9a has winds that are known to be too high, according to the text. Figure 9b shows a systematic underestimate of emissions. Figure 9c shows varied results.

45. Figure 10. What is the level of uncertainty in the release rate? If this release is being used to evaluate the methodology, then the release rate must be carefully calibrated. How has this been done?

46. Page 10, line 22-23. Only one outlier with zero emission is shown in Figure 10.

47. Table 2. Please report the observe and simulated winds, and the observed and simulated stability conditions. Please include more descriptive information in the table caption. In particular, please note how many transects were collected at each site, the time span of the transects, and the source of the uncertainty values given in the table. If these are standard deviations based on emissions derived from individual transects, and the purpose is to compare mean values, the authors should report the standard deviation of the mean, not the standard deviation.

48. Table 3. Please include more descriptive information in the table caption. In particular, please note how many transects were collected at each site, the time span of the transects, and the source of the uncertainty values given in the table. If these are standard deviations based on emissions derived from individual transects, and the purpose is to compare mean values, the authors should report the standard deviation of the mean, not the standard deviation. Also, please report winds and stability. And were the winds observed, or taken from NOAA reanalyses?

49. Page 10, lines 27-28. See my earlier concerns about centering of the plumes on each transect. I believe this is an erroneous interpretation of plume dispersion.

50. Page 11, lines 7-8. The overall differences are small? The flux estimates in Table 3 differ by as much as a factor of two. This seems large to me.

51. Figure 11. This figure points to many questions and problems. 1) What are the equations for the vertical and horizontal flux profiles in the Gaussian plume model? 2) What is the position in the domain where the profiles (figure d-g) are computed? 3) What is the distance downwind for everything shown in this figure? 4) How are the

multiple sources chosen? As best I can tell this is never described anywhere in the manuscript. 5) Does the LES flux include subgrid and resolved fluxes? Please explain, and delineate these two sources so that it is clear what fluxes are explicitly resolved. 6) What does z=3 mean? 3 what? The same comments and consternation hold for the similar figures in the supplementary materials.

52. Page 11, lines 12-14, sentence starting with "regardless." I do not understand what you are trying to say.

53. Table 4. What is the distance from source of the measurements? Are "emissions" those estimated from transect measurements? How many transects? What are the meteorological conditions for these measurements?

54. The text in the first paragraph of section 5.1 does not appear to match the results in Table 4. The text notes that unless the change in source location in the x direction is similar to the distance of the measurement downwind, that the impact of the estimated emission is small. Table 4 shows two examples, one of which has a 150% change in source strength estimate, and another a 7.5% change in source strength estimate, and as best I can tell (Table S1?) the measurements are both from about 150m downwind of the source, with the site with a larger % change having measurements farther down-wind. Based on the results presented by the authors, as best I can interpret them, I disagree with their conclusions.

55. Figure S10 (b) is uninterpretable. What is the "Ratio between the sum in y and distance x of the scenarios"? Is this truly a ratio of distances that is being plotted? Please clarify.

56. Figure S11 (b). See comment above regarding Figure S10.

57. Page 11. "NOAA wind speeds differed from the tower data on average by 50%." This is uninterpretable. Please rewrite this to be meaningful.

58. Figure S12. Please define this ratio, as with Figures S10 and S11.

59. Page 12, lines 7-8, "The magnitude of the difference between consecutive stability classes is relatively consistent, averaging 40%." Figure S12 does not suggest simply defined average, as the signs and magnitudes of the differences change as a function of distance downwind. Please explain what this 40% value means.

60. Page 12, lines 12. Please define "absolute terrain slope."

61. Page 12, lines 24-28. The methods for quantifying the uncertainties should be explained carefully in the methods. This rapid-fire, qualitative overview of the methods behind Table 5, arguably the most important result of the manuscript, is insufficient.

62. Page 12, lines 30-34. See previous concerns about the lack of larger scale turbulent motions in the LES used for this study. I agree in general that the LES used here should provide a lower estimate of turbulent sampling error, but it also will not contain the full spectrum of atmospheric turbulence and true turbulent sampling error. This should be explained, and references to the rich literature of the full spectrum of atmospheric turbulence and the limitations of LES that is limited to the atmospheric surface layer should be included in the manuscript.

63. Page 13, line 6. The ranges of the input values used in the Monte Carlo simulation and their distributions must be described, or these results are meaningless.

64. Table 5. See prior concerns about the lack of documentation of the methods for assessing these uncertainties. In addition, how is the "total"uncertainty assessed? Is this the result of the Monte Carlo simulation described in the text? If so, see prior concerns about documentation of this experiment. If not, please explain how this total is computed.

65. Figure 12 results vary by a factor of 1000 with no explanation.

66. Section 5.3. As best I understand this work, the order of averaging that is varied is certain to cause no significant change since the relationship between concentration and source strength is linear. Thus this comparison is not informative or significant. If

anything is nonlinear that could cause a difference, please explain.

67. Page 14, lines 14-15. I do not see how your results justify this statement.

68. Page 14, lines 22-23. I do not yet agree with the assessment that no bias is observable. This cannot be assessed until the uncertainty bounds in Table 2 are clarified. If the mean values differ significantly, then there is observable bias.

69. Page 14, line 23,"LES is therefore not required for studies where source strength calculation is the main goal." The authors have clearly avoided complex terrain where LES is most likely to be needed. This broad and general statement is not justified by the research presented in this manuscript.

70. Page 14, lines 24-25. "From this we use Monte Carlo analysis to extrapolate that the 95% confidence interval for sites with standard sampling (n=2) ranges from 0.05x–6.0x where x is the emission rate." This is a potentially important result, but this is offered as a passing comment. The methods behind this calculation are not presented. This is not acceptable for publication. This is an important result whose methods must be clearly explained and defended.

71. Page 15, lines 3-4. Rella et al, (2015) did not use a Gaussian plume dispersion approach, and attempted to measure the entire vertical extent of the plume. This manuscript notes that vertical dispersion is a major source of uncertainty in their results. It seems very likely that Rella et al's approach should, therefore, yield a significantly smaller uncertainty estimate than a method that relies on a Gaussian plume dispersion model to quantify vertical dispersion.

72. Page 15, lines 5-6. "As described in Sect. 3.2, the observed atmospheric variability can range from 10-200% meaning that in-situ observations of variability and post screening out conditions with unacceptably high variability may be a viable way to reduce uncertainty." What is the quantity "atmospheric variability," that can range from 10-200%, and 10-200% of what? What criteria of "unacceptably high variability" can

be used to screen out data and thus reduce uncertainty? This reasoning is either explained poorly or imprecise, and the conclusions stated are thus either uninterpretable or inaccurate. This statement is not suitable for publication.

73. Page 15, recommendations. Point 1 cannot be satisfied in many cases. What happens when measurements are needed for a location that does not satisfy these criteria? I don't understand point 5.

74. Point 6 is not a significant recommendation resulting from this research.

75. I do not understand "the strategy" proposed in Point 8. It is not clear how this collection of measurements is proposed as an integrated sampling strategy.

76. Section 8. Data should be publicly available, and not restricted to access only via correspondence with the authors.

---

## Referee Comment (RC2) · Anonymous Referee #2 · 23 Jan 2018

This manuscript presents a methodological and large-scale study of dispersion estimates, comparing Gaussian estimates to LES and tracer releases. A large dataset of wellpad emissions is analyzed, and a set of estensive measurements at a smaller number of sites provides better data for intercomparison. Methods are clearly described, and the results are formulated into recommendations so as to be most useful to the atmospheric measurement community. This important manuscript publishes a robust assessment of uncertainties that result from commonly-used Gaussian dispersion methods on mobile plume transect data.

I particularly appreciated the effort the authors went to to express uncertainties in the

95% confidence framework, with skewed bounds if needed, for best comparison to previous studies (e.g. section 6). These uncertainties are larger than previously measured, but the assessment is much more comprehensive, making them all the more worthy of publication. These uncertainties are a major result of the paper. I recommend including uncertainty bounds for SS gaussian, MS gaussian and MS LES in Table 1, as well as in the abstract.

I recommend publication after considering the minor comments listed here.

Main points:

1. Timescale

p. 3, Section 1.1: I think this section warrants a discussion of timescales of Gaussian simulations. There is a good deal of variation in what the the atmospheric modeling community determines to be the appropriate timescale for the A-D stability classes. The consensus seems to be on the order of 10-15 minutes. See Fritz et al. DOI:10.13031/2013.18501

A discussion of timescales is also relevant to Reviewer #1's second comment about averaging of plume transects vs the meandering plume.

2. Winds

p. 6, line 7: Explain why the choice was made to use downloaded interpolated hourly winds (3-hour interval data, originally) when there were presumably measured winds on-site for the intensive IGM sites? I am also concerned in how well this interpolated 1-hour met will work for determining stability parameters of quick transects. See also previous comment on timescale. How would results differ with measured met?

The mobile laboratory also presumably had a measurement of mobile wind, and yet this measurement has not been mentioned at all. Do these data exist? What problems were encountered with this wind (equipment failure, uncertainty in calculating true winds for moving vehicle, etc)? Depending on the conclusions, item 2 in the Recommendations

(p. 15) might be updated. Table 2 might also be updated.

3. Clarity in describing source data

In many parts of the manuscript, it was difficult to understand the nature of the underlying dataset, or equivalently, which hierarchical "level" of data was being used. For example:

p. 10, section 4.3: Are these tracer releases? Are they real emissions measured with high N and simulated with all methods?

Fig 11: Is this purely simulated data? If measured data is available, show a sample transect on the same scale.

4. Others

Table 1: Add a column to this table describing the the type of uncertainty noted (e.g. 95% CI, std dev, etc. See also previous comment about adding in results of this study.

p. 7, line 1: What is the purpose of allowing the simulation to use other point sources in order to best simulate the emissions? Are these other point sources input by the user based on observed equipment/sources, or are they automatically generated? What is the reasoning behind comparing multi-point LES to a single-point gaussian simulation at all (as described p. 9, line 10 and Fig 7)? If it is for ease/quickness of data processing, describe this.

p. 9, line 5: Did scientists have site-access or other means to verify this assumption? If tank batteries are present, tank emissions can often overwhelm wellhead emissions. Is this dealt with in the data somehow?

p. 13, line 9: A figure showing the distribution, where it is possible to note the mode and 95% confidence intervals, is needed to understand this skewed distribution and should be included as SI

Minor comments/ typos:

Table 3. Describe SS, MS and LES abbreviations in caption.

Table 5: It seems like the line for Source Location should differentiate between x, y and z directions, as done in the text, or at least include a range of uncertainties.

Table 6: reformat Sources of Uncertainty column (e.g. left-justify) for ease of reading.

Figure 2: Consider breaking up this graphic into more panels because it is impossible to compare individual traces as-is. Also homogenize the vertical scales. There is currently a 3 order of magnitude difference in the scales. Is this correct?

Figure 4: Make the bounds of Yindex the same in all graphs. Furthermore, it would be useful to produce a mixing ratio vs Y index graph (or Scalar vs Yindex) at a height of 3 m to show what hypothetical measured data would look like.

Figure 5: Is it possible to show uncertainties at 95% confidence on these graphs? This would be more useful than the 10-90th percentiles. Replace "50% percentile" and similar with "50th percentile" throughout the caption and text

Figure 7: expand SS, MS and LES in the graph, or note their meaning in the caption.

p. 4, line 13: "Additionally, implementing..." These last two sentences seem out of place in the paragraph discussing uncertainties. Elaborate or move.

p. 5, lines 1, 5, 9: LICOR is typically written as "Li-COR"

p. 7, line 31: briefly define percent difference - what is the "true" value used as denominator.

p. 8, lines 2-4 and Figure 5: renumber figure panels so that consecutive letters refer to the same transect interval. As it is, line 2 should read "(b & d)" instead of "(c-d)"

p. 8 lines 14-19: reference Figure 6 here somewhere

p. 10, line 14: which result shows no apparent bias?

p. 10, line 20 and Fig 10: When discussing Fig 10, the over/underestimate of results

as they converge seems important to mention.

p. 11, Section 5.1: It would be interesting to express these comparisons in relative distances as well.

p. 12, line 1: reference Table 5 in this section.

p. 15, line 1: include range, e.g "...standard sampling uncertainty range of 0.05x - 6.0 x is greater..."

---

## Author Comment (AC1) · 3 May 2018

**Responses to Anonymous Referee #1**

We thank the reviewer for the meticulous review. The comments have helped to refine and clarify the manuscript. All comments are addressed below and changes in text, figures, captions or tables have been included as appropriate. The reviewer's comments are in bold with our specific responses below in regular text and quoted sections from the manuscript in italics.

**Anonymous Referee #1 Comments**

**Major issues.**

**1. Basic issues concerning the methods are not explained, thus many of the most important results cannot be clearly interpreted. I have included specific comments for many figures, tables, and text in my detailed comments. I mention what I consider to be the most important overall points here.**

    a. **Table 5 is arguably the most important product of the manuscript, but the basic elements that are used to build this table are never clearly articulated. The manuscript should be reorganized to clearly explain the basis for the uncertainties being imposed in the inputs to the Gaussian plume emissions estimates, and the uncertainties in these inputs should be clearly stated. This is true with a limited number of inputs in this table (instrument precision, stability class), though even for one of these inputs, the source of the uncertainty in the inputs to the model is not clear (e.g. why is one class the assumed bound of uncertainty?, what time resolution is associated with 5 ppb precision in the LiCor 7700?). Atmospheric variability, the most important issue according to the table, is never defined. The main finding of the manuscript must be clearly defined to be useful and interpretable.**
    Various clarifications have been made throughout the text that we will reference in response to later specific questions that we believe help clarify the methodology. For instance stability classification is discussed in Sect. 5.1 (p. 15-16). The time resolution and uncertainty of the LI-COR are now included in Table 6 (previously Table 5). While atmospheric variability was discussed in Sect. 3.1, we have added an additional clarification on p.4 lines 24-25.
    *"Insufficient averaging would result in random errors that we refer to in this paper as the uncertainty related to 'atmospheric variability'."*
    Additional clarification of this concept has also been made on p. 12 lines 2-6.
    *"Another way of describing the atmospheric variability as used in this work would be transect to transect variability, which encompasses all random errors that lead to differences between one transect through the plume and the next. These are hence the errors associated with insufficient averaging of the turbulent field and would be reduced as the number of averaged transect increases (Salesky and Chamecki 2012)."*
    We have also added additional discussion in Sect. 5.2 to facilitate comprehension of this concept as we have specifically used it and at this important section of the work on p. 20 lines 22-23.
    *"The largest contributor to total uncertainty is atmospheric variability, the random error induced by insufficient averaging of the turbulent instantaneous plume."*
    b. **The method for determining single vs. multiple sources, or the method for evaluating the number and spatial distribution of multiple sources, is never explained.**

Additional discussion of how sources were determined has been added to Sect. 4.1. on p.13 line 16-19.

*"As on-site access was not available and well pads may contain multiple sources, all large structures were treated as separate point sources. These were visually identified during measurements and exact coordinates were confirmed in Google Earth. The center of all sources was used as a point source. The identified sources were always gas processing units or storage tanks."*

c. **The method used to remove background concentrations from the field measurements is critical, and is never discussed. This is a major source of uncertainty in plume dispersion estimates, and is not included in the results. This must be addressed.**
We have added a line to Table 6 to indicate the expected contribution of the background selection. In addition, we have added the following text to Sect. 5.1. on p. 19 lines 10-19.

*"An important consideration is the assignment of the background value to calculate plume enhancement. For this work, the background was calculated as the minimum value from the plume transect because the averaged 1 Hz data generally showed a uniform background near the plumes. This criteria was also compared to a scenario where the background was calculated from the average of the lowest 2% of observations in a transect with very similar results again indicating that the background value is very stable. Across all ~1000 samples the background had a median standard error of 5 ppb. When the 5 ppb tolerance is applied to the same data set, the median change in calculated flux was 4.4%. This means that the background is expected to contribute an additional ~5% uncertainty. It should be noted, however, that for very low signals the background can become a major source of uncertainty. The average peak enhancement was ~1250 ppb with a median value of ~260 ppb. Signals in this data set were screened to remove sites with peaks less than 50 ppb as this was deemed to be below the limit of detection of our system."*

**2. The averaging applied to the downwind transects is arguably inappropriate and suggests a fundamental misuse of the Gaussian plume model. The Gaussian plume model describes the ensemble average of atmospheric concentrations downwind of a point source. This includes the fact that the instantaneous plume will meander over the course of time due to atmospheric turbulence. The model is based on the assumption of a separation of scales between atmospheric turbulence, and mesoscale or synoptic scale atmospheric flow variations. The authors have chosen, however, to align each measurement downwind of their point sources by the peak concentration measured on each transect. The wind direction, therefore, changes with every transect. The mean wind in the Gaussian plume model should be the average wind across the time span of all of the downwind transects (which are conducted within an hour, over which the atmospheric dispersion conditions are assumed to be steady-state), and should not vary every few minutes with each transect. This approach to averaging would create a significantly broader observed plume. It is not clear to me how this might alter the estimated emissions, but this is a misapplication of the plume dispersion model that should be corrected. If the dispersion conditions within the time span of the downwind transects are not steady, this must be explicitly identified and treated in the analyses.**

We thank the reviewer for this comment and the opportunity to better defend and define our work. We understand and agree the definition of 'meandering' which has been used in different ways in the literature has not been well defined for this work and the centering approaches have not been validated. For expediency we break the reviewer's comment into three points:

1) The definition of meandering in the context of this work and whether it is included in Gaussian models. Related to this is the choice to integrate to calculate emissions.
2) The centering of the Gaussian on individual transects to calculate emission rates for the bulk of our data.
3) The centering of the observations to the LES output for the 5 specific cases in this work.

Our responses are provided in this order below.

1) A point of clarification about the Gaussian plume must be made. As an analytical expression of plume diffusion, the Gaussian model will only include plume meandering in its output if the contribution of meandering scales is included in the diffusivity constants employed. This is related to Referee 2's point citing Fritz et al. (2005) discussion of the proper averaging intervals to produce plume cross sections that match the predicted model. They observed timescales ranging from a few minutes to hours meaning the model could over or under predict the cross section observed by the data. This is actually the physical basis for the integration used to calculate emission rates. The scales of 'meandering' as we define them pertain to scales outside the range the LES can include (>100m). However, these are not scales that do not contribute to vertical diffusion. While this would change the Gaussian model, the advantage of the IGM approach is that $\sigma_y$ and $\sigma_z$ are independent :

$$C(x, y, z) = \frac{Q}{2\pi\sigma_y\sigma_z u} \cdot e^{\frac{-y^2}{2\sigma_y^2}} \cdot \left[ e^{\frac{-(z-h)^2}{2\sigma_z^2}} + e^{\frac{-(z+h)^2}{2\sigma_z^2}} \right]$$

Thus, when integrating (i.e. y goes from $-\infty$ to $\infty$) and the transect is perpendicular to wind, the integral method becomes independent of the choice of $\sigma_y$:

$$\int_{-\infty}^{\infty} c(x,y,z)dy = \frac{Q}{\sqrt{2\pi}\,\sigma_z u}\left[e^{\frac{-(z-h)^2}{2\sigma_z^2}} + e^{\frac{-(z+h)^2}{2\sigma_z^2}}\right].$$

Physically, the influence of meandering as it pertains to the accuracy of $\sigma_y$ becomes negligible as long as the transect captures the entire plume. Of course, this is an ideal case; transects are often not perpendicular and occasionally may not cover the entire plume width due the physical constraints of the road. This is, however, arguably a better alternative than, a regression method where the choice of $\sigma_y$ will certainly affect the results. Additionally, the integration method minimizes the effect of the random variation in the plume, which could greatly affect the regression results. This approach has also been used by Albertson et al. (2016). This is also the reason why we have chosen to separate 'meandering' at longer scales from the eddy scales that contribute to the diffusion of the plume and scales that are included in either the Gaussian or the LES. This discussion has been condensed and summarized in Sect. 2.3.

2) We have now discussed the appropriateness of integration in the previous point. We have clarified in the text that we do indeed center the Gaussian to the observed concentration peak for each transect on p. 8 lines 17-20.

*"Second, Eq. 1 is solved for the x and y measurement points of the measured transect using a reference model emission rate ($Q_{ref}$) taken arbitrarily to be 1 kg s$^{-1}$ to produce $C_{model}$. The ratio between the observations and model is used to infer the observed emission rate. Again, the peak observation value is used to define the plume centerline for the Gaussian model for each transect."*

This is done for three reasons: (1) the majority (~90%) of the data processed consisted of 1-3 transects which may not be collected under conditions that match the mean and (2) the true mean wind direction is not known at most sites. For this limited amount of transects, picking an average wind direction is arguably inappropriate. Consider a case where two plumes are observed that deviate from the 'average' wind direction (shown below). In such a case the average Gaussian plume would be calculated with a distance ($x_0$) that is smaller than the observations ($x_1$ and $x_2$). This would underestimate the emissions.

[Figure]

Another way to explore this concept is to consider the average plume of all the individual Gaussian plumes and apply it to the data. An example of this was actually originally shown in Sect. 5.3 (which has now been removed due to apparent redundant discussion to newly added Sect. 4.2). Under ideal conditions, when an additional 'meander' is added to a Gaussian, the average of realizations with different downwind distances that have been re-centered will reproduce the profile of the Gaussian for the average direction. This is shown in newly added Fig. 8:

[Figure]

*"Figure 8. A theoretical site with an additional plume meandering scale added shown in a top-down view at 3 m of (a) Gaussian with no meandering, (b) instantaneous Gaussian with 10° meandering. The comparison of (c) horizontal distributions of concentration and (d) vertical distributions of concentrations for three scenarios (no meandering, averaged meandering and centered meandering) are also shown. All units are arbitrary."*

When not averaged, the profile is broader, but in any case the vertical profile is preserved. The results using the averaged Gaussian profile produced very similar results in Sect. 5.3 (1.09 vs. 1.15 kg hr$^{-1}$). We note here that the Referee commented on Sect. 5.3 in **point 66** saying "**As best I understand this work, the order of averaging that is varied is certain to cause no significant**

**change since the relationship between concentration and source strength is linear.**" This approach actually *is* mathematically identical to using the average profile with all individual results (thus not centering the Gaussian on the observations), but *is not* mathematically identical to averaging the results of averaging each individual transect ratio due to the order of operations:

$$\overline{\left(\frac{\Sigma\, C_{Observation}}{\Sigma\, C_{Model}} \times Q_{ref}\right)} \neq \left(\frac{\Sigma\, \overline{C_{Observation}}}{\Sigma\, \overline{C_{Model}}}\right) \times Q_{ref}$$

We point this out to highlight that the 'best' approach is not obvious and the differences between the approaches would theoretically appear to be small. One of the goals of this work is to understand how this application of the inverse Gaussian method will affect our results with the understanding that a single or a few transects are not representative of the average conditions of the atmosphere. We expect that the error due to these differences in approaches will be encompassed in our uncertainty analysis where we examine how variable the transect-to-transect results actually are. We also concede that there are times when the individual transect centering is not the best approach, such as when information about actual source location or atmospheric conditions is a greater priority to extract from the observations.

The only method to independently verify which approach is best is to compare to a known emission rate. To that end, we have compared the results for three approaches, using the Gaussian centered to the observations, the Gaussian using the averaged profile from all centered profiles during a release period, and a single Gaussian aligned to the average wind direction during a release period. The results, shown in newly added Fig. S3, for each scenario are virtually identical. A total least squared regression for the controlled release data is shown below:

[Figure]

Equally important in the context of this work is the spread of the data. The results using an average profile, or profile in the average wind direction, are slightly more scattered; the centered approach gives estimates with a standard deviation 0-20% lower than the other approaches. Again, most of our data has only a few transects so this is an important consideration. The results when allowing the Gaussian to shift produce lower standard deviations, meaning any individual transect is slightly less likely to over or underestimate the mean obtained by repeat measurements.

We also not that the centering approach is not without precedent. Yacovitch et al. (2015) use the instantaneous peak location in their iterative forward modeling to estimate emissions from unknown sources and specifically state they use the bearing between the source and peak in their emissions estimates, with the reasoning the source location is more certain than the wind direction. Rella et al. (2015) effectively construct an instantaneous plume using a hybrid-Gaussian model where they measure the lowest ~50% of the profile and model the remaining portion using a Gaussian model that best fits their instantaneous measurements. Additionally, Foster-Wittig et al. (2015) presented a method that build upon the EPA OTM33A stationary method for emission estimates. In the EPA method, concentration is plotted against wind direction to produce a Gaussian shape that is then used to calculate an emission. However, this is based upon the assumptions that the plume will be in the center line and observations spread symmetrically about the wind direction. Foster-Wittig et al. (2015) presented a method to plot the data correcting for the fact that this is not always the cased which they termed a 'conditional reconstruction' of the Gaussian plume. This is not entirely analogous, but it underscores that the baseline assumption that an average wind direction can be used indiscriminately is too simplistic for real world applications and some thought must go into correcting for observations that do not match the ideal conditions. They also noted that the differences between approaches were small, errors by other assumptions of the methodology were more important to total uncertainty and repeat observations were the best way to reduce error. This discussion has been added to newly created Sect. 4.2 which includes new figures 8 and S3.

3. The final consideration here is the effect of centering the observations to the LES results, as opposed to the case of centering the Gaussian to the observations previously discussed. Because the observations themselves are integrated the same way whether they are centered or not, this is mathematically identical to not centering the results. Thus this is not expected to affect the comparison. The plume centering that we have chosen to apply to the observations is expected to correct for the large scale plume meandering not simulated in the LES and will not affect the calculated results; it is a way to filter out the meandering impact of the large scales. Filtering observations for the purpose of matching the domain on the LES is also discussed by Agee and Gluhovsky (1999ab) and Horst et al. (2004). Agee and Gluhovsky (1999ab) specifically address the need to remove the influence of larger scales from observations to appropriately compare observations to the LES results. As the LES plume is not aligned to the observations, this is another potential way to compare how the results vary when an average wind direction is used. We see generally that these results compare well, within a factor of two. Given the large uncertainties that have been reported in this method, this is considered a reasonable result. In the application of our work, the lognormal range of emissions means that even if the results

differ by a factor of two a 'large' emission will still be identifiable from a 'normal' emission. This discussion has been added to newly created Sect. 4.2.

We have added additional clarifying information in the text pertaining to the different scales that some of these techniques could be used at to encourage readers to investigate techniques more thoroughly if they are interested in a specific application. Text has been added to p. 3 line 3-9.

*"These techniques have been used in different ways, from direct point source emission estimation to area source emission rate estimation, and they use data that may span a few hours or years. For example, Kort et al. used data over 6 years at 0.33° resolution covering the entire U.S. to estimate a large emission rate. Karion et al. 2013 reported emissions from a large natural gas field (~60 km diameter) using aircraft mass-balance with data collected in a few hours while Caulton et al. 2014 reported well pad emissions (<1 km diameter) using the same technique. It is also feasible that as new instruments and*

*data processing techniques become available, any of these techniques may be used at spatial and timescales not represented by the works cited here."*

Table 1 is intended to illustrate the motivation for this work and not as a comprehensive review of all these methods. While we agree that caution is needed in applying and comparing these methods, many methods have been used at different scales even in the same category making it difficult to add this information.

**13. Page 4, lines 3-12. Are these uncertainties for single emissions estimates from a site using an individual "transect" of data? Please clarify. See my concerns above. Without specifying more about the temporal and spatial domains, these uncertainties are difficult to compare or interpret.**

The text has been clarified on p. 5 lines 12-14.

*"These methods report uncertainty analyses that apply to a single 'site' (i.e. 1 well pad regardless of how many times it was actually sampled), though it should be noted that Rella et al. (2015) and Yacovitch et al. (2015) used downwind transects while Lan used stationary time averaged measurements."*

Authors describe their error estimate in terms of 'sites' which does not distinguish estimates for sites with different density of data collected. We will assume this is meant to apply to sites with single or multiple transects.

**14. Page 4, line 19. What is "an averaging plume for unconstrained instantaneous measurements"? Please clarify. I study boundary layer meteorology but do not understand what you are trying to say.**

The text has been clarified on p. 5 line 28-29.

*"Such situations would include using the average Gaussian plume model with instantaneous measurements without uncertainty or sensitivity analysis or applications over complex topography."*

**15. Page 4, line 25. I don't understand, "this method." Are the authors proposed a new standardized methodology for quantifying emissions? Or is this a study of the uncertainties in plume measurement / source quantification methods?**

The text has been clarified on p. 6 lines 4-7. *"The basic architecture of our approach to quantify errors associated with mobile Gaussian methods uses (1) advanced modeling of a preselected sample site to enable investigation of optimum sampling strategies, (2) application of strategies to the sample collection process and (3) evaluation of additional sources of uncertainty and bias using advanced modeling and a controlled release."*

While we necessarily have to explain our own methodology and uncertainty analysis to support our own studies, we have tried to extend the usefulness of such an endeavor to the broader community by making general recommendations for the best practices of this method. We have added additional descriptions and reworked Fig. 6, combining it with the previous Fig. 7, to clarify out workflow.

**16. Page 5, line 5. The methane span gas concentration is quite low for the measurement of plumes at close range. Are there any tests of the linearity of the instrument at higher concentrations that I assume are encountered driving downwind of strong methane sources?**

The LI-COR 7700 has a stated measurement range accurate up to 40 ppm. The average enhancement that we saw was only ~1.25 ppm above background. Given the stable performance of the LICOR, this calibration standard should not adversely affect the LI-COR in the ambient range of our observations (McDermitt et al. 2010). The text has been modified on p. 6 lines 18-20.

*"The high stated measurement range of the LI-COR 7700 (40 ppm) and the excellent stability of the instrument allow for calibration with a relatively low concentration standard (McDermitt et al. 2010). In addition, enhancements observed were usually less than a few ppm above the ambient concentration."*

**17. Page 5, lines 19-20. "site were screened": Does this mean that sites with trees were eliminated from the sample? The text later states that it is hard to measure a plume at a distance greater than 300m if trees are present. Can you please explain the site selection algorithm more carefully?**

Sites with trees were not ideal and generally not added to our database of possible measurement sites. However, practically this was difficult to achieve. Sites with full tree cover were avoided during this screening process, but sites with a few trees were not as this would have been too strict a threshold. The text has been clarified to discuss site screening on p. 7 lines 4-6.

*"Sites were screened to remove those that were far (>300 m from public road), had obstructions (buildings and full tree lines) and large elevation difference (>50 m)."*

**18. Page 6, line 7-9. Does the NOAA web site provide observations, or numerical model reanalyses? Please clarify. How does this NOAA web site account for the impact of local roughness on atmospheric stability? The hilly, forested, heterogeneous landscape of Pennsylvania will have a significant impact on local dispersion.**

NOAA produced meteorological data from a climate analysis model with data assimilation. The text has been clarifies on p. 7  line 31-p. 8 line 8.

*"Unless measured at intensive sites with a tower, wind speed and stability are taken from NOAA's Ready Archived meteorology ([https://www.ready.noaa.gov](https://www.ready.noaa.gov), Rolph et al., 2017) because mobile wind data showed artefacts after corrections for vehicle heading. These artefacts included unreasonably high wind speed and little correlation to stationary tower measurements. The NOAA Ready archive meteorology dataset is from the National Center for Environmental Prediction's Eta Data Assimilation System model (EDAS, information at [https://ready.arl.noaa.gov/edas40.php](https://ready.arl.noaa.gov/edas40.php), Black, 1994). These climate analysis data are available in 3-hour increments at 40 km resolution and constant pressure coordinates. The data were interpolated to 1 hour resolution for use in the model."*

The NOAA uses a model with an observation network as inputs. More information can be found on their specific website given in the text above. Surface data such as vegetation type and soil are included and they do provide data on surface validation. However, the 40 km scale likely diminishes some aspects of the surface inputs. Note that due to the extensive nature of this study we chose this scale as it was applicable for a larger area of our sample and minimized the number of different inputs for a given day.

**19, Page 6, equation 2. As with equation 1, please make equation 2 part of the sentence.**

The text has been changed accordingly.

**20. Page 6, line 20. This description is insufficient. The Gaussian plume model describes only the concentrations caused by the local source. The measurements do not. A concentration background must be defined and subtracted from the observed time series. This is a critical and non-trivial step. How was this done? I am also concerned that this method does not take into account the correlation between the modeled and observed concentration enhancements. Systematic errors in the dispersion would not be identified by simply matching the integrated downwind enhancement.**

We believe we have addressed background removal in Main Point 1.c. The use of LES to understand the pitfalls of Gaussian dispersion is a major focus of this work. More generally, the discussion of the advantages of integration are addressed in the response to Main Point 2.

**21. Figure 2. This figure is concerning. First, the figure shows concentrations that differ by a factor of 1000, and no explanation is offered. Second, the background that must have been removed from the observations is not discussed. Third, there appears to be a systematic difference in the width of the observed vs. modeled plume. This implies a systematic error in the dispersion coefficient, which will lead to a systematic error in the emissions estimate. How is this addressed? Finally, given that instrument characteristics are expressed in units of ppb, it would be useful to also present downwind concentrations in these units. It would be easy to include both mass density and mole fraction as two different y axis labels.**

Figure 2 has been clarified to explain the model is simulated with a 1 kg/s source in the figure caption and the ppm axis has been added. In addition, the text on p. 8 lines 19-23.

*"The ratio between the observations and model is used to infer the observed emission rate. Again, the peak observation value is used to define the plume centerline for the Gaussian model for each transect. A comparison of observations and model output from 21 downwind transects is shown in Fig. 2. Note that the roads were not necessarily perpendicular to the wind, therefore the superposition of the plume on the roadway may not show a full Gaussian profile."*

We believe we have addressed the question of the background in Main Point 1.c and have also included additional clarification in this section on p. 8 lines 15-17 and Eq. 2.

*"First, the local background ($C_{Background}$), defined as the $CH_4$ minimum over the transect, is subtracted from observations of a plume ($C_{observation}$) to produce an enhancement value. The uncertainty of the background selection in the specific context of this work is discussed in Section 5.1."*

We have not intended to neglect the importance of plume dispersion, but have chosen to not delve into it at this point. We have discussed horizontal and vertical plume dispersion differences between the Gaussian and LES later in Sect. 4.4. We reiterate that the cross-wind integration removes any influence of the lateral dispersion of the plume on the inferred emission rate.

**22. Page 7, line 11. How is steady-state evaluated?**

The text has been clarified and the relevant figures provided in the supplemental have been referenced in the text on p. 10 line 9-11.

*"Sites were simulated for at least 30 min to allow the simulated turbulence to reach a statistically stationary state, where average and standard deviations of wind and scalars approach a constant value as shown in Fig. S2."*

**23. Page 7, line 11. What does it mean to "rescale" the LES output to the measured momentum flux? Isn't the LES set up to simulate these sites? Why is rescaling necessary?**

The numerical simulations of LES are generally non-dimensionalized for simplicity. The text has been clarified on p. 10 lines 11-13.

*"The equations solved in LES are non-dimensionalized using the friction velocity ($u_*$). The advantage is that results from LES apply more broadly to any problem when the non-dimensional quantities, e.g. as $u_{nondimensional}=u/u_*$, are considered."*

**24. Section 3.1 The LESs are truncated in the vertical and thus cannot include largescale turbulence that will influence the sampling statistics. This will be particularly important for unstable conditions. How is this dealt with in designing the sampling strategy?**

We have addressed this in Main Point 3. Additional discussion and references for have been added in Sect. 2.4 to encompass the theory and work that has proven that the important scales of eddies are ~10x the perturbation allowing a smaller boundary layer to be suitable for this analysis. LES was not used to investigate stability as indicated in Table 3, which has been added to the main text.

**25. Section 3.1. The LESs say nothing about sensible heat fluxes. Turbulence statistics change dramatically as a function of stability conditions. What stability conditions are simulated?**

This has been clarified in Table 3. Only neutral conditions are simulated.

**26. Page 7, line 25. Short intervals will sample the same structures in the turbulence. This is well known. Some of the existing literature on sampling statistics in atmospheric turbulence should be cited here.**

This is precisely why multiple transects are needed, and why they need to be spaced by some minimal time to avoid resampling the eddies. Additional citations have been added on p. 10 line 29-p. 11 line 2.

*". . . however, the time interval may also affect results as measurements with short spacing may resample the same coherent plume and thus the same plume realization (Metzger et al., 2007, Shah and Bou-Zeid, 2014)."*

**27. Page 7, line 26. What does "1-N" mean? What is N? I read the description, but I do not understand what this means. What defines the number of samples available in a numerical simulation?**

The text has been clarified to point out the LES time resolution (1 Hz) imposed for the sake of minimizing file size and remove the ambiguity of the value 'N' on p. 11 lines 3-7.

*"Using the LES output, which is saved with 1 Hz resolution to match our instrument sampling frequency, sample transects were picked from the full time series, varying the number of repeat transects and their time intervals. Time intervals of 30s, 1 min, 2 min and random (meaning the time interval was not consistent or constrained) were imposed upon the sample picks and the number of repeat transects ranged from 1 to 70."*

**28. Page 8, lines 8-9. What is the difference between turbulence and plume meandering? Isn't plume meandering caused by atmospheric turbulence?**

See comments in Main Point 2. We have added Sect. 4.2 on p. 13 to specifically address this question and clarify that while meandering has been used to describe plume dispersion more generally, we are specifically using it to refer to larger scale motions. A fundamental difference is that meandering does not increase the cross sectional area of the instantaneous plume since it is effected by scales larger that the plume diameter (scales that simply move and instantaneous plume around). Eddies with sizes similar to the plume diameter diffuse an instantaneous plume.

**29. Page 8, lines 10-12. What does it mean to "reflect the actual variability in the atmosphere"? This is not meaningful. Please define a quantitative tradeoff between transects and flux retrieval.**

The text has been clarified on p. 12 line 2-7.

*"Another way of describing the atmospheric variability as used in this work would be transect to transect variability, which encompasses all random errors that lead to differences between one transect through the plume and the next. These are hence the errors associated with insufficient averaging of the turbulent field and would be reduced as the number of averaged transect increases (Salesky and Chamecki, 2012). These results indicate that in order to sample such that the measurements minimize the effect of these random error, sites must be sampled with at least 10 transects with >1 min spacing."*

The quantitative tradeoff is reflected in Fig. 5, which shows reduced variability in the returned emissions estimates with increasing numbers of passes.

**30. Figure 3 is not very helpful. Please include vegetation and topography.**

Figure 3 has been changed to include satellite imagery.

**31. Figure 4. Images of instantaneous vs. averaged plume structures do not need to be published. This is well known. The manuscript has no references to several decades of simulations of plume dispersion. This is a serious hole in the scholarship of this manuscript. Look up, for example, the publications of Jeffrey Weil, circa 1990.**

Additional references have been added to Sect. 3.1 on p. 10 lines 21-23.

*"LES has been previously used to investigate plume dispersion and is used here as a reference that represents the best estimate for the 'truth' of how a plume evolves in a turbulent near-neutral environment (Nieuwstadt and de Valk, 1987, Weil 1990. Wyngaard and Weil, 1991, Mason 1992, Weil et al. 2004)."*

We feel the figure is instructive for those with limited exposure to LES, thus we have opted to keep the figure in. This manuscript is not intended to be a literature review thus we have limited our discussion to the most relevant papers.

**32. Figure 5. Issues.**

> **1. There is a lot of blank space.**

> **2. The percent difference figures show results from a distribution. What are the elements of the distribution? Is each "pseudo-transect" an element of the distribution? Please define this clearly in the caption.**

**3. A standard deviation is a description of a population. Are there multiple populations whose statistics are being compared? Please clarify.**

**4. What does "random seconds" mean? Why is this useful?**

**5. There is little additional information given in the 30/60/120 second sample repeat figures, and the differences are difficult to see. It would be more effective to show one of these, then have one additional figure showing the differences caused by changing the sampling interval.**

Figure 5 has been condensed for clarity and the original figures have been moved to the supplemental as Figure S4. The text has been clarified on p. 11 lines 9-16.

*"For each combination of transect number and time spacing, 100 random samples were picked and the mean and standard deviation of the emission rate were calculated and compared to the known LES emission rate. An ideal scenario would result in a low percent difference and a standard deviation of the sample that is roughly equal to the standard deviation of a fully random sample (random time spacing). The random time spacing should be representative of a fully random sample as points are drawn from the full 30 min time simulation and are less likely to resample similar plume structures. Standard deviations are being compared instead of standard error as each sample strategy is being treated as a population so that the resulting standard deviation may be used as an approximation of the population standard deviation. Box plot of the results for the 100 random samples are shown in Fig. 5."*

**33. Page 8, lines 18-19. The original plan is not relevant at this point. Present the number of samples obtained.**

The text has been adjusted on p. 12 lines 13-16 to:

*"As depicted in Figure 6, the goal of the sampling strategy was to produce more standard sampling sites, with fewer replicate sampling sites and even fewer intensive sampling sites. This was based upon the approximate amount of time to acquire each sample and the limited amount of time to collect samples overall."*

The actual sample numbers were already available on p. 12 lines 17-18.

**34. Page 8, line 24. What is the "average maximum percent difference"? Maximum among what? Average across what? And why is this relevant? Please also explain the populations used to define the standard deviations.**

The text has been clarified on p. 12 line 21-26.

*"For the population of standard sampling sites with multiple passes, the average rsd of repeat passes was 67% and the average maximum percent difference between emission estimates at a single site was 58%. The average rsd of the population of 53 emissions estimates for the replicate sampling sites was 77% and the average maximum percent difference (highest observational deviation from the mean of repeat measurements) was 150%. The rsd ranged from 12% to 260%. These populations offer insight into how sampling strategy may change estimates of these statistics and offer the chance to compare real results to the LES results shown in Sect. 3.1."*

**35. Page 8, last sentence. What is the population being discussed here? Why are these numbers important?**

The importance of the LES derived values, which we are comparing to, has been clarified in Sect. 3.1. For each combination of transect number and time spacing, 100 random samples were picked and the mean and standard deviation of the emission rate was calculated. An ideal scenario would result in a low percent difference and a standard deviation of the sample that is roughly equal to the standard deviation of a fully random sample (random time spacing). The random time spacing should be representative of a fully random sample as points are drawn from the full time simulation and are not likely to resample similar plume structures. The text has been clarified with additional discussion on p. 12 line 21-29.

*"For the population of standard sampling sites with multiple passes, the average rsd of repeat passes was 67% and the average maximum percent difference between emission estimates at a single site was 58%. The average rsd of the population of 53 emissions estimates for the replicate sampling sites was 77% and the average maximum percent difference (highest observational deviation from the mean of repeat measurements) was 150%. The rsd ranged from 12% to 260%. These populations offer insight into how sampling strategy may change estimates of these statistics and offer the chance to compare real results to the LES results shown in Sect. 3.1. These results are consistent with the LES results shown in Sect. 3.1 predicting small numbers of transects will yield an artificially low rsd and more transects are needed to produce an accurate measure of variability. Additionally, the lower maximum percent difference for standard sampling is consistent with the Sect. 3.1 LES results showing few transects will sample more similar plume structures."*

**36. Figure 7 does not explain how the number of sources will be determined. I also disagree with the far right-hand bar. A more complex simulation does not ensure less uncertainty, and "model uncertainty" is not defined. If there is a precise definition of uncertainty that can be shown to be reduced with the LES, please explain and define this.**

Figure 7 has been combined with Figure 6 and redone for more clarity in what is now Figure 6. The combined figure shows the full workflow including that sites with more measurements were also used with the higher complexity model. The uncertainty column is meant to be representational of the expected reduced uncertainty due to more available measured input information, not necessarily the model complexity. Clarifying discussion has been added to p. 13 lines 15-16.

*"Generally, the more information available (e.g. source location), the less uncertain the results are likely to be. Results will be compared from different scenarios to address to what extent uncertainty can actually be reduced."*

The source locations are discussed in the text as it would be difficult to succinctly portray in a figure. See comments to point 41.

**37. Page 9, line 17. What is a "warm up time"? (I see here that the determination of steady state is explained.)**

The text has been clarified on p. 13 line 23.

*"LES time series spanning ~30 minutes were averaged to produce a pseudo Gaussian distribution excluding a ~5 min initialization period (time until stationary state is achieved)."*

**38. Page 9, line 21-24, and figure 8. This discussion and figure displays a fundamental problem with the LES setup, and the authors' interpretation of these data. The "real world" meandering of the wind that is being described is turbulence in the atmospheric boundary layer. A Gaussian plume model describes the average state downwind of a point source, which would be properly represented by the average of all of the transects, without "aligning" them to match the peak concentration on each transect. Aligning these is the equivalent of performing some sort of high-pass filter.**

See comments in the response to Main Point 2.

**39. Similarly, I am now concerned about the processing of all of the transect data. In Figure 2a, for example, have all of the transects been arranged so that the x direction changes from transect to transect? If so, the same odd filtering of turbulence has been applied. This is not appropriate for comparison to a Gaussian plume model. I expect this could explain the difference in plume widths that is evident in Figure 2 (note the modeled plume is wider that the observed plumes that may have been filtered to remove large-scale turbulence).**

See comments in the response to Main Point 2.

**40. Please explain the theoretical basis for equation (3)?**

Equations 3 and 4, which dealt with the proper dimensionalization of the LES non-dimensional output to calculate and emission rate have been combined and explained on p. 13 line 25 – p. 14 line 8 to improve clarity.

*"Because all LES runs on non-dimensionalized, the LES output needs to be scales to represent the actual field conditions for a given observation. The LES non-dimensional and dimensionalized (denoted with a superscript D) outputs can be related by Eq. 3, which can be inferred from the stationary advection diffusion equation or by analogy to the scaling of the Gaussian model in Eq. 1:*

$$\frac{C_{LES}}{\dot{M}_{LES}} \times u_{LES} = \frac{C^D_{LES}}{\dot{M}^D_{LES}} \times u^D_{LES} \tag{3}$$

*Rearranging Eq.3, the LES non-dimensional scalar output ($C_{LES}$) was dimensionalized ($C^D_{LES}$) to units of kg m$^{-3}$ according to Eq. 4:*

$$C^D_{LES} = \frac{\dot{M}^D_{LES}}{\dot{M}_{LES}} \times \frac{u_{LES}}{u^D_{LES}} \times C_{LES} \tag{4}$$

*Here, $\dot{M}^D_{LES}$ is the normalized mass (1 kg s$^{-1}$) and $\dot{M}_{LES}$ is the actual non-dimensional emission rate introduced into the LES system. Normalizing by $\dot{M}_{LES}$ is necessary to match the reference emission rate ($Q_{ref}$) also used in the Gaussian retrievals. The non-dimensional wind speed ($u_{LES}$) is divided by the dimensionalized ($u^D_{LES}$) wind speed which is set equal to the on-site tower measurements."*

**41. Section 4.1. How do you distinguish among single vs. multiple sources? This is not explained.**

The text has been clarified to explain how sources were identified on p. 13 lines 16-19.

*"As on-site access was not available, all large structures were treated as sources. These were visually identified and exact coordinates were confirmed in Google Earth. The identified sources were always gas processing units or storage tanks."*

**42. Page 10, line 2. The two Gaussian retrievals "compare"? What does this mean?**

The text has been clarified on p. 16 lines 15-19.

*"Figure 9a shows the emission rates using NOAA winds while Fig. 9b shows the emission rates calculated with measured wind. The two Gaussian retrievals are shown to compare using NOAA winds, which is the base scenario for all sites, and using the in-situ measured wind. Since the controlled release used one release point, there is only one source at a known height and the SS Gaussian approach is used. The agreement increases greatly when using the in-situ wind data; NOAA overestimated the winds during the latter two release rates."*

**43. Figure 9. Once again, please explain the populations of data that go into these box and whisker plots. Is each point a transect? If so, how many transects make up each population?**

The populations of Fig. 9 are now defined on p. 16 lines 14-15.

*"Boxplots of the populations of emission retrievals from three scenarios are shown in Fig. 9 . . ."*

We have moved supplementary Table S1, which contains site specific details, to the main text as Table 3 and referenced it in the text on p. 16 lines 12-13.

*"Site set-up was discussed in Sect. 2.2 and specific site details are available in Table 3."*

Specific numbers of transects for each release have been added to Table 4.

**44. Page 10, line 13-14. I do not agree that the results show no apparent bias. Figure 9a has winds that are known to be too high, according to the text. Figure 9b shows a systematic underestimate of emissions. Figure 9c shows varied results.**

The text has been clarified on p. 16 line 27-30.

*"In general, the close agreement across a range of release rates shows no apparent bias in the Gaussian with tower winds results, with the results scattered low and high relative to the release rate and no results significantly different (at the 95% CI) from the release rate."*

As shown in Table 4, panel b does not consistently underestimate the emissions. The issues concerning panel c are discussed in the text on p. 16 line 30-p. 17 line 3.

*"While LES can readily account for unstable or stable conditions, this would come at an increased computational cost as multiple simulations would be needed. The results here show that this in fact may not be necessary, at least over flat homogeneous terrain, as the Gaussian model provides a comparable performance at a very small fraction of the modeling effort."*

**45. Figure 10. What is the level of uncertainty in the release rate? If this release is being used to evaluate the methodology, then the release rate must be carefully calibrated. How has this been done?**

The release rate uncertainty has been added to the text on p. 16 line 11-12 and Table 4.

*"Release rates were controlled by two MKS mass flow controllers with stated flow accuracy of 1% of the set point (Model GE250A) or 0.05 SLM (Model 1179C)."*

The flow controllers used were factory calibrated and tested in the lab before use.

**46. Page 10, line 22-23. Only one outlier with zero emission is shown in Figure 10.**

In release 1, 0 is not considered an outlier given the spread of the data. The text has been updated on p. 17 lines 11-12.

*"During release 1 and 3, even though a constant source is being emitted, a few (1-3) transects showed no observed plume whatsoever."*

**47. Table 2. Please report the observed and simulated winds, and the observed and simulated stability conditions. Please include more descriptive information in the table caption. In particular, please note how many transects were collected at each site, the time span of the transects, and the source of the uncertainty values given in the table. If these are standard deviations based on emissions derived from individual transects, and the purpose is to compare mean values, the authors should report the standard deviation of the mean, not the standard deviation.**

We have moved supplementary Table S1 to the main text as Table 3 (which contains time intervals, transect # etc.) and referenced it in the text to mitigate confusion. What is now Table 4 has been updated with the standard deviation of the mean as well as transect numbers.

**48. Table 3. Please include more descriptive information in the table caption. In particular, please note how many transects were collected at each site, the time span of the transects, and the source of the uncertainty values given in the table. If these are standard deviations based on emissions derived from individual transects, and the purpose is to compare mean values, the authors should report the standard deviation of the mean, not the standard deviation. Also, please report winds and stability. And were the winds observed, or taken from NOAA reanalyses?**

What is now Table 5 (formerly Table 3) has been updated with standard deviation of the mean and the number of transects. In addition Table 3 has now been moved to the main text with site specific information. Table 2 has also been added to more generally explain data available for each level of data.

**49. Page 10, lines 27-28. See my earlier concerns about centering of the plumes on each transect. I believe this is an erroneous interpretation of plume dispersion.**

See comments in Main Point 2.

**50. Page 11, lines 7-8. The overall differences are small? The flux estimates in Table 3 differ by as much as a factor of two. This seems large to me.**

The text has been clarified on p.18 lines 1-2.

*"In the range of distances investigated in this study (<200 m) the overall discrepancy between the different model outputs is, however, small."*

We would also clarify that while this may seem large, emissions may span many orders of magnitude so this level of comparison is actually quite good for this method.

**51. Figure 11. This figure points to many questions and problems.**

1. **What are the equations for the vertical and horizontal flux profiles in the Gaussian plume model?**
   An appendix with the pertinent equations has been added on p. 25-26.
2. **What is the position in the domain where the profiles (figure d-g) are computed?**
   The figure caption has been updated to indicate that these are computed for the road plane (as shown in Fig. 3 with distances specified in Table 3).
3. **What is the distance downwind for everything shown in this figure?**
   The figure caption has been updated to indicate that these are computed for the road plane (as shown in Fig. 3 with distances specified in Table 3).
4. **How are the multiple sources chosen? As best I can tell this is never described anywhere in the manuscript.**
   See comments in point 1b.
5. **Does the LES flux include subgrid and resolved fluxes? Please explain, and delineate these two sources so that it is clear what fluxes are explicitly resolved.**
   Only resolved fluxes are shown in these figures. Subgrid scale (SGS) contributions are important for higher order statistics, but when dealing with the mean concentrations they are not directly relevant. That is, for the mean concentrations used to infer emissions, the SGS contribution is theoretically zero. For this figure, the horizontal flux is a mean advective flux and there is no SGS contribution, but for the vertical flus an SGS contribution exists and is modeled in the LES, but only the resolved flux is plotted here. The text has been clarified in the Fig. 11 caption.
   *"The vertical LES flux corresponds to the resolved fluxes only."*
6. **What does z=3 mean? 3 what? The same comments and consternation hold for the similar figures in the supplementary materials.**
   Z is the vertical altitude as consistent with panels d and e. The title has been clarified to add the 'm' unit.

**52. Page 11, lines 12-14, sentence starting with "regardless." I do not understand what you are trying to say.**

We have changed the text on p. 18 lines 5-12 to:

*"For this analysis, we have assumed the source to be constant during the time span of the measurements (typically less than 1 hour). This may not be true for all sites and may be a driver of variability, for instance tanks are known to emit sporadically and Goetz et al. (2015) have shown emissions varying over the course of a few hours. However, it is not clear that there is a need to quantify emission variability at scales less than 1 hour for most sources as there is a practical limit to the time resolution that can be included in inventory estimates, for instance. We thus expect any changes in emission rate at <1 hour to be reflected in what we have termed atmospheric variability, or transect to transect variability."*

**53. Table 4. What is the distance from source of the measurements? Are "emissions" those estimated from transect measurements? How many transects? What are the meteorological conditions for these measurements?**

The supplemental table with this information (number of transects, distance, meteorology etc.) has been moved to the main text as Table 3. Table 4 has been removed as it was intended to be an example and not a systematic analysis as described later in the text.

**54. The text in the first paragraph of section 5.1 does not appear to match the results in Table 4. The text notes that unless the change in source location in the x direction is similar to the distance of the measurement downwind, that the impact of the estimated emission is small. Table 4 shows two examples, one of which has a 150% change in source strength estimate, and another a 7.5% change in source strength estimate, and as best I can tell (Table S1?) the measurements are both from about 150m downwind of the source, with the site with a larger % change having measurements farther downwind. Based on the results presented by the authors, as best I can interpret them, I disagree with their conclusions.**

Table 4 is a simple exploration of 2 samples sites where as Figures S11 and S12 actually systematically look at the effect of a specified uncertainty with changing downwind distance and height for different downwind measurement distance. Table 4 has been removed to reduce confusion and streamline the discussion.

**55. Figure S10 (b) is uninterpretable. What is the "Ratio between the sum in y and distance x of the scenarios"? Is this truly a ratio of distances that is being plotted? Please clarify.**

What is now Figure S12 has been redone to present data as a percent reduction from the original model to improve clarity. The percent change in the model is equivalent to the change in the calculated emission rate for these scenarios.

**56. Figure S11 (b). See comment above regarding Figure S10.**

What is now Figure S13 has been redone to present data as a percent reduction from the original model to improve clarity. The percent change in the model is equivalent to the change in the calculated emission rate for these scenarios.

**57. Page 11. "NOAA wind speeds differed from the tower data on average by 50%." This is uninterpretable. Please rewrite this to be meaningful.**

The text has been clarified on p.19 lines 6-7.

*"In the context of this analysis, we compared the NOAA wind to the mean on-site tower measurements of winds at 16 tower sites. NOAA wind speeds reported higher and lower values than the mean tower winds; the absolute difference was 50% on average."*

**58. Figure S12. Please define this ratio, as with Figures S10 and S11.**

Figure S14 has been redone to present data as a percent reduction from the original model to improve clarity. The percent change in the model is equivalent to the change in the calculated emission rate for these scenarios.

**59. Page 12, lines 7-8, "The magnitude of the difference between consecutive stability classes is relatively consistent, averaging 40%." Figure S12 does not suggest simply defined average, as the signs and magnitudes of the differences change as a function of distance downwind. Please explain what this 40% value means.**

Figure S14 has been replotted to clarify this point. The text has been clarified on p. 19 lines 29-31.

*"The magnitude of the difference between +/-1 stability class is relatively consistent at farther downwind distance, averaging a change of 40% at 200 m downwind."*

**60. Page 12, lines 12. Please define "absolute terrain slope."**

The text has been updated on p. 20 lines 2-3.

*"The geometric mean of the absolute terrain slope, defined as the absolute value of terrain rise over the distance between sources and observation . . ."*

**61. Page 12, lines 24-28. The methods for quantifying the uncertainties should be explained carefully in the methods. This rapid-fire, qualitative overview of the methods behind Table 5, arguably the most important result of the manuscript, is insufficient.**

The details of the uncertainty work are described throughout the manuscript. Sect. 5.1 has been expanded to clarify many points including the separate uncertainties for source location and height. Discussion of the Monte Carlo approach are also expanded in this section.

**62. Page 12, lines 30-34. See previous concerns about the lack of larger scale turbulent motions in the LES used for this study. I agree in general that the LES used here should provide a lower estimate of turbulent sampling error, but it also will not contain the full spectrum of atmospheric turbulence and true turbulent sampling error. This should be explained, and references to the rich literature of the full spectrum of atmospheric turbulence and the limitations of LES that is limited to the atmospheric surface layer should be included in the manuscript.**

See comments in Main Point 3. It is true that LES will not resolve the entire spectrum of atmospheric motions, but as we explain, the motions important for turbulent diffusion are resolved. In addition we have added the following text and references to p. 20 line 29-p. 21 line 3:

*"LES has a limited ability to represent the very-large scale motions (Kunkel and Marusic, 2006) or some eddy features (Glendending, 1996) due to its limited horizontal domain size and idealized forcing (e.g. de Roode et al, 2004, Agee and Gluhovsky, 1999ab). In addition, the limitation of LES in the surface layer due to applications of the Monin-Obukhov similarity theory is also one of the concerns (e.g. Khanna and Brasseur, 1997). We do not intend to represent all sources of uncertainties pertaining to the atmospheric conditions. Instead, the focus of the LES is to resolve the range of scales that are critical for the turbulent diffusion of the plume, which is often represented as a single eddy diffusivity coefficient in the Gaussian plume models. "*

**63. Page 13, line 6. The ranges of the input values used in the Monte Carlo simulation and their distributions must be described, or these results are meaningless.**

Table 7 has been added with inputs and additional text has been added on p. 21 lines 9-20.

*"To combine the remaining sources of error, a Monte Carlo simulation of errors through the Gaussian equation was performed. Inputs are available in Table 7 and an example output if Fig. S15. This approach was determined to be the most appropriate as uncertainties may not be normally distributed and emissions are constrained to be above zero, causing a skew to the emission retrievals. A generic scenario matching average conditions experienced during the measurements was devised using a downwind distance of 200m, 1.5 m s$^{-1}$ wind speed, neutral stability and 260 ppb observation enhancement. The*

*specifics with regards to distributions assumed for each uncertainty parameter are shown in Table 7 for the SS Gaussian and MS Gaussian approaches with 1 and 10 transects, as well as a theoretical lower limit scenario. For each scenario, 1000 randomly generated samples of $C_{Observation}$, $C_{Background}$, and $C_{Model}$ were obtained and used according to Eq. 2 to obtain a distribution of Q samples. The Q samples were then used to estimate the 95% confidence interval. The combined effects produce a skewed distribution of emission rates as shown in Fig. S15."*

**64. Table 5. See prior concerns about the lack of documentation of the methods for assessing these uncertainties. In addition, how is the "total" uncertainty assessed? Is this the result of the Monte Carlo simulation described in the text? If so, see prior concerns about documentation of this experiment. If not, please explain how this total is computed.**

See comments to point 63 to address Monte Carlo documentation. We have removed the total uncertainty line and added Table 7 to display the 95% CI for the scenarios.

**65. Figure 12 results vary by a factor of 1000 with no explanation.**

The caption for Fig. 12 has been updated to explain the scale difference.

**66. Section 5.3. As best I understand this work, the order of averaging that is varied is certain to cause no significant change since the relationship between concentration and source strength is linear. Thus this comparison is not informative or significant. If anything is nonlinear that could cause a difference, please explain.**

Due to the expanded discussion in Sect 4.3 we believe Sect 5.3 is now repetitive and have opted to remove it to streamline the manuscript.

**67. Page 14, lines 14-15. "Likewise, the single transect method, though fast, has many sources of uncertainty. Hence, the combination of all of these approaches described in Sect. 3 is expected to maximize sampling efficiency while minimizing uncertainty." I do not see how your results justify this statement.**

By explaining the Monte Carlo approach we have shown that there is value in using any and all of these approaches (single transect, multiple transect and LES) to reduce uncertainty. We have clarified the text on p. 22 lines 27-30 to:

*"Hence, the strategic combination of all of these approaches described in Sect. 3 is expected to maximize sampling efficiency while minimizing uncertainty. Less frequent higher intensity measurements (from on-site meteorological data and multiple transects) can be used to provide a better estimate of uncertainty for single transect approaches."*

**68. Page 14, lines 22-23. I do not yet agree with the assessment that no bias is observable. This cannot be assessed until the uncertainty bounds in Table 2 are clarified. If the mean values differ significantly, then there is observable bias.**

See comments in point 44. We have also clarified the text on p. 23 lines 7-10.

*"Using the LES and a controlled release, we confirm that the Gaussian model performs well when in-situ winds are available. The NOAA estimated winds can be a source of error, but we did not observe a*

*systematic difference between the NOAA and in-situ winds, thus no sources of bias using our approach are expected on average."*

**69. Page 14, line 23,"LES is therefore not required for studies where source strength calculation is the main goal." The authors have clearly avoided complex terrain where LES is most likely to be needed. This broad and general statement is not justified by the research presented in this manuscript.**

The text has been amended on p. 23 lines 10-11.

*"LES is therefore not required for studies where source strength calculation is the main goal and other complicating factors such as complex topography are not present."*

**70. Page 14, lines 24-25. "From this we use Monte Carlo analysis to extrapolate that the 95% confidence interval for sites with standard sampling (n=2) ranges from 0.05x–6.0x where x is the emission rate." This is a potentially important result, but this is offered as a passing comment. The methods behind this calculation are not presented. This is not acceptable for publication. This is an important result whose methods must be clearly explained and defended.**

Table 7 has been added with inputs and additional text has been added on p. 21 lines 9-20.

*"To combine the remaining sources of error, a Monte Carlo simulation of errors through the Gaussian equation was performed. Inputs are available in Table 7 and an example output if Fig. S15. This approach was determined to be the most appropriate as uncertainties may not be normally distributed and emissions are constrained to be above zero, causing a skew to the emission retrievals. A generic scenario matching average conditions experienced during the measurements was devised using a downwind distance of 200m, 1.5 m s$^{-1}$ wind speed, neutral stability and 260 ppb observation enhancement. The specifics with regards to distributions assumed for each uncertainty parameter are shown in Table 7 for the SS Gaussian and MS Gaussian approaches with 1 and 10 transects, as well as a theoretical lower limit scenario. For each scenario, 1000 randomly generated samples of $C_{Observation}$, $C_{Background}$, and $C_{Model}$ were obtained and used according to Eq. 2 to obtain a distribution of Q samples. The Q samples were then used to estimate the 95% confidence interval. The combined effects produce a skewed distribution of emission rates as shown in Fig. S15."*

**71. Page 15, lines 3-4. Rella et al, (2015) did not use a Gaussian plume dispersion approach, and attempted to measure the entire vertical extent of the plume. This manuscript notes that vertical dispersion is a major source of uncertainty in their results. It seems very likely that Rella et al's approach should, therefore, yield a significantly smaller uncertainty estimate than a method that relies on a Gaussian plume dispersion model to quantify vertical dispersion.**

The text has been clarified to include discussion of their vertical plume information. p. 23 lines 24-27.

*"In addition, Rella et al. (2015) incorporated vertical information to inform their Gaussian plume model creating a mass balance Gaussian hybrid, though only a small vertical profile was available (4 measurement points up to 4 m).  As the vertical flux was seen to be a potential source of error in this work, measurements of this metric may reduce uncertainty."*

Though Rella et al. do have vertical information it is not enough to resolve the entire plume, thus they still rely on a Gaussian plume equation to calculate emissions.

**72. Page 15, lines 5-6. "As described in Sect. 3.2, the observed atmospheric variability can range from 10-200% meaning that in-situ observations of variability and post screening out conditions with unacceptably high variability may be a viable way to reduce uncertainty." What is the quantity "atmospheric variability," that can range from 10-200%, and 10-200% of what? What criteria of "unacceptably high variability" can be used to screen out data and thus reduce uncertainty? This reasoning is either explained poorly or imprecise, and the conclusions stated are thus either uninterpretable or inaccurate. This statement is not suitable for publication.**

The definition of atmospheric variability has now been clarified as discussed in Main Point 1a. The text has been clarified on p. 23 lines 28-p. 24 line 3.

*"Atmospheric variability (random error) was the single largest driver of uncertainty in the Monte Carlo simulations (see for example Fig. S15) because the model can be improved with better information and wind measurement. To reduce the uncertainty from atmospheric variability more observations are needed; however, sometimes this is impractical. In-situ observations of variability from repeat measurements at a single site may potentially be used as a post screening method to exclude conditions that will lead to extremely high uncertainty in single transect sites."*

**73. Page 15, recommendations. Point 1 cannot be satisfied in many cases. What happens when measurements are needed for a location that does not satisfy these criteria? I don't understand point 5.**

We have clarified point 5 on p. 24 to:

*"For experiments where sampling frequency is at a premium, at least 1 site per sampling outing should be repeated with ≥ 10 sampling transects to reliably constrain atmospheric variability which is expected to be the largest source of the uncertainty estimate."*

We believe we have presented sufficient evidence in defense of point 5 now with the expanded discussion of the Monte Carlo approach and how uncertainty ranges change for multiple and single transects. The Referee's comment on Point 1 is valid, that is why we make a point to clarify this is best practice and necessary to compare to our results.

**74. Point 6 is not a significant recommendation resulting from this research.**

In this manuscript we looked at uncertainty from stability, source location, source height, wind speed, dispersion model and number of transects. Previous studies have published work that rarely accounted for any of these. Therefore we feel it is always useful to reiterate the importance of planning for uncertainty analysis and have opted to leave this point in as it remains a valid, and forgotten, recommendation.

**75. I do not understand "the strategy" proposed in Point 8. It is not clear how this collection of measurements is proposed as an integrated sampling strategy.**

We have attempted to clarify this throughout the text and with Fig. 6. which shows more succinctly how the workflow is divided for data of different time intensity. We emphasize again that multiple transects are needed to quantify atmospheric variability, which is the single largest source of uncertainty. Routinely measuring the state of the random error imposed by the atmosphere will allow better uncertainty analysis for sites with fewer transects.

**76. Section 8. Data should be publicly available, and not restricted to access only via correspondence with the authors.**

As this dataset is expected to be reference by multiple works, a data file containing the emission rates, date/time, LOD, uncertainty estimate, meteorology, site locations and traits including spud date, operator, production and status will be submitted to DataSpace at Princeton University (https://dataspace.princeton.edu/jspui/).  This archive is free and open to the public and will provide a stable url where the data can be accessed. The URL location of the dataset is now explicitly mentioned in the supporting information. Note that the University understandably makes it very difficult to modify datasets one posted, and for this reason, we have held off on publishing the data until the review process is finished.

[revised manuscript text omitted]

25   ~~estimate. We have chosen to average the emissions deduced from individual transects. Alternatively, the multiple transects themselves could be averaged to produce a single emission estimate. The latter method should theoretically be more Gaussian in shape and more comparable to the model, but requires enough transects to produce a Gaussian profile and may not be appropriate for sites with a limited number of transects. However, we analyzed a representative site as a comparison to provide information~~
30   ~~as to what, if any, difference in retrieved emissions may be expected with this method. As an example, Site 3 was chosen to average the multiple transects previously shown in Fig. 2. The averaged plume, shown in Fig. 12, was then used to calculate the emission rate using the IGM approach, also using the model averaged across all transects. The averaged transect emission rate is 1.2 kg hr$^{-1}$, extremely close~~

~~Averaging multiple transects has the benefit of reducing the influence of atmospheric variability on the uncertainty of the measurement; however, longer measurement time is still required due to the need for many transects. Each approach, averaging emissions vs. averaging transects, produces similar final results indicating that averaging method is not a driver of emission uncertainty. Either approach may be acceptable given the constraints and intent of a sampling session. The variability of single transects of emissions may be a useful tool for data quality control while averaged transects may be useful in additional analysis intent on pinpointing the location of an unknown source.~~

[revised manuscript text omitted]

a   Single Source Gaussian

b   Multi Source Gaussian

c   Neutral LES

d   Vertical Conc. Dist.

e   Vertical Flux Dist.

f   Horizontal Conc. Dist. (Z=3)

g   Horizontal Flux Dist. (Z=3)

a

**Controlled Release 1**

b

**Controlled Release 2**

c

**Controlled Release 3**

**Figure S3. A comparison of release experiments 1-3 (a-c) results using the Gaussian aligned to the observation peaks (Centered) and the average Gaussian (Uncentered Avg. Plume) and the single Gaussian in the average wind direction (Uncentered Avg. Dir.).**

[Figure]

**Figure S4. The rsd (a,c,e,g) of the emission retrieval using and the percent difference (b,d,f,h) between emission retrieval and known simulation emission rate using various amounts of transects and 30 second (a,b), 1 minute (c,d), 2 minute (e,f) or random (g,h) transect spacing. Box and whiskers plots show the 50th percentile (red), 25th and 75th percentile (blue) and 2.5 and 97.5 percentile (black). The recommended 10 transect criteria is shown in green.**

**Site 1**

[Figure]

**Figure S5.** Comparison of three scenarios for Site 1 showing images of (a) single Gaussian, (b) multi-source Gaussian and (c) averaged LES. The comparison of vertical distributions (d) concentrations and (e) fluxes, and horizontal distributions of (f) concentrations and (g) fluxes are also shown.

[Figure]

**Figure S4S6.** Comparison of three scenarios for Site 2 showing images of (a) single Gaussian, (b) multi-source Gaussian and (c) averaged LES. The comparison of vertical distributions of (d) concentrations and (e) fluxes, and horizontal distributions of (f) concentrations and (g) fluxes are also shown.

[Figure]

**Figure S5S7.** Comparison of three scenarios for Site 4 showing images of (a) single Gaussian, (b) multi-source Gaussian and (c) averaged LES. The comparison of vertical distributions of (d) concentrations and (e) fluxes, and horizontal distributions of (f) concentrations and (g) fluxes are also shown.

[Figure]

**Figure S8.** Comparison of scenarios using different diffusion models for Site 1 showing images of the comparison of vertical distributions of (a) concentration, (b) flux and of the horizontal distributions of (c) concentration and (d) flux.

[Figure]

**Figure S9.** Comparison of scenarios using different diffusion models for Site 2 showing images of the comparison of vertical distributions of (a) concentration, (b) flux and of the horizontal distributions of (c) concentration and (d) flux.

[Figure]

5  **Figure S10. Comparison of scenarios using different diffusion models for Site 3 showing images of the comparison of vertical distributions of (a) concentration, (b) flux and of the horizontal distributions of (c) concentration and (d) flux.**

[Figure]

**Figure S9S11.** Comparison of scenarios using different diffusion models for Site 4 showing images of the comparison of vertical distributions of (a) concentration, (b) flux and of the horizontal distributions of (c) concentration and (d) flux.

[Figure]

[Figure]

Figure S12. (a)  Modeled CH₄ at three source locations at different __downwind__ x positions assuming 3 m receptor height, 1 m source height, neutral stability and 1.5 m s⁻¹ wind speed. (b) Change in % of the modeled CH₄ in each of the scenarios and base scenario (S=0). __(c) The same as panel (a), normalized by the perturbation distance of 5 0m. (d) The same as panel (b), normalized by the perturbation distance of 50 m.__

[Figure]

[Figure]

**Figure S13.** (a) Modeled $CH_4$ using 8 source heights assuming 3 m receptor height, neutral stability, and 1.5 m s$^{-1}$ wind speed. (b) Change in % between the modeled $CH_4$ for each of the scenarios and the base scenario (h=1 m).

[Figure]

[Figure]

Figure S14. **Change in percent** between  modeled CH₄ **between** +/- one stability class **and base scenario** for (a) Class B, (b) Class C, (c) Class D, and (d) Class E assuming 3 m receptor height, 1 m source height, and 1.5 m s⁻¹ wind speed.

[Figure]

**Figure S15. Monte Carlo simulations using 1,000 replicates for the standard sampling, 1 transect case showing (a) normalized observation distribution, (b) normalized model distribution and (c) normalized emission rate distribution.**

---

## Author Comment (AC2) · 3 May 2018

**Responses to Anonymous Referee #2**

We thank the reviewer for their helpful comments which we hope will help make this work more comprehensive and easier to understand. All comments are addressed below and requested changes in text, figures, captions or tables have been included. The reviewer's comments are in bold with our specific responses below in regular text and quoted sections from the manuscript in italics.

**Anonymous Referee #2 Comments**

**I recommend including uncertainty bounds for SS gaussian, MS gaussian and MS LES in Table 1, as well as in the abstract.**

The abstract has been modified to include uncertainty ranges on p.2 line 2-3.

*"The uncertainty bounds calculated for this work were 0.05q-6.5q (where q is the emission rate) for single transect sites and 0.5q-2.7q for sites with 10+ transects."*

We have chosen to leave these out of Table 1 as the table shows the current state-of-the-art and our methods have yet to be explained at this point in the manuscript.

**Main points:**

**1. Timescale**

**p. 3, Section 1.1: I think this section warrants a discussion of timescales of Gaussian simulations. There is a good deal of variation in what the atmospheric modeling community determines to be the appropriate timescale for the A-D stability classes. The consensus seems to be on the order of 10-15 minutes. See Fritz et al. DOI:10.13031/2013.18501 A discussion of timescales is also relevant to Reviewer #1's second comment about averaging of plume transects vs the meandering plume.**

The suggested discussion has been added including the reference mentioned on p. 4 lines 20-25.

*"The comparison of instantaneous and modeled concentrations is thus impacted by the averaging timescale associated with the measurements. Studies suggest that appropriate time scales depend on downwind distance and stability ranging from 2-60 min (Fritz et al. 2005). For example, in class D at 200 m downwind, 3 min would be sufficient when using the Gifford dispersion coefficients. Insufficient averaging would result in random errors that we refer to in this work as the uncertainty related to 'atmospheric variability.'"*

**2. Winds**

**p. 6, line 7: Explain why the choice was made to use downloaded interpolated hourly winds (3-hour interval data, originally) when there were presumably measured winds on-site for the intensive IGM sites? I am also concerned in how well this interpolated 1-hour met will work for determining stability parameters of quick transects. See also previous comment on timescale. How would results differ with measured met? The mobile laboratory also presumably had a measurement of mobile wind, and yet this measurement has not been mentioned at all. Do these data exist? What problems were encountered with this wind (equipment failure, uncertainty in calculating true winds for moving vehicle, etc)? Depending on the conclusions, item 2 in the Recommendations (p. 15) might be updated. Table 2 might also be updated.**

Newly created Table 2 has been added to clarify when measured winds were available. The mobile laboratory wind measurements were deemed unreliable for analysis due to strong artefacts in the data. We have clarified this point on p. 7 line 31-p. 8 line 4.

*"Unless measured at intensive sites with a tower, wind speed and stability are taken from NOAA's Ready Archived meteorology (https://www.ready.noaa.gov, Rolph et al., 2017) because mobile wind data showed artefacts after corrections for vehicle heading. These artefacts included unreasonably high wind speed and little correlation to stationary tower measurements."*

We have added additional discussion to p. 8 lines 9-13 to clarify that the interpolation to 1-hour is indeed not ideal.

*"While there is uncertainty in using interpolated model wind speed and stability, especially as conditions can change in the morning and evening, the sampling period per site lasted on timescales of a few minutes. At this temporal scale, wind speed, stability, and turbulence statistics are assumed to be constant. During rapidly changing conditions, the model interpolated wind speed and stability could indeed be incorrect. The effects of uncertainties in wind speed and stability are discussed in Sect. 5.1."*

In general, this method (and all advective based emission estimates) work best under conditions that are not fast changing. The discussion of stability was intended to discuss the potential effects of using the wrong stability class. However, rapidly changing conditions are beyond the scope of this work. The mobile-mounted met station data was not used as it was deemed to be inaccurate while the vehicle was moving.

**3. Clarity in describing source data**

**In many parts of the manuscript, it was difficult to understand the nature of the underlying dataset, or equivalently, which hierarchical "level" of data was being used. For example: p. 10, section 4.3: Are these tracer releases? Are they real emissions measured with high N and simulated with all methods?**

This data scheme is based on the inverse Gaussian approach therefore we never used the tracer correlation method. However, all sites in this section were modeled with LES and had a high number of passes. We have attempted to clarify the data collection and calculation schemes used by including Table 2, which summarizes the data types and combining and simplifying figures 6 and 7 into a single figure (now Figure 6). Table S1 has also been moved to the main text as Table 3 with additional data. We have also modified some of the language throughout to better reflect the level of data being referenced. For instance, Table 4 now contains the number of transect that were released.

**Fig 11: Is this purely simulated data? If measured data is available, show a sample transect on the same scale.**

All the data shown in Figure 11 is simulated. This data is intended to compare the models, which simulate at a standard emission rate (1 kg/s) as necessary to compute an emission rate from Eq. 2:

$$Q = \frac{\sum C_{Observation} - C_{Background}}{\sum C_{Model}} \times Q_{ref}$$

therefore the concentrations do not match observation levels. Furthermore, we do not have real data in the vertical direction, thus we cannot show real data that matches most of these subpanels and it would

seem confusing to show the real data on only a few subpanels. The observational data corresponding to this plot are available in Figure 7.

**4. Others**

**Table 1: Add a column to this table describing the type of uncertainty noted (e.g. 95% CI, std dev, etc. See also previous comment about adding in results of this study.**

Unfortunately many authors do not explicitly explain the assumptions and considerations in their uncertainty analyses. In these cases, we assume this is meant to represent a 1 standard deviation interval as this is the most conventional metric. Additionally, some authors described their methodology, but the method is specific to their set-up and not easy to summarize in a table (e.g. Petron et al. (2012) compare a minimum and maximum range). We have emphasized further in the text that this table is meant to summarize our motivation in pursuing this work and not as an exhaustive comparison between the techniques themselves on p. 3 lines 3-11:

*"These techniques have been used in different ways, from direct point source emission estimation, to area source emission rate estimation, and they use data that may span a few hours or years. For example, Kort et al. used data over 6 years at 0.33° resolution covering the entire U.S. to estimate a large emission rate. Karion et al. 2013 reported emissions from a large natural gas field (~60 km diameter) using aircraft mass-balance with data collected in a few hours while Caulton et al. 2014 reported well pad emissions (<1 km diameter) using the same technique. It is also feasible that as new instruments and data processing techniques become available, any of these techniques may be used at spatial and timescales not represented by the works cited here. Table 1 compares the author reported uncertainties for several techniques; this table is intended to illustrate the motivation for this work and not as an exhaustive review of each of these techniques."*

And also regarding the uncertainty on p. 3 lines 13-17:

*"Most of the author reported uncertainties appear to correspond to 1 standard deviation measurements, though, the ways these statistics are derived are not consistent. Notably, Kort et al. (2014) report a 2 standard deviation range and Petron et al. (2012) reports a minimum and maximum range for emissions. In addition, several studies report 95% confidence intervals, such as all ground-based mobile dispersion estimates, likely because of the asymmetric uncertainty that is more easily reported in this manner."*

**p. 7, line 1: What is the purpose of allowing the simulation to use other point sources in order to best simulate the emissions? Are these other point sources input by the user based on observed equipment/sources, or are they automatically generated? What is the reasoning behind comparing multi-point LES to a single-point gaussian simulation at all (as described p. 9, line 10 and Fig 7)? If it is for ease/quickness of data processing, describe this.**

Comparing scenarios with more available data inputs is expected to reduce the uncertainty in the resulting emission rate. For instance, we normally assume a central emission source at a height of 1 m, but if we have more detailed information as to the specific location of the sources, incorporating that should reduce uncertainty in the result. We have reworked Fig. 6 to better illustrate this. More description of why the comparison was done has been added to p. 13 lines 14-16.

*"A schematic of the emission rate calculation strategy is shown in Fig. 6. Generally, the more information available (e.g. source location), the less uncertain the results are likely to be. Results will be compared from different scenarios to address to what extent uncertainty can actually be reduced."*

To address how sources were assigned, the readers have now been directed to Sect. 4.1 on p. 9 line 17-18 as a description seemed out of place in this section.

*"A more detailed description of how sources were selected is presented in Sect. 4.1."*

The description in 4.1 has also been expanded on p. 13 lines 16-19.

*"As on-site access was not available and well pads may contain multiple sources, all large structures were treated as separate point sources. These were visually identified during measurements and exact coordinates were confirmed in Google Earth. The center of all sources was used as a point source. The identified sources were always gas processing units or storage tanks."*

We believe the increased explanation of data availability also helps explain why the comparison is necessary as we are trying to match comparisons to scenarios with different amounts of available data.

**p. 9, line 5: Did scientists have site-access or other means to verify this assumption? If tank batteries are present, tank emissions can often overwhelm wellhead emissions. Is this dealt with in the data somehow?**

In this analysis we did not have site-access, see response to previous comment. We consider emissions from all sources, including tanks as the response to the previous point shows. We have added an explanation of this in Sect. 4.1. as cited in the previous response. It would be impossible to only measure only well-heads with a mobile platform at most sites thus we are only interested in emissions from the well-pad as a whole. We consider this suitable because the goal was to get integrated well-pad level emission rates. Section 5.1 discusses the uncertainty from not knowing the exact location and height of the source.

**p. 13, line 9: A figure showing the distribution, where it is possible to note the mode and 95% confidence intervals, is needed to understand this skewed distribution and should be included as SI**

A sample Monte Carlo distribution has been added as figure S15.

**Minor comments/ typos:**

**Table 3. Describe SS, MS and LES abbreviations in caption.**

The abbreviations in what is now Table 5 have been spelled out to avoid confusion.

**Table 5: It seems like the line for Source Location should differentiate between x, y and z directions, as done in the text, or at least include a range of uncertainties.**

The x and z locations have been separated in what is now Table 6. As discussed in the text, the y direction uncertainty is not expected to be significant. p. 19 line 1-2.

*"Uncertainty in the cross-wind direction is not expected to contribute to significant change in emissions."*

**Table 6: reformat Sources of Uncertainty column (e.g. left-justify) for ease of reading.**

Sources of Uncertainty column in what is now Table 8 has been left-justified.

**Figure 2: Consider breaking up this graphic into more panels because it is impossible to compare individual traces as-is. Also homogenize the vertical scales. There is currently a 3 order of magnitude difference in the scales. Is this correct?**

The Figure 2 caption has been clarified to explain the scale difference, which is due to the Gaussian model using a reference 1 kg/s emission rate. This is the procedure for calculating the unknown emission rate by comparing to the model with a known emission rate. To allow for the requested comparison between plume transects we have added additional subplots to what is now Figure 7 where we feel it is more useful to observe differences in the individual transects as we are talking about the actual data at this point, not describing the calculation process more generally.

**Figure 4: Make the bounds of Yindex the same in all graphs. Furthermore, it would be useful to produce a mixing ratio vs Y index graph (or Scalar vs Yindex) at a height of 3m to show what hypothetical measured data would look like.**

The requested changes in z and y index have been made to Figure 4.

**Figure 5: Is it possible to show uncertainties at 95% confidence on these graphs? This would be more useful than the 10-90th percentiles. Replace "50% percentile" and similar with "50th percentile" throughout the caption and text**

Figure 5 has been redone to improve clarity of the analysis. The original Figure 5 had the requested changes made and was moved to Figure S4.

**Figure 7: expand SS, MS and LES in the graph, or note their meaning in the caption.**

Figure 7 has been combined with Figure 6 in what is now Figure 6. The abbreviations have been explained.

**p. 4, line 13: "Additionally, implementing..." These last two sentences seem out of place in the paragraph discussing uncertainties. Elaborate or move.**

We believe describing the challenge of a controlled release is relevant to discussing how previous work has been implemented. The text has been expanded on p. 5 lines 23-24.

*"As a goal of this work is to identify best practices for quantifying uncertainty, it is important to understand how feasible methods are likely to be."*

**p. 5, lines 1, 5, 9: LICOR is typically written as "Li-COR"**

We have changed LICOR to LI-COR throughout paper.

**p. 7, line 31: briefly define percent difference - what is the "true" value used as denominator.**

The requested change has been made on p. 11 lines 19-20.

*"The 5-95% range of observations for percent difference (pd, always relative to the mean of the compared observations)…"*

**p. 8, lines 2-4 and Figure 5: renumber figure panels so that consecutive letters refer to the same transect interval. As it is, line 2 should read "(b & d)" instead of "(c-d)"**

Figure 5 has been redone. The original Figure 5 had the requested changes made and was moved to Figure S4.

**p. 8 lines 14-19: reference Figure 6 here somewhere**

Reference to Fig. 6 has been added on p. 12 line 13-14.

*"As depicted in Figure 6, the goal of the sampling strategy was to produce more standard sampling sites. . ."*

**p. 10, line 14: which result shows no apparent bias?**

Box plots of the Gaussian with on-site winds show no apparent bias. The text has been clarified to point this out on p. 16 line 27-30.

*"In general, the close agreement across a range of release rates shows no apparent bias in the Gaussian with tower winds results, with the results scattered low and high relative to the release rate and no results significantly different (at the 95% CI) from the release rate."*

**p. 10, line 20 and Fig 10: When discussing Fig 10, the over/underestimate of results as they converge seems important to mention.**

The requested discussion has been added on p. 17 lines 8-11.

*"Additionally, Fig. 10 shows that the IGM rarely overestimates (>2x release rate), but more often underestimates (<0.5x release rate). By this definition the IGM results underestimated the release rate by up to 30% of the time during the release rate using only 1-5 transects."*

**p. 11, Section 5.1: It would be interesting to express these comparisons in relative distances as well.**

Figure S12 has been expanded to include panels showing the change as the distance relative to the perturbation distance of 50 m. We believe this is the relative distance the referee was interested in, but if not please clarify and we will of course add additional figures as necessary.

**p. 12, line 1: reference Table 5 in this section (Sect. 5.1).**

The reference to what is now Table 6 is available in Sect. 5.2 on p. 20 lines 17-18.

"The sources for uncertainty and bias in the Gaussian measurements discussed in this analysis are summarized in Table 6."

**p. 15, line 1: include range, e.g "...standard sampling uncertainty range of 0.05x - 6.0x is greater..."**

The requested change has been made on p. 23 lines 20-23.

*"In addition, our standard sampling uncertainty range is greater than other Gaussian approaches with controlled releases which reported 0.28q – 3.6q and 0.334q – 3.34q (Rella et al., 2015, Yacovitch et al., 2015); the uncertainty range of the multi-transect sites studies here was similar in magnitude (0.05q– 6.5q)."*

Note that x has been replaced with q to reduce confusion when talking about downwind distance (also x) and emission rate.

[revised manuscript text omitted]

25  ~~estimate. We have chosen to average the emissions deduced from individual transects. Alternatively, the multiple transects themselves could be averaged to produce a single emission estimate. The latter method should theoretically be more Gaussian in shape and more comparable to the model, but requires enough transects to produce a Gaussian profile and may not be appropriate for sites with a limited number of transects. However, we analyzed a representative site as a comparison to provide information~~
30  ~~as to what, if any, difference in retrieved emissions may be expected with this method. As an example, Site 3 was chosen to average the multiple transects previously shown in Fig. 2. The averaged plume, shown in Fig. 12, was then used to calculate the emission rate using the IGM approach, also using the model averaged across all transects. The averaged transect emission rate is 1.2 kg hr$^{-1}$, extremely close~~

~~Averaging multiple transects has the benefit of reducing the influence of atmospheric variability on the uncertainty of the measurement; however, longer measurement time is still required due to the need for many transects. Each approach, averaging emissions vs. averaging transects, produces similar final results indicating that averaging method is not a driver of emission uncertainty. Either approach may be acceptable given the constraints and intent of a sampling session. The variability of single transects of emissions may be a useful tool for data quality control while averaged transects may be useful in additional analysis intent on pinpointing the location of an unknown source.~~

[revised manuscript text omitted]

**Site 1**

[Figure]

**Figure S5.** Comparison of three scenarios for Site 1 showing images of (a) single Gaussian, (b) multi-source Gaussian and (c) averaged LES. The comparison of vertical distributions (d) concentrations and (e) fluxes, and horizontal distributions of (f) concentrations and (g) fluxes are also shown.

[Figure]

**Figure S6. Comparison of three scenarios for Site 2 showing images of (a) single Gaussian, (b) multi-source Gaussian and (c) averaged LES. The comparison of vertical distributions of (d) concentrations and (e) fluxes, and horizontal distributions of (f) concentrations and (g) fluxes are also shown.**

[Figure]

**Figure S5S7.** Comparison of three scenarios for Site 4 showing images of (a) single Gaussian, (b) multi-source Gaussian and (c) averaged LES. The comparison of vertical distributions of (d) concentrations and (e) fluxes, and horizontal distributions of (f) concentrations and (g) fluxes are also shown.

[Figure]

**Figure S6S8.** Comparison of scenarios using different diffusion models for Site 1 showing images of the comparison of vertical distributions of (a) concentration, (b) flux and of the horizontal distributions of (c) concentration and (d) flux.

[Figure]

**Figure S7S9.** Comparison of scenarios using different diffusion models for Site 2 showing images of the comparison of vertical distributions of (a) concentration, (b) flux and of the horizontal distributions of (c) concentration and (d) flux.

[Figure]

5 **Figure S10.** **Comparison of scenarios using different diffusion models for Site 3 showing images of the comparison of vertical distributions of (a) concentration, (b) flux and of the horizontal distributions of (c) concentration and (d) flux.**

[Figure]

**Figure S9S11.** Comparison of scenarios using different diffusion models for Site 4 showing images of the comparison of vertical distributions of (a) concentration, (b) flux and of the horizontal distributions of (c) concentration and (d) flux.

[Figure]

a

**Source Location Uncertainty Effect**

b

[Figure]

**Figure S12.** (a)  Modeled CH₄ at three source locations at different downwind x positions assuming 3 m receptor height, 1 m source height, neutral stability and 1.5 m s⁻¹ wind speed. (b) Change in % of the modeled CH₄ in each of the scenarios and base scenario (S=0). (c) The same as panel (a), normalized by the perturbation distance of 5 0m. (d) The same as panel (b), normalized by the perturbation distance of 50 m.

[Figure]

[Figure]

**Figure S13.** (a) Modeled CH$_4$ using 8 source heights assuming 3 m receptor height, neutral stability, and 1.5 m s$^{-1}$ wind speed. (b) Change in % between the modeled CH$_4$ for each of the scenarios and the base scenario (h=1 m).

[Figure]

[Figure]

Figure **S14. Change in percent** between  modeled CH$_4$ **between** +/- one stability class **and base scenario** for (a) Class B, (b) Class C, (c) Class D, and (d) Class E assuming 3 m receptor height, 1 m source height, and 1.5 m s$^{-1}$ wind speed.

[Figure]

**Figure S15. Monte Carlo simulations using 1,000 replicates for the standard sampling, 1 transect case showing (a) normalized observation distribution, (b) normalized model distribution and (c) normalized emission rate distribution.**

---

## Author Response (AR2)

Response to Referee

We thank the referee for their comments and appreciate the helpful remarks on the revised manuscript. The additional comments from the referee helped us to clarify the remaining points of confusion including rewording the title. We show the referee's comments in bold below with our response in normal text and any quoted text in italics.

**I thank the authors for their thoughtful and informative responses to my initial review comments. They have clarified the micrometeorological issues I raised to my satisfaction, and I enjoyed reading the enhanced discussion of these measurement methods and the associated possible choices for treating atmospheric dispersion. I acknowledge that the choices they have made are reasonable and defensible.**

**I also appreciate the addition of Table 7, and the effort to bring together the uncertainty assessment in an organized and coherent fashion. I do, however, have some remaining suggestions related to the uncertainty assessment and its presentation.**

**Major comments.**

**1. The authors note that they discard cases with plume maxima whose mole fraction is less than 50 ppb because this is below their detection limit. This raises an overall serious methodological concern. Many industrial sites may have small or no emissions. Measurements of very small emissions are very important, even if the relative uncertainty (% of emission rate) may be very large. Since the authors survey nearly 1000 sites, I do not understand the logic here. The authors must explain this more clearly in the text. Perhaps they do not want to quantify a fractional uncertainty for these cases, and that their uncertainty estimates do not apply to these cases. If so, they should define this limitation of their uncertainty estimates clearly and in the abstract, and include this caveat in their comparison to other measurements. It would be very unfortunate, however, if they are arguing that the IGM approach cannot be applied to cases with very small emission rates, since this would hamstring our ability to sample a broad distribution of emitters and non-emitters.**

First, this manuscript does not deal with the results/analyses of the full data set but rather the methodology and justification of the possible sources of error for the dataset. Indeed, we do not neglect these "zero" emissions in the full analyses for the very reason noted by the referee.

The 50 ppb threshold is used to discriminate between results that have a clearly measurable signal. Data below this threshold are not thrown out in the analyses, but rather are recorded as events indistinguishable from 0 (even though the results are still calculated). Again, we emphasize that those emissions that are below the detection limit will not be neglected. We are including these results in our macro analysis of all of the data, as removing them would indeed lead to a large bias for all sites. However, since the uncertainty analysis is being designed for sites with measurable signal, this manuscript focuses on those emissions in this range, where indeed we can distinguish from zero.

The text has been clarified on p. 18 lines 4-6 to indicate how the 50 ppb threshold was used. *"Sites with enhancements less than 50 ppb were identified as having emissions that were indistinguishable from zero, when considering other sources of error. These data were designated as non-emitting sites in the emission database."*

While we are arguing that the IGM method is not suitable for very low emissions due to its large uncertainties, this is not an issue unique to this method. Every method has a limit of detection. Other methods such as tracer release and mass-balance do not necessarily perform any better in this regard (e.g. tracer release requires accurate location and one leak location – otherwise it could have significant errors). For reference, the median LOD was ~0.1 kg hr$^{-1}$ for our results in the Marcellus, which is quite low and an order of magnitude lower than the average emission rate. The discussion of the use of the limit of detection (including how the emissions are affected by different methods to screen the data) are the subject of a forthcoming manuscript dealing with the results of the ~1000 sites. Due to the length of this current manuscript and the breadth of topics to be discussed, we did not feel it possible to add that discussion to start analyzing individual and cumulative emissions in the current manuscript (which entails additional details/discussions/analyses).

The abstract has been clarified on p. 2 line 1-3. *"The uncertainty bounds calculated for this work for sites with > 50 ppb enhancements were 0.05q-6.5q (where q is the emission rate) for single transect sites and 0.5q-2.7q for sites with 10+ transects."*

**2. It is unfortunate that the authors do not deal with the overall accuracy of their mean or median emission rates. This is an important problem, and the authors have some indications of potential sources of bias in their measurements, and it would aid in their interpretation of the nearly 1000 emission rate measurements that they have collected. I understand, however, that this is a difficult problem that may be beyond the scope of their study.**

Again, here we must clarify the focus of this study is NOT to report the shape of the distribution and the error of the distribution (and cumulative emissions) of all well pads. This manuscript is concerned with the uncertainties of individual well pad emissions measurements and the methods used to justify the stated uncertainties. Such a study has never been undertaken in the literature for mobile laboratories at this level of detail (i.e. using information obtained from multiple transects of the mobile laboratory, transects with additional micrometeorological measurements, large eddy simulation of sites, and controlled tracer releases). With the emission estimates for well pads information justified, a forthcoming manuscript addresses how uncertainties in the individual well pad emissions factor into the overall leak rate of the basin.

While we understand the reviewer's desire to consider all sources of error in the total distribution, we again emphasize that we were specifically focused on understanding the inherent limitations in the mobile Gaussian plume methodology for an individual well pad, not the aggregate compilation of such results. We note that our methodology is broadly applicable to other point-like sources, instead of specifics of our sampling procedure (which would change for any campaign). The reasoning behind this is that the mobile Gaussian plume methodology is attractive for rapid plume quantification: no site access needed, rapid monitoring, and easy adaptability to many vehicles. However, it is less well characterized than tracer releases, for instance.

To address the confusion over Main point 1 and Main point 2, we have reworded the title to reflect the specific aim and context of this study more clearly: *"Quantifying Uncertainties from Mobile Laboratory Derived Emissions of Well Pads Using Inverse Gaussian Methods."*

**3. The link between the results presented, and Tables 6 and 7, is murky and should be clarified.**

**For example, the text on page 18, line 13, describing Table 6, reads:**

*"These include the uncertainty in the Gaussian diffusion constant by comparing to LES calculated diffusion, uncertainty due to source location and height and uncertainty due to wind speed and stability class. In addition, the LES was used to observe bias in the Gaussian derived concentration distributions and the controlled release was used to evaluate bias in both the Gaussian and LES results. Finally, the LES was used to determine the optimum sampling pattern to constrain actual atmospheric variability."*

**This text should note the portions of the results that are the bases for these statements. E.g. "…the LES was used (section X) to observe bias in … (Figure Y) …and the controlled release (section Z)…"**

**In addition, the experiments conducted from section 3.1 up to this point are difficult to follow in that their objective is not clear. They appear to me to be constructing the basis for Table 6, but it is hard to match the elements of Table 6 (and their propagation into Table 7) with the contents of these sections. Please state clearly, for each section, the objective, and the primary result, so that its realization in Table 6 is evident. I would also recommend more logic to the section numbering and names to make this progression easier to follow.**

Several clarifications have been made with regard to naming in Sect. 4 and the pertinent sections are now explicitly referenced in the text in Sect. 5.2 on p. 19 lines 5 – 9. *"These include the uncertainty in the Gaussian diffusion constant by comparing to LES calculated diffusion (Sect. 4), uncertainty due to source location and height and uncertainty due to wind speed and stability class (Sect. 5.1). In addition, the LES was used to observe bias in the Gaussian derived concentration distributions (Sect. 4.4) and the controlled release was used to evaluate bias in both the Gaussian and LES results (Sect. 4.3). Finally, the LES was used to determine the optimum sampling pattern to constrain actual atmospheric variability (Sects. 3.1 and 3.2)."*

Table 6 has been annotated to explain the origin of the numbers.

**Details:**

**1. page 8, line 25. paragraph has some minor English grammar problems.**

The paragraph has been edited.

**2. page 9, line 8. larger, not large**

The line has been corrected.

**3. page 11, lines 15-20. please define these statistics when they are presented. RSD of what? mean deviation is explained the second time it is presented.**

The text has been clarified on p. 11 lines 21-26 to indicate which populations the statistics refer to.

*"For the population of standard sampling sites with multiple passes, the average rsd of emissions from repeat passes was 67% and the average maximum percent difference (highest observational deviation from the mean of repeat measurements) between emission estimates at a single site was 58%. The average rsd of the population of 53 emissions estimates for the replicate sampling sites was 77% and the average maximum percent difference (highest observational deviation from the mean of repeat measurements) was 150%. The rsd of emissions from repeat passes ranged from 12% to 260%."*

**3. Same spot. These results, and comparison to LES experiment, might be clearer in a table. Authors' discretion.**

The text has been clarified on p. 11 lines 21-26 to indicate which populations the statistics refer to.

*"For the population of standard sampling sites with multiple passes, the average rsd of emissions from repeat passes was 67% and the average maximum percent difference (highest observational deviation from the mean of repeat measurements) between emission estimates at a single site was 58%. The average rsd of the population of 53 emissions estimates for the replicate sampling sites was 77% and the average maximum percent difference (highest observational deviation from the mean of repeat measurements) was 150%. The rsd of emissions from repeat passes ranged from 12% to 260%."*

**4. page 11, lines 13-15. Replicate sampling sites were chosen to test the sensitivity of the emissions estimates to changing atmospheric conditions? Where is this presented in the manuscript?**

The actual line reads *"These replicate sampling sites were generally chosen at the beginning and end of each four hour sampling period to observe changes in variability over the course of the sampling period that may be due to changes in atmospheric conditions."*

Thus, the replication at different times was to ensure we had statistics for the range of conditions we sampled. The sensitivity of the analysis to inaccurately classified stability is discussed in Sect. 5.1.

**5. The ordering of sections 3 and 4 is difficult to follow. What is the point of section 3.2? Why is the source strength determination strategy (section 4.1) presented after the field implementation (of source strength determination)? (section 3.2) is presented? See major comment 3, above.**

Section 4 was designed to detail results from the procedures outlined in sections 2 and 3. Section 3.2. is designed to summarize the implementation of the sampling strategy. We feel there is useful discussion of how the actual data compared to the sampling strategy discussed in Sect 3.1 so have elected to keep it as is. Section 4 has been renamed to *"Results of Source Strength Determination"* and Section 4.1 has been renamed to *"Strategy for Comparing LES and IGM Results."*

**6. What is the purpose of section 4.1? What are "LES emissions" (page 13, lines 1-2)? This seems out of place. See major comment 3, above.**

Section 4.1 has been renamed to *"Strategy for Comparing LES and IGM Results"* to clarify its purpose and the difference between IGM and LES emissions.

**4. If section 4.1 is intended to explain how the authors solve for an emission rate, it would help to have one of the equations solve for the emission rate.**

We have referred the reader back to Sect. 2.3 for the discussion of the emission rate calculation on p. 12 line 5.

**5. Page 13, "Meandering" discussion. This discussion lacks the implicit issue of the distance downwind from the source where dispersion is measured. The rule of thumb of maximum eddy size that is relevant must be limited to dispersion relatively close to the source relative to the size of the large eddies. Far enough downwind, the largest eddies in the boundary layer will mix and disperse the plume. Since the authors' measurements are all (I believe) close to the source compared to the size of these large eddies, I am comfortable with their rule of thumb arguments about the sizes of eddies they find to be relevant. Question: Are the dispersion coefficients employed in the Gaussian model specific to this "near-field" dispersion argument? In any case, I am satisfied with this discussion and the tests conducted.**

We thank the referee for their comments. We have modified a statement to clarify that this discussion is intended for nearfield dispersion on p. 13 lines 11-12. *"As terminology in boundary layer meteorology can be ambiguous and is not always used consistently, the specific meaning of 'meandering' is discussed here in the context of the nearfield regime applicable to this work."*

In addition we have moved the following sentence (previously the last sentence of the first paragraph of Sect 4.2) up to p. 13 lines 16-17. *"Thus we clarify that large scale 'meandering' as used in this work corresponds to the effect of larger (>> plume diameter) scale motions (>~100 m)."*

 The Gaussian dispersion parameters are a function of distance as shown in Appendix A. Individual details for each model can be found in the cited literature, but for the Briggs model used the parameters do include regimes for 'near-field' (>100 m and <10,000 m) and far field >10,000 m. The Briggs downwind range has been added to p. 7 line 23.

**6. Page 15, line 1. SS Gaussian is not defined.**

SS Gaussian was defined earlier is Sect. 4.1 (starting on p. 12). The description on p. 12 line 11 was expanded to make this more obvious. *". . . (i) the base scenario is the IGM approach used for all sites that assumes there is a single- source Gaussian at the well-head at 1 m (SS Gaussian). . . "*

**7. Table 3. Please define the error bars for wind speed and wind direction.**

An indication of the 1 sigma level has been added to Table 3.

**8. Table 4. What are the "NOAA winds" vs. "tower winds?" It would help if these were explained. And which mean wind was imposed on the LES? My apologies if I am missing something simple here. (I think I found that tower winds are used, apologies…but maybe make this easier to find. And don't you need to impose a wind profile, not just a point?)**

'Tower wind' has been replaced with 'measured wind' throughout to make the meaning more obvious. The Gaussian equation does not use a wind profile. The LES generated wind is scaled to the tower winds at the height of observation and the profile is generated from LES itself. The LES description on p. 13 lines 2-4 has been clarified. *"The non-dimensional wind speed ($u_{LES}$) is divided by the dimensionalized ($u_{LES}^D$) wind speed, which is set equal to the on-site tower measurements at the measurement altitude. The LES generated winds thus retain a vertical wind profile."*

**9. Section 4.3. Controlled releases. What is the conclusion? The IGM does not converge on the true flux. The LES is biased for the test site. NOAA winds are biased relative to the tower-measured winds. How is this reflected in the conclusions? What is learned? See major comment 3, above.**

It is unclear what the referee is referring to. Given the reported uncertainty bounds of these experiments, the IGM *does* converge on the true flux when using measured winds as shown in Figure 9. As stated in the manuscript, the conclusion is that the results are not biased when the wind is accurate. As we discuss in section 5.1, the comparison between NOAA winds and the measured winds for many sites in this area does not show any apparent bias on average. This point is reiterated in the conclusion section Sect. 6, p. 21 lines 3-7:

 *"Using the LES and a controlled release, we confirm that the Gaussian model performs well when in-situ winds are available. The NOAA estimated winds can be a source of error, but we did not observe a systematic difference between the NOAA and in-situ winds, thus no sources of bias using our approach are expected on average. We note that this result is valid for this general area, and other locations, which have different challenges and data density may differ. Area specific analysis, or on-site winds, should always be used to reduce bias."*

If the referee is referring to Fig. 10, where the convergence of the *mean* is examined for various numbers of transects, the box plots do not represent the total uncertainty for each scenario. A note of clarification has been added to p. 15 line 30-32 *"Note that the boxplots correspond to the distribution of means obtained from randomly selecting the indicated number of transects, not total uncertainties on the means for each number of transects."*

**10. Section 4.4 Is the point that the number of sources is not known? Thus, is MS vs. SS roughly a factor of two source of uncertainty in emissions estimates, unless the true number of sources can be determined? The experiment being conducted here is not clear. See major comment 3, above.**

Indeed, the number of sources is likely to be unknown, especially as there can be multiple sources on even one well. We believe the renaming of Sections 4 and 4.1 help clarify the purpose of this section. The various scenarios are designed to compare the ways the IGM could be used depending on how much information is available. The text has been clarified on p. 16 line 7-8 *"This indicated there is consistency in retrieved emissions from a range of scenarios despite using different amounts of information to set up the sources."*

In addition the discussion has been expanded on p. 16 lines 23-27. *"However, if a difference in vertical dispersion exists, this can significantly change the retrieved emission rate. There is considerable difference between the SS and MS results, for instance. Without observations at multiple heights, it is impossible to verify which assumption is correct. However, the SS results are typically closer to the LES results in these simulations, suggesting that this is the better approach than assuming source height and locations."*

**11. Page 17, lines 10-15. Background uncertainty. As I understand it, the authors 1) used the minimum value measured on a transect as the background and 2) compared this to using the lowest 2% of measurements on a transect as the background. And from some method that I cannot determine, an uncertainty of 5 ppb from all 1000 sample sites was determined.**

The text has been clarified on p. 17 line 33- p. 18 line 1. *"When comparing samples with repeat transects, the backgrounds identified for each transect had a median standard error of 5 ppb."*

**But, sites with a plume of less than 50 ppb were eliminated from the measurements since this was said to be below the detection limit of the system. This paragraph raises questions. First, I agree that if the plume size is large compared to fluctuations in the background, this problem is minimized. Identifying the background, however, as a % of the plume mole fraction (lowest 2%) is not sound. Fluctuations in the background may have nothing to do with the plume being measured. Thus I question this approach for quantifying the uncertainty in the background. A value based on deviations from the Gaussian shape away from plume center would be more defensible.**

There is some inherent arbitrariness is defining the background, thus we have compared two options (the minimum of the transect and the lowest 2$^{nd}$ percentile of the transect). Both of these were designed to identify the regions of the transect away from the plume. Transects were always designed to extend beyond the plume enhancement area. Again, for the majority of the data, this is a small error source that would not be large even if some other method of determining the background was used.

**I expect, as the authors suggest, that this will remain a relatively small source of uncertainty for large magnitude plumes. More worrisome is the note that plumes with a maximum mole fraction enhancement of less than 50 ppb are discarded because these are below the detection limit of the system. First, I believe that the instrumental precision is much better than 50 ppb, so I'm puzzled by this value. And if the background uncertainty is only 5 ppb, what makes the detection limit 50 ppb? Most importantly, in a survey of many, many sites, some / many may have very small emissions. You may have large uncertainty in the determined flux, but that does not make the measurement invalid. And retaining small emission rate measurements, even if the relative uncertainty is large, is an important issue for surveying emissions from industrial facilities. How many of the ~1000 sites fall into this category? Would the results of this work (uncertainty in emissions measurements) change significantly if these sites were included?**

The 50 ppb threshold is used to discriminate between results that have a measurable signal that is indistinguishable from zero considering all sources of error (see response to Main Point #1). The variation in the background is much greater than the instrumental noise (few ppbv in general) making the instrumental noise an inappropriate statistic to distinguish peaks. Data below this threshold are not thrown out, but are recorded as event indistinguishable from 0 even though the results are still calculated. As the contribution of the uncertainty of the background to the total uncertainty increases as the peak magnitude decreases, the 50 ppb threshold was selected because it reliably retained values where the emission was actually distinguishable from 0. The importance of sites that are below the detection limit has not been neglected. We are including these results in our macro analysis of all of the data as removing them would indeed lead to an inaccurate mean for all sites (about 30% of all sites are classified as "zero" in this fashion). However, since the uncertainty analysis is being designed for sites with measurable signal, we believe this is the most accurate approach to estimating the uncertainty

range for these values in this manuscript. These results are not affected by the choice to impose a limit of detection threshold. We note that the analysis of the mean emission rate is the subject of a separate forthcoming manuscript, and will be addressed there.

**12. Page 19, lines 4-5. The inputs in Table 7 are all assuming a Gaussian distribution of errors, are they not? This is not clear. The text notes that errors may not be Gaussian in nature.**

The distribution are either Gaussian or uniform (for stability and height). This is noted in Table 7 in parentheses in each row of column 1.

**13. Page 19, line 5. If emissions from equation 2 are less than zero, is that a valid outcome that should be retained for a proper description of the measurement statistics? This seems linked to the practice of eliminating plume maxima less than 50 ppb. Mean values may be biased if valid small or negative plume concentrations are eliminated, and a negative emission rate may be a valid result from a noisy measurement system.**

This has been addressed in the responses to Main point 1 and point #11.

**14. page 19, line 16. The computational demands associated with LES vs. a Gaussian plume model is not a new finding. It is not worthy of much text.**

We agree this is not a new finding, but have opted to keep the text in for those unfamiliar with the procedure.

**15. Page 20, lines 10-11. Reanalysis winds differ by source, and this one comparison effort is limited in terms of locations and times of day. A note of caution is warranted.**

Indeed, the reanalysis data will be affected by the quality of the source data and may be especially inadequate in complex terrain with sparse coverage.

We have included the following caveat on p. 21 line 5-7: *"We note that this result is valid for this general area, and other locations, which have different challenges and data density, may differ. Area specific analysis, or on-site winds, should always be used to reduce bias."*

**16. Page 21, line 5. This conclusion that the approach can identify emissions that are orders of magnitude larger than the mean is relatively weak, and not a new finding. It is disappointing that the manuscript doesn't deal with biases in mean values when accumulating data over tens of transect and hundreds of sites, which seems to be the focus of this overall research effort.**

Again, the point of this manuscript is focus on uncertainties of individual well pads, not the aggregate distribution. It is not to demonstrate that we can quantify emissions larger than the mean. We note that even with the conservative 50 ppb threshold, we sampled non-zero emissions from about 2/3rds of the well pads. Smaller emissions certainly were observed, but their uncertainties were not analyzed as they would be indistinguishable from zero when including all sources of error/bias as described in this manuscript. As noted above, we have tried to clarify the point of this manuscript is to quantify the uncertainties of individual measurements and not the distribution.

**17. Data availability. I understand that the data will become available upon publication, and that this text will be edited. That seems acceptable to me.**

We thank you for your understanding, and we fully support open-access to data as a field measurement group.

[revised manuscript text omitted]